# Targeting GLP-1 receptor trafficking to improve agonist efficacy

Ben Jones[1], Teresa Buenaventura[2], Nisha Kanda[2], Pauline Chabosseau[2], Bryn M. Owen[1], Rebecca Scott[1], Robert Goldin[3], Napat Angkathunyakul[3,4], Ivan R. Corrêa Jr [5], Domenico Bosco[6], Paul R. Johnson[7], Lorenzo Piemonti [8,9], Piero Marchetti[10], A.M.James Shapiro[11], Blake J. Cochran[12,13], Aylin C. Hanyaloglu [14], Asuka Inoue[15], Tricia Tan[1], Guy A. Rutter [2], Alejandra Tomas[2] & Stephen R. Bloom[1]

Glucagon-like peptide-1 receptor (GLP-1R) activation promotes insulin secretion from pancreatic beta cells, causes weight loss, and is an important pharmacological target in type 2 diabetes (T2D). Like other G protein-coupled receptors, the GLP-1R undergoes agonist-mediated endocytosis, but the functional and therapeutic consequences of modulating GLP-1R endocytic trafficking have not been clearly defined. Here, we investigate a series of biased GLP-1R agonists with variable propensities for GLP-1R internalization and recycling. Compared to a panel of FDA-approved GLP-1 mimetics, compounds that retain GLP-1R at the plasma membrane produce greater long-term insulin release, which is dependent on a reduction in β-arrestin recruitment and faster agonist dissociation rates. Such molecules elicit glycemic benefits in mice without concomitant increases in signs of nausea, a common side effect of GLP-1 therapies. Our study identifies a set of agents with specific GLP-1R trafficking profiles and the potential for greater efficacy and tolerability as T2D treatments.

[1] Section of Investigative Medicine, Imperial College London, London W12 0NN, UK. [2] Section of Cell Biology and Functional Genomics, Imperial College London, London W12 0NN, UK. [3] Centre for Pathology, Imperial College London, London W2 1NY, UK. [4] Department of Pathology, Faculty of Medicine, Ramathibodi Hospital, Mahidol University, Bangkok 10400, Thailand. [5] New England Biolabs, Inc., Ipswich 01938 MA, USA. [6] Department of Surgery, University of Geneva, Geneva CH-1211, Switzerland. [7] Nuffield Department of Surgical Sciences, University of Oxford, Oxford OX3 9DU, UK. [8] Diabetes Research Institute (HSR-DRI), San Raffaele Scientific Institute, Milan 20132, Italy. [9] Vita-Salute San Raffaele University, Milan 20132, Italy. [10] Department of Clinical and Experimental Medicine, Islet Cell Laboratory, University of Pisa, Pisa 56124, Italy. [11] Clinical Islet Laboratory and Clinical Islet Transplant Program, University of Alberta, Edmonton T6G 2C8 AB, Canada. [12] Section of Renal and Vascular Inflammation, Imperial College London, London W12 0NN, UK. [13] School of Medical Sciences, UNSW Sydney, Sydney 2052 NSW, Australia. [14] Department of Surgery and Cancer, Imperial College London, London W12 0NN, UK. [15] Tohoku University, Sendai 980-8574, Japan. These authors contributed equally: Alejandra Tomas, Stephen R. Bloom. Correspondence and requests for materials should be addressed to G.A.R. (email: g.rutter@imperial.ac.uk) or to A.T. (email: a.tomas-catala@imperial.ac.uk)

Many G protein-coupled receptors (GPCRs) undergo agonist-mediated endocytosis[1]. Surprisingly, this process does not always result in the termination of intracellular signaling, with several receptors known to generate responses from the endosomal compartment[2–6]. Control of receptor trafficking might therefore be a useful strategy to enable sustained signaling, with significant implications for drug development[7].

In this study, we have investigated the role of receptor trafficking in glucagon-like peptide-1 receptor (GLP-1R) agonism, an important treatment modality for type 2 diabetes (T2D) which improves pancreatic beta cell function and insulin sensitivity[8]. The GLP-1R is rapidly internalized when activated by its cognate agonist[9], but the effect of internalization and subsequent post-endocytic trafficking on overall GLP-1 responses is not clear. Sustained signaling by internalized GLP-1Rs has been reported, but without increasing insulin release[10]. The latter study also identified lysosomes as a major post-endocytic GLP-1R destination, raising the possibility that prolonged agonist exposure might result in GLP-1R degradation. In contrast, a proportion of GLP-1Rs is recycled back to the plasma membrane (PM)[9], an important resensitization mechanism[11].

Here we develop a series of peptides closely related to the GLP-1 homolog exendin-4, used clinically as exenatide[12], but with widely varying trafficking properties. We use these to establish a robust relationship between GLP-1R trafficking and insulin release in a manner not predicted by the standard pharmacological potency testing for cyclic adenosine monophosphate (cAMP), a primary second messenger coupling GLP-1R activation to insulin secretion[13]. We examine the role of receptor binding kinetics and β-arrestin-biased signaling in the observed trafficking profiles, identifying a linked set of agonist characteristics optimally suited for insulin secretion, not shared by GLP-1R agonists in the current clinical use. We find that β-arrestin recruitment to GLP-1Rs during sustained agonist exposure has the opposite effect on insulin release to the known positive role of β-arrestin-1 during acute GLP-1R stimulation in beta cells[14]. We also uncover how the rate of receptor agonist dissociation within the endosomal compartment predicts the rate of receptor recycling, itself a key determinant of sustained insulin secretion.

Finally, we explore the therapeutic potential of these peptides in a mouse model of T2D, uncovering a divergence between agonist-specific insulin release and appetite reduction. Nausea is a side effect which affects 30–50% of patients taking GLP-1R agonists at clinically licensed doses[15], with higher doses glycemically more effective but consistently associated with unacceptable tolerability[16–19]. By selectively augmenting insulin release, modulation of GLP-1R trafficking may be a viable strategy to achieve greater metabolic control in T2D without increasing the rate of unwanted side effects, such as nausea.

## Results

**GLP-1R trafficking controls pharmacological insulin release.** Interaction between the surface regions of receptor transmembrane helices and the agonist N-terminus is critical for the activation of class B GPCRs, including the GLP-1R[20]. Based on this, we synthesized a panel of exendin-4 analogs with single amino acid substitutions close to the N-terminus, which we hypothesized could modulate receptor trafficking and/or signaling (Supplementary Fig. 1a). Using the SNAP-tag system, in which the GLP-1R N-terminus contains a small genetically encoded tag to allow specific labeling of surface receptors, we measured the dose responses for the agonist-induced cell surface loss of human GLP-1R in CHO-K1 cells, identifying the analogs with different net internalization efficacy than the reference compound exendin-4

(Supplementary Fig. 1b). When these compounds were tested in INS-1 832/3 beta cells[21] with a prolonged incubation to mimic in vivo drug exposure, we found that compounds exhibiting higher internalization also had reduced maximal insulin release (Fig. 1a, Supplementary Fig. 1c). Several compounds with reduced internalization exhibited improved insulinotropic efficacy vs. exendin-4. To avoid identifying a species-specific effect, we also used MIN6B1 beta cells[22] and found a similar relationship between internalization and insulin release, albeit less marked (Fig. 1b, Supplementary Fig. 1c). Notably, this effect was not apparent with shorter incubations (Fig. 1c, d). Furthermore, the potential for this therapeutically desirable property was not suggested from the measurement of acute cAMP responses in CHO-GLP-1R cells, a standard in vitro metric for GLP-1R agonist performance in drug development[23], where the more insulinotropic compounds displayed reduced potency vs. exendin-4 (Supplementary Fig. 1b, c).

One peptide with reduced internalization and improved insulinotropism, exendin-phe1, and one with opposing characteristics, exendin-asp3, were selected for further studies (Fig. 1e, f). The increased internalization of exendin-asp3 vs. exendin-4 was relatively small but reproducible. Additional dose response and kinetic analyses indicated that the differences in insulin release emerged only after several hours of incubation (Fig. 1g–i). Greater insulin release with exendin-phe1 was unlikely to result from cross-reactivity with other incretin receptors, as it was blocked with the GLP-1R orthosteric antagonist exendin(9-39) (Supplementary Fig. 2a). In accordance with its reduced capacity for stimulating insulin release, exendin-asp3 failed to reduce apoptosis following glucolipotoxicity[24] or endoplasmic reticulum stress[25], both involved in T2D pathogenesis. However, anti-apoptotic responses to exendin-phe1 were no greater than for unmodified exendin-4 (Supplementary Fig. 2b, c).

Overall, these findings suggests that under pharmacologically relevant conditions, GLP-1R agonists that induce less internalization achieve greater maximal insulin release.

**Endocytosis and post-endocytic sorting of GLP-1R.** To better characterize the agonist-related differences in GLP-1R trafficking, we performed further experiments with SNAP-GLP-1R stably expressed in CHO-K1 and MIN6B1 cells. Using diffusion enhanced resonance energy transfer (DERET[26]), we observed rapid loss of cell surface SNAP-GLP-1Rs when exposed to exendin-4 and exendin-asp3, but much slower internalization with exendin-phe1 (Supplementary Fig. 3a, b). Similar results were seen by the flow cytometric measurement of internalized receptor reversibly labeled with fluorescent SNAP-tag probes prior to stimulation, with residual surface receptor probe removed before detection[27], in MIN6B1 (Fig. 2a, b) and CHO-K1 cells (Supplementary Fig. 3c). The findings were corroborated by confocal microscopy (Fig. 2d and Supplementary Fig. 3e).

In the latter assays, performed to measure internalization at earlier time-points, the modestly greater internalization originally seen with exendin-asp3 (Supplementary Fig. 1b, performed with a longer incubation of 90 min), was not replicated. An explanation for this discrepancy was suggested by the differences in SNAP-GLP-1R recycling, reduced with exendin-asp3 vs. exendin-4 (Fig. 2c, Supplementary Fig. 3d and 3f, g), which might result in progressively greater loss of cell surface receptors over time, despite similar acute endocytosis rates. Recycling was in contrast rapid with exendin-phe1. In order to monitor the intracellular translocation of agonists, as well as GLP-1Rs, we used fluorescently labeled exendin-4, exendin-phe1, and exendin-asp3 conjugates, with fluorescein isothiocyanate (FITC) at position K12, which we previously found useful for labeling GLP-1R[28]. Potencies for all

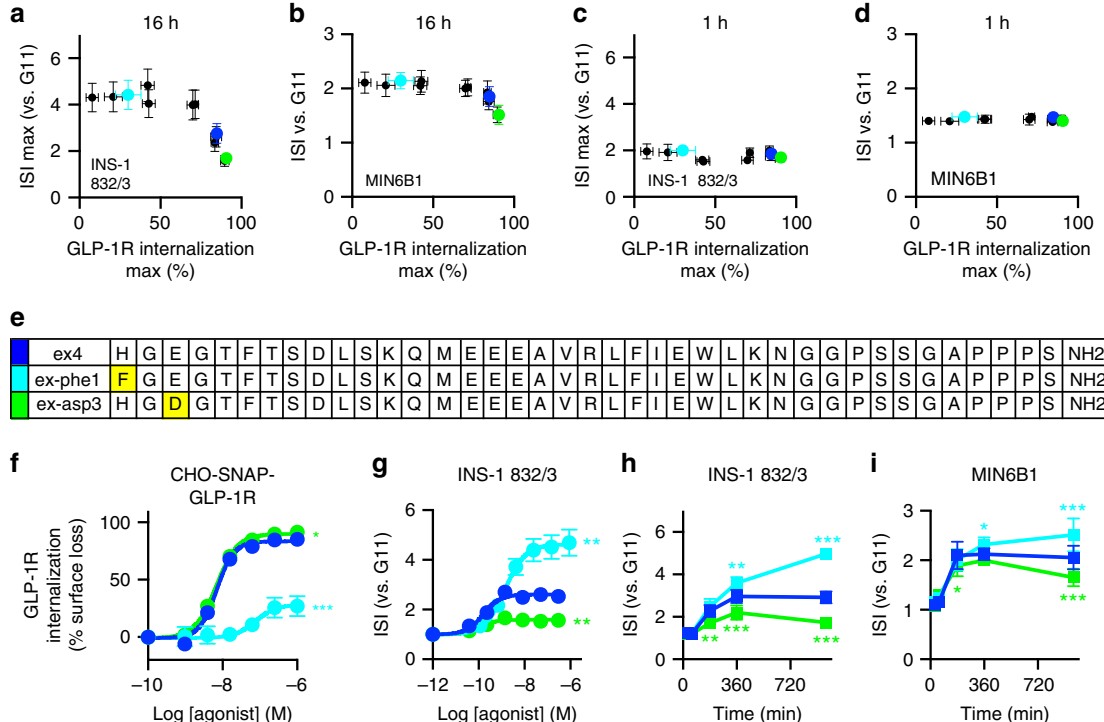

**Fig. 1** Insulin secretion is predicted by agonist-induced GLP-1R endocytosis. **a** Relationship between maximal agonist-induced SNAP-GLP-1R internalization in CHO-SNAP-GLP-1R cells (90 min, $n = 5$) and maximal prolonged insulin secretion in INS-1 832/3 cells (16 h, $n = 6$), expressed as insulin secretion index (ISI), i.e., fold increase vs. 11 mM glucose (G11) alone; full data in Supplementary Fig. 1. **b** As for **a**, but insulin secretion with 100 nM agonist in MIN6B1 cells, $n = 7$. **c**, **d** As for **a**, **b**, respectively, but over 1 h, $n = 5$. **e** Sequences in single letter amino acid code for exendin-4 (ex4), exendin-phe1 (ex-phe1), and exendin-asp3 (ex-asp3). **f** Net GLP-1R internalization (90 min) in CHO-SNAP-GLP-1R cells, full data in Supplementary Fig. 1, $n = 5$, four-parameter fit and statistical significance for $E_{max}$ via one-way ANOVA with Dunnett's test vs. exendin-4. **g** Prolonged insulin secretion in INS-1 832/3 cells, 16 h, $n = 5$, separate experiments to Supplementary Fig. 1, four-parameter fit and statistical significance for $E_{max}$ via one-way ANOVA with Dunnett's test vs. exendin-4. **h** Time-course insulin secretion in INS-1 832/3 cells, $n = 5$, two-way ANOVA with Dunnett's test vs. exendin-4. **i** As for **h**, but in MIN6B1 cells, $n = 5$. Agonists applied at 100 nM, except where indicated. *$p < 0.05$, **$p < 0.01$, ***$p < 0.001$. Error bars indicate standard error of mean (SEM)

FITC-conjugates were within 1 log unit of the unmodified ligand (Supplementary Fig. 4a, b); exendin-phe1 was most affected by the introduction of the FITC group. In keeping with the fast recycling rate with exendin-phe1, immunofluorescence analysis showed increased exendin-phe1-FITC accumulation in Rab11-positive recycling endosomes[29], compared to exendin-4-FITC and exendin-asp3-FITC (Fig. 2e). As FITC fluorescence is pH-dependent, we analyzed the pH differences in the internalized FITC-agonist environment by measuring their fluorescence in cells treated with or without bafilomycin, a vacuolar-type $H^+$ ATPase which restores neutral endosomal pH[30]. The relative signal change induced by bafilomycin for each ligand was used to determine its local pH, using a pH calibration for each compound. We found that internalized exendin-phe1-FITC was situated in the least, and exendin-asp3-FITC in the most, acidic endosomes (Supplementary Fig. 4c-e), suggesting that each ligand may induce differential endosomal GLP-1R sorting. This was further corroborated by ultrastructural analysis of SNAP-GLP-1R subcellular distribution by electron microscopy, which revealed greater receptor localization to tubular recycling endosomes and PM following exendin-phe1 vs. exendin-4 treatment, while the latter resulted in increased localization to late endosomes and lysosomes (Fig. 2f, g and Supplementary Fig. 5a).

With these differences in mind, we measured SNAP-GLP-1R degradation in CHO-SNAP-GLP-1R and MIN6B1-SNAP-GLP-1R cells by immunoblotting. After 4 and 16 h exposure to exendin-4, significant degradation was noted in both cell types, as indicated by disappearance of the band corresponding to full-length SNAP-GLP-1R. Whilst no additional degradation was

observed with exendin-asp3, this was noticeably reduced with exendin-phe1 (Fig. 2h, Supplementary Fig. 5c).

Finally, we determined the net surface GLP-1R downregulation with prolonged agonist incubations by confocal microscopy and flow cytometry. In MIN6B1-SNAP-GLP-1R (Fig. 2i–k), wild-type MIN6B1, and INS-1 832/3 cells (Supplementary Fig. 5d-h), overnight treatment with exendin-phe1 resulted in the relative preservation of surface GLP-1R vs. exendin-4, in keeping with their acute trafficking differences. Loss of surface receptors was more pronounced for exendin-asp3 than for exendin-4 in some, but not all assays. In CHO-SNAP-GLP-1R cells, a small but reproducible increase in surface downregulation potency was seen with exendin-asp3, as measured by cell surface ELISA (Supplementary Fig. 5b).

These studies indicate how agonist-specific endocytosis and recycling influences the surface and total cellular GLP-1R over prolonged exposure times. However, the relatively small differences with exendin-asp3 suggest that additional mechanisms beyond receptor trafficking may be involved in the blunted insulin secretory response with this compound.

**Prolonged cAMP signaling and homologous desensitization.** Intuitively, reduced loss of surface GLP-1R with exendin-phe1 should permit greater access to extracellular ligand during continuous exposure, with fast recycling ensuring that the receptors are maintained in a sensitized state. These conditions might facilitate continual re-stimulation of GLP-1R to maintain the ongoing insulin release, as suggested by the trajectory of insulin accumulation in Fig. 1h. Accordingly, in INS-1 832/3 and MIN6B1 cells,

accumulation of cAMP (with a low concentration of 3-isobutyl-1-methylxanthine (IBMX) to prevent cAMP degradation) was greater with exendin-phe1 than with exendin-4 and exendin-asp3 (Fig. 3a, b). In a further set of experiments, IBMX (at a higher dose) was added only for the final 10 min out of the 16 h incubation for a point estimate of cAMP synthesis rate; again, cAMP generation was greatest with exendin-phe1, and least with exendin-asp3 (Fig. 3c). In MIN6B1 cells, cAMP production after extended treatment with exendin-4 and exendin-asp3 was marginally reduced compared to the vehicle (Fig. 3d). This might reflect the extensive GLP-1R degradation noted in Fig. 2h, leading to a loss of GLP-1R

constitutive activity or capacity to respond to locally produced GLP-1[31]. Interestingly, when these experiments were widened to include other test agonists, a similar pattern was observed to that of chronic insulin secretion in Fig. 1a; namely, maximal (1 μM agonist) cAMP production increased as maximal internalization fell, but reached a plateau beyond which further decrease in internalization did not yield additional increase in cAMP efficacy. As weak agonists tend to become partial agonists when receptor density is limited[32], this may indicate failure of the slowly internalizing compounds to maximally exploit the residual surface receptors; accordingly, when a supramaximal dose (1 μM) of exendin-4 (to fully activate

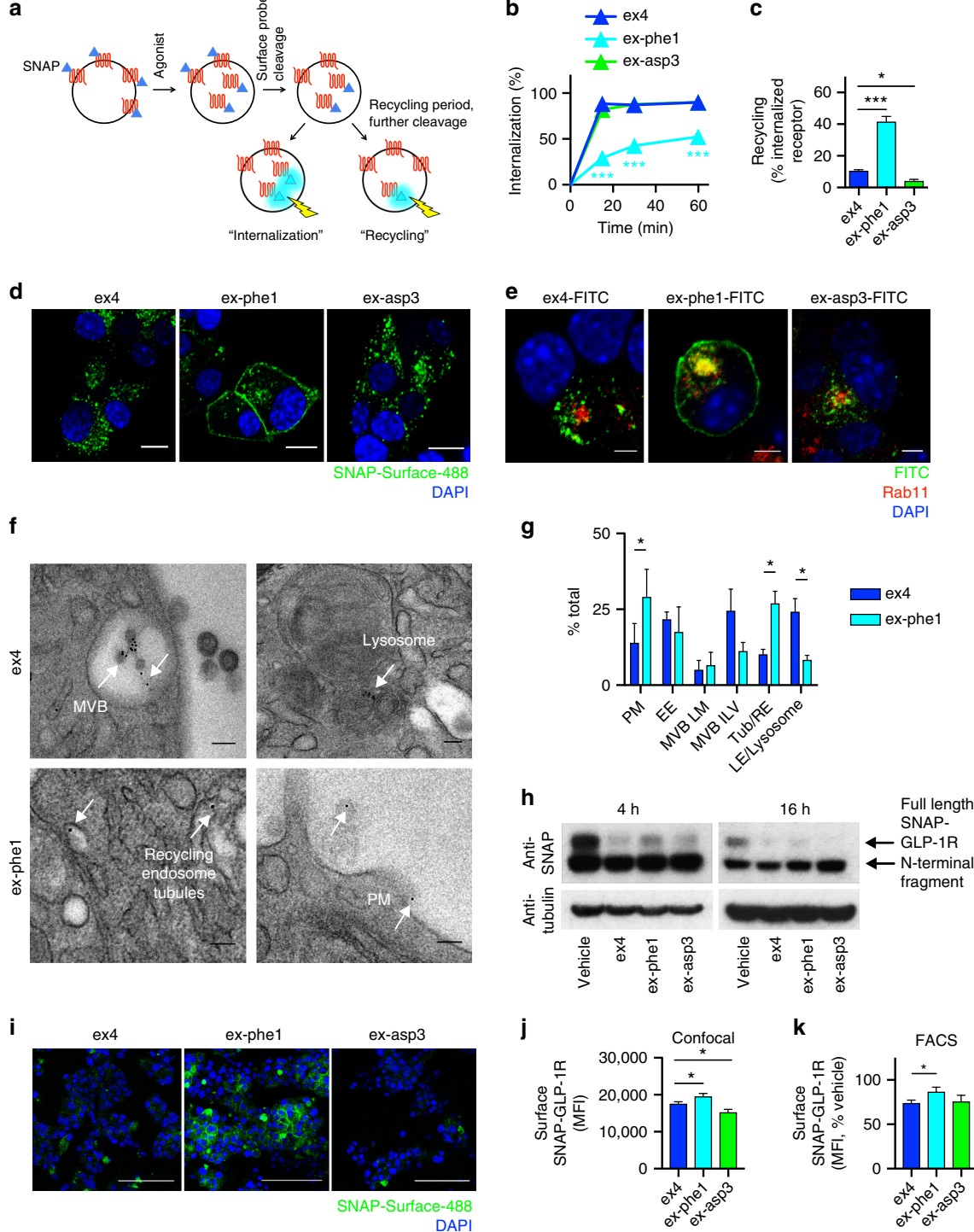

remaining surface receptors) was added in addition to IBMX after prior incubation with each test agonist, a linear relationship between internalization and cAMP response was restored (Fig. 3e). We found no evidence of downregulation in the overall cellular capacity to generate cAMP, as responses to the direct adenylate cyclase activator forskolin were unchanged after prolonged agonist stimulation (Fig. 3f, g). We also measured the response to rechallenge with GLP-1 after agonist exposure, washout, and resensitization. Pretreatment with all agonists resulted in a blunted cAMP response to GLP-1 in INS-1 832/3 cells, this being least marked with exendin-phe1 (Fig. 3h). Homologous desensitization was also reduced with exendin-phe1 in CHO-GLP-1R cells, as measured by intracellular $Ca^{2+}$ release (Fig. 3i).

Thus, we found that agonists that preserve higher levels of surface and total receptor, either by reduced internalization, increased recycling, or both, allow responsiveness retention during continuous or interrupted stimulation. This effect appears, however, to be limited by concomitant reductions in the inherent efficacy of ligand-bound receptors such that the peak efficacy does not continue to increase throughout the full range of internalization.

**Role of GLP-1R biased signaling**. We next investigated further the agonist-specific factors that might explain the marked differences in trafficking and secretion highlighted above. GPCR endocytosis is in many cases dependent on the recruitment of β-arrestins, which interact with clathrin adaptors to promote internalization via clathrin-coated pits[33]. Of note, the role of β-arrestins in GLP-1R trafficking is unclear, as loss of β-arrestin-1 does not affect GLP-1R internalization[14,34]; on the other hand, enhancing β-arrestin-2 action by the overexpression of G protein receptor kinase 5 (GRK5) increases GLP-1R endocytosis[35]. β-arrestins also mediate noncanonical signaling via ERK1/2 and other kinases, which, for the GLP-1R, is linked to insulin secretion and inhibition of beta cell apoptosis[14,36]. In this context, the concept of biased agonism[37] has emerged, in which ligands can selectively engage in different intracellular pathways, potentially allowing specific cellular responses. We therefore measured G protein-dependent cAMP generation and β-arrestin-1 and -2 recruitment with exendin-4, exendin-phe1, and exendin-asp3 at two time-points to avoid kinetic artefacts[38], and calculated the pathway bias by fitting the response data to a modified form of the operational model of agonism[39] (Fig. 4a–d). We found that exendin-phe1 favored G protein signaling, whilst exendin-asp3 favored β-arrestin recruitment. To exclude any artifacts from an irreversible interaction between β-arrestin and receptor in the

PathHunter assay, we visualized GFP-tagged β-arrestin-2 in MIN6B1-SNAP-GLP-1R cells, showing that exendin-4 and exendin-asp3, but not exendin-phe1, induce robust translocation to SNAP-GLP-1R-positive PM and endosomes (Fig. 4e). We repeated bias analysis with the full panel of agonists for β-arrestin-2 (Supplementary Fig. 1b and 6a-c). As for internalization, biased signaling was highly predictive of maximal agonist-induced insulin release, with compounds biased away from β-arrestin-2 recruitment the most effective (Fig. 4f). β-arrestin-1 maximal responses closely mirrored those of β-arrestin-2 (Supplementary Fig. 6d). Of note, exendin-phe1 acted as a competitive antagonist against the β-arrestin response to GLP-1 itself, suggesting that it might reduce the in vivo β-arrestin recruitment by endogenous GLP-1 (Supplementary Fig. 6e, f).

Combined with our trafficking and insulin release results, these data show a canonical role for β-arrestin recruitment in promoting GLP-1R endocytosis and desensitization. However, GLP-1R internalization was previously found to be independent of β-arrestin-1[14,34], and a positive role for β-arrestin-1 in stimulating insulin release has been reported[14]. We explored this further by depleting cells of both β-arrestin isoforms to better represent the loss of recruitment of both β-arrestins seen with exendin-phe1. We found that prolonged exendin-4-induced insulin secretion was increased in INS-1 832/3 and MIN6B1 cells after treatment with small interfering RNA (siRNA) to silence both β-arrestins, and also in human EndoC-βH1[40] beta cells lentivirally transduced with β-arrestin-1 and β-arrestin-2 small hairpin RNA (shRNA) (Fig. 4g–i, Supplementary Fig. 7a-e). While GLP-1R internalization was reduced after dual arrestin siRNA in MIN6B1-SNAP-GLP-1R cells (Supplementary Fig. 7f,g), this effect was more modest than agonist-related differences. To investigate whether this partial effect was due to an incomplete knockdown, we performed experiments in HEK293 cells with both β-arrestin isoforms deleted by CRISPR-Cas9[41]. Interestingly, we found that, whilst exendin-4-induced SNAP-GLP-1R internalization was delayed in β-arrestin-less vs. wild-type HEK293 cells, extensive endocytosis was still achieved with longer incubations (Supplementary Fig. 7h), with no effect on GLP-1R recycling. Nevertheless, cAMP signaling in response to exendin-4 was enhanced in β-arrestin-less cells, as evidenced by the increased potency relative to wild-type, as well as a tendency to increased efficacy over time (Supplementary Fig. 7i, j).

These observations suggest that during pharmacological GLP-1R agonism, the net effect of β-arrestin recruitment is to reduce prolonged insulin release. However, this may not be exclusively through modulation of receptor trafficking.

**Fig. 2** GLP-1R agonist trafficking in MIN6B1-SNAP-GLP-1R cells. **a** Schematic for GLP-1R internalization and recycling measurements by FACS after labeling with cleavable SNAP-Surface probe. **b** Agonist-induced GLP-1R internalization in MIN6B1-SNAP-GLP-1R cells, $n = 3$, two-way randomized block ANOVA with Dunnett's test vs. exendin-4. **c** GLP-1R recycling in MIN6B1-SNAP-GLP-1R cells 30 min after an initial 15 min agonist pulse to induce internalization; recycling measured in the presence of exendin(9-39) to block further endocytosis, $n = 5$ (exendin-4) or 3 (exendin-phe1 and -asp3), one-way ANOVA with Dunnett's test vs. exendin-4. **d** Confocal images indicating GLP-1R internalization in MIN6B1-SNAP-GLP-1R cells, 30 min agonist incubation after SNAP-Surface-488 labeling, representative image from $n = 3$ experiments; scale bars, 8 μm. **e** Immunofluorescence showing increased co-localization of exendin-phe1-FITC vs. exendin-4-FITC and exendin-asp3-FITC with Rab11-positive recycling endosomes after 60 min agonist exposure, representative image from $n = 2$ experiments; scale bars, 4 μm. Individual red and green channels shown in Supplementary Fig. 12. **f** Representative electron micrographs showing subcellular localization of SNAP-GLP-1R (labeled with cleavable SNAP-Surface-biotin plus streptavidin-10 nm gold, arrows), 60 min agonist exposure; scale bars, 0.1 μm; larger area micrographs shown in Supplementary Fig. 5a. **g** Gold-particle quantification from **f**; $n = 3$ experiments, paired $t$-tests. PM plasma membrane, EE early endosome, MVB LM multivesicular body limiting membrane, MVB ILV multivesicular body intraluminal vesicle, Tub/RE tubular/recycling endosome, LE late endosome. **h** Immunoblots showing SNAP-GLP-1R (~73 kDa) levels in MIN6B1-SNAP-GLP-1R cells after 4 and 16 h agonist exposure, representative of $n = 3$ experiments. A smaller band possibly corresponding to deletion of GLP-1R C-terminal domain is detected under all conditions analyzed. **i** Confocal images demonstrating agonist-induced surface GLP-1R downregulation in MIN6B1-SNAP-GLP-1R cells; surface receptor labeled after 16 h agonist treatment; scale bar, 100 μm. **j** Quantification of experiments from **i**, five images analyzed per condition from $n = 3$ coverslips, mean cellular fluorescence indicated, one-way ANOVA with Dunnett's test vs. exendin-4. **k** As for **j**, but quantified by FACS in separate experiments, results normalized to vehicle control, $n = 4$, one-way randomized block ANOVA with Dunnett's test vs. exendin-4. Agonists applied at 100 nM. *$p < 0.05$, ***$p < 0.001$, by statistical test indicated above. Error bars indicate SEM

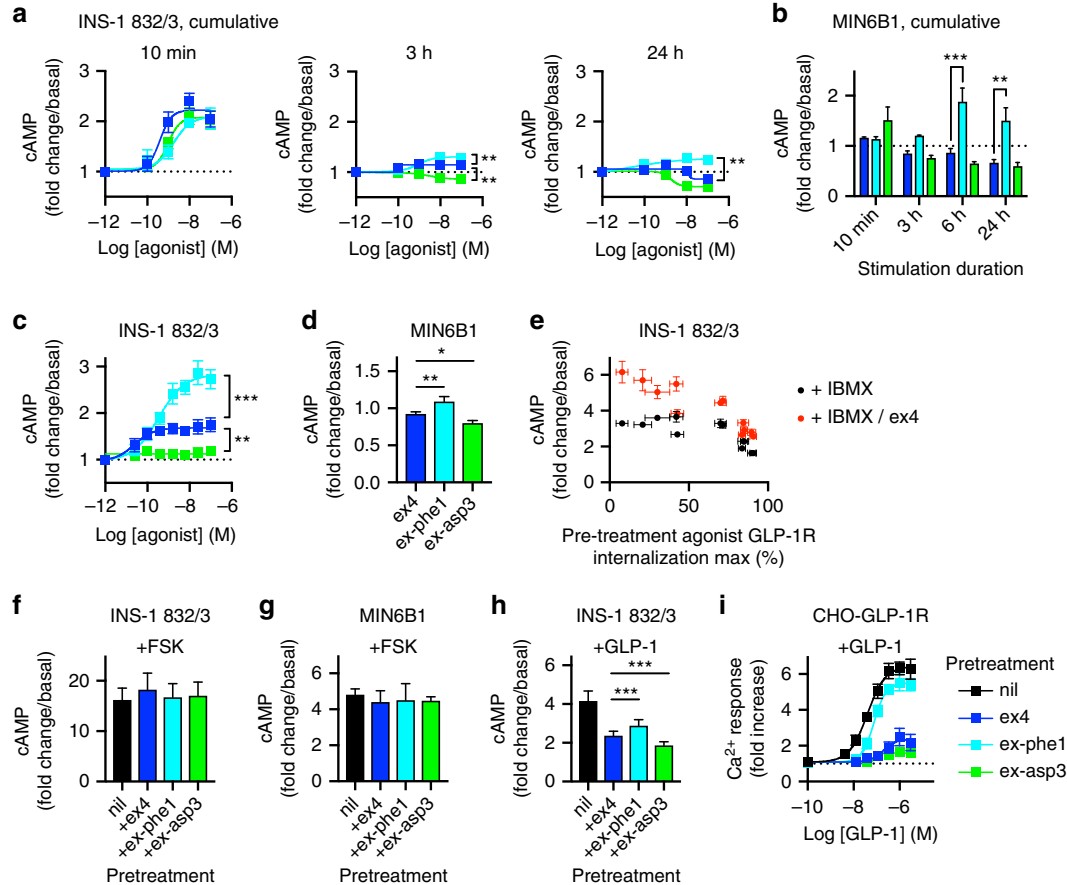

**Fig. 3** Prolonged cAMP generation in beta cells. **a** cAMP measurement in INS-1 832/3 cells in response to continuous agonist exposure for indicated times. Incubations performed in the presence of 25 μM IBMX and the results expressed relative to IBMX-only response for each time-point, $n = 4$, one-way randomized block ANOVA comparing $E_{max}$ with Dunnett's test vs. exendin-4. **b** As in **a**, but with 100 nM agonist in MIN6B1 cells, $n = 3$, two-way randomized block ANOVA with Dunnett's test vs. exendin-4. **c** cAMP accumulation dose response in INS-1 832/3 cells at the end of the 16 h agonist incubation; accumulation induced with 10 min addition of 500 μM IBMX, $n = 6$, one-way randomized block ANOVA comparing $E_{max}$ with Dunnett's test vs. exendin-4. **d** As in **c**, but 100 nM agonist in MIN6B1 cells, $n = 5$, one-way randomized block ANOVA with Dunnett's test vs. exendin-4. **e** cAMP responses in INS-1 832/3 cells induced by the addition of 500 μM IBMX or 500 μM IBMX + 1 μM exendin-4 for the final 10 min after 16 h exposure to 1 μM agonist, expressed relative to response without agonist pretreatment, $n = 4$. **f** Response to 10 μM forskolin (FSK) in INS-1 832/3 cells pretreated with indicated agonist for 16 h, 10 min stimulation plus 500 μM IBMX, $n = 5$. **g** As for **f**, but with MIN6B1 cells, $n = 5$. **h** Homologous desensitization in INS-1 832/3 cells exposed to the indicated agonist for 24 h, washout, 1 h resensitization, and rechallenge ± GLP-1 100 nM, $n = 5$, one-way randomized block ANOVA with Dunnett's test vs. exendin-4. **i** Cytosolic Ca$^{2+}$ response to the indicated doses of GLP-1 in PathHunter CHO-GLP-1R cells exposed to 1 μM agonist for 90 min before a 30 min resensitization period, expressed as peak fold change from baseline reading, $n = 5$. *$p < 0.05$, **$p < 0.01$, ***$p < 0.001$ by statistical test defined in the text. Error bars indicate SEM

**Role of GLP-1R binding kinetics**. The length of time that an agonist remains bound to its receptor, or residence time, is an important factor influencing the duration of drug action and a suggested driver of sustained signaling from internalized receptors[7]. Residence time is defined by the rate of dissociation from the receptor ($1/k_{off}$), so we used time-resolved Forster resonance energy transfer (TR-FRET) to measure agonist-binding kinetics in CHO-SNAP-GLP-1R cells. Here, FITC-conjugated agonists act as FRET acceptors when bound to GLP-1Rs labeled with lanthanide (terbium) SNAP-tag probes[42]. We monitored the real-time cell surface dissociation of exendin-4-FITC, exendin-phe1-FITC, and exendin-asp3-FITC in the presence of exendin(9-39), having first inhibited endocytosis using a cocktail of metabolic inhibitors[9] (Fig. 5a, Supplementary Fig. 8a, b). Exendin-phe1-FITC dissociated the fastest (short residence time), and exendin-asp3-FITC the slowest (long residence time). To exclude artifactual alterations by the FITC group, we performed kinetic-binding experiments with unlabeled agonists, in competition with

exendin-4-FITC, and defined both association rate constants and residence times[43] with consistent effects (Fig. 5b–d).

We also analyzed whether the agonists would differ in their propensity to remain bound to GLP-1R in endosomes. Exendin-4-FITC and exendin-phe1-FITC co-localized with SNAP-GLP-1R (Fig. 5e), but to determine the persistence of agonist–receptor complexes, we adapted our TR-FRET binding assay to include reversible labeling using cleavable SNAP-biotin complexed to streptavidin-terbium to label the surface GLP-1Rs and, after internalization with different FITC-agonists, stripping residual surface receptors of label to ensure that FRET was derived only from internalized receptors (see Fig. 5f for explanation). The ratiometric FRET signal takes account of differences in the number of internalized receptors, and is indicative of avidity of binding. We observed that exendin-asp3 remained bound with highest avidity and exendin-phe1, the least (Fig. 5g). As agonist dissociation within endosomes is a determinant of post-endocytic targeting to recycling or degradative pathways[44], these

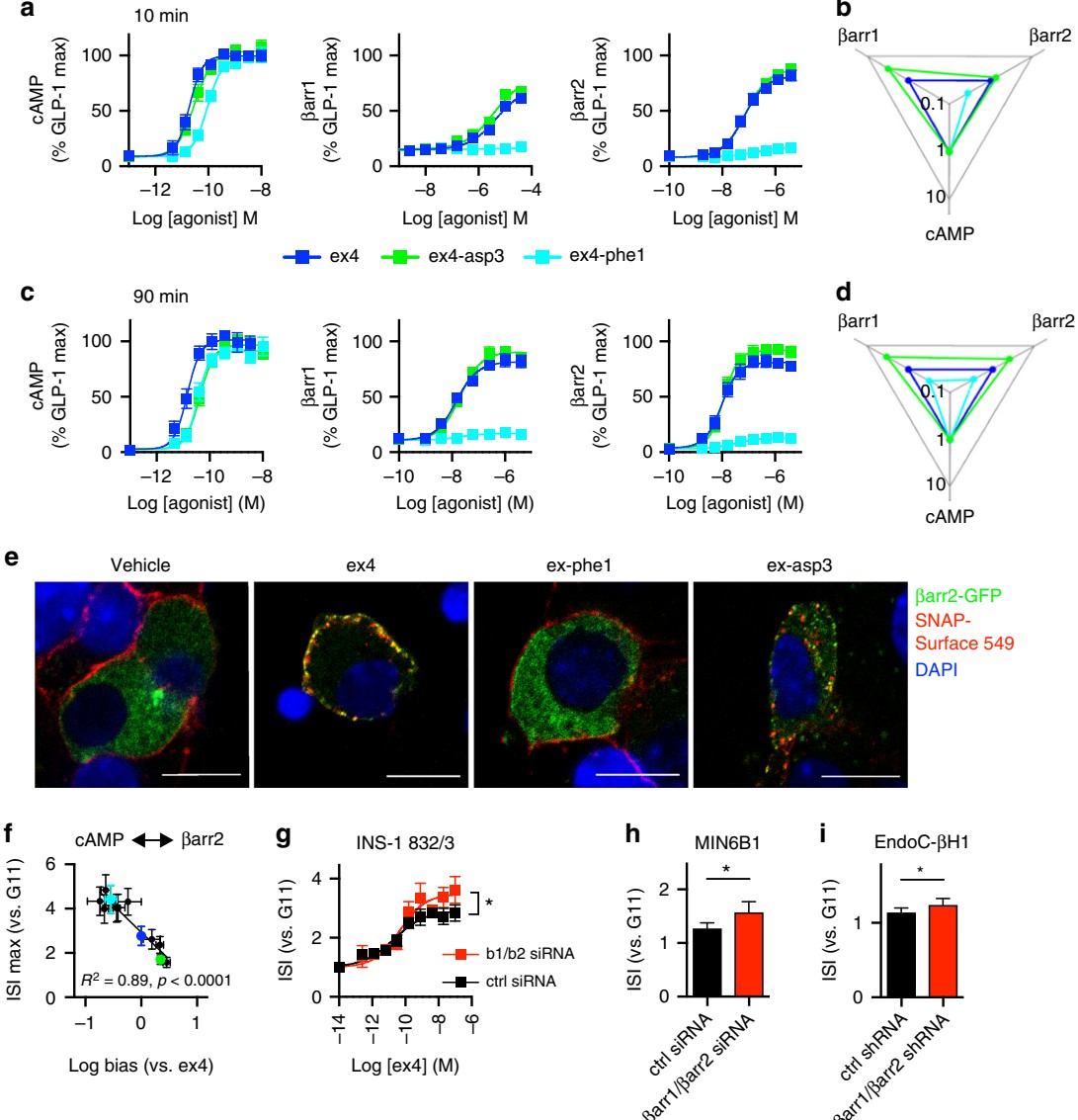

**Fig. 4** β-arrestin-biased signaling reduces insulin secretion. **a** Agonist-induced cAMP, β-arrestin-1 (βarr1), and β-arrestin-2 (βarr2) responses in PathHunter CHO-GLP-1R cells, 10 min incubation, $n = 4$–6; the four-parameter logistic fit of averaged data shown. **b** Web of bias, depicting relative pathway preference for each agonist; data represent the inverse logarithm of normalized log ($\tau/K_A$) values derived from **a** normalized to a reference agonist (exendin-4) and a reference pathway (cAMP); for further details, see Methods. Note that, β-arrestin-1 log ($\tau/K_A$) values for exendin-phe1 could not be calculated due to absence of detectable response. **c**, **d** as for **a**, **b** but for 90 min incubation. **e** Confocal images of MIN6B1-SNAP-GLP-1R cells transiently expressing β-arrestin-2-GFP, labeled with SNAP-Surface-549, and treated with indicated agonist for 5 min before fixation, representative images from $n = 2$ experiments; scale bars, 10 μm. Individual red and green channels shown in Supplementary Fig. 12. **f** Relationship between biased signaling (Supplementary Fig. 6) in PathHunter CHO-GLP-1R cells and maximal prolonged insulin secretion (Supplementary Fig. 1) in INS-1 832/3 cells. Association quantified by linear regression. **g** Effect of dual β-arrestin silencing on prolonged (16 h) exendin-4-induced insulin secretion in INS-1 832/3 cells, $n = 4$, paired $t$-test comparing $E_{max}$. **h** As for **g**, but in MIN6B1 cells, $n = 5$, paired $t$-test. **i** As for **g**, but in EndoC-βH1 cells with stable knockdown of β-arrestin-1 and -2 by lentiviral transduction of shRNAs, $n = 5$, paired $t$-test. Agonists applied at 100 nM, except where indicated. *$p < 0.05$, by statistical test indicated above. Error bars indicate SEM

observations may explain the contrasting recycling patterns of exendin-phe1 vs. exendin-asp3.

To explore through an alternative strategy whether GLP-1R binding kinetics influence PM recycling and insulin secretion, we utilized the GLP-1R allosteric modulator 4-(3-(benzyloxy)phe-nyl)-2-(ethylsulfinyl)-6-(trifluoro-methyl)pyrimidine (BETP), which alters the residence time of orthosteric GLP-1R agonists[45]. We confirmed that exendin-4 residence time is increased by BETP, without changes in the association rate (Fig. 5h, i). In CHO-SNAP-GLP-1R cells, BETP slowed GLP-1R recycling after exendin-4 exposure, with minimal effect on internalization

(Fig. 5j, k). In contrast to its known potentiating effect on insulin secretion with the GLP-1 metabolite and weak agonist GLP-1(9-36)NH$_2$[46], BETP paradoxically reduced the exendin-4-induced insulin secretion with prolonged incubations (Fig. 5l). This did not appear to be through changes to signal bias, as BETP at this dose did not affect cAMP or β-arrestin-2 recruitment responses (Fig. 5m, n). Therefore, via its effects on GLP-1R binding kinetics and trafficking, BETP reduces the agonist responsiveness when exposed for extended periods. This observation holds implica-tions for the design and therapeutic use of positive GLP-1R allosteric modulators.

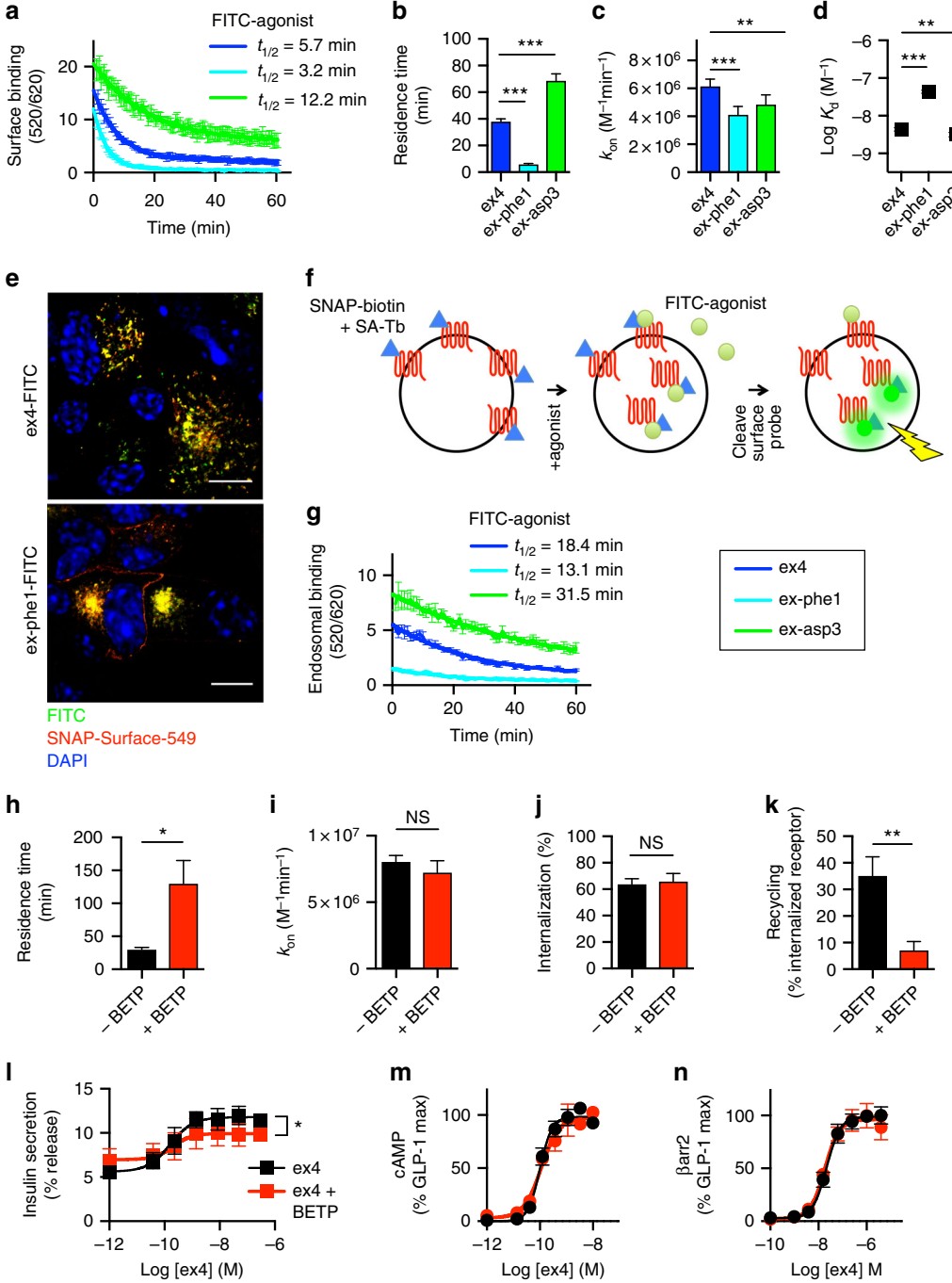

**Fig. 5** Binding kinetics influence GLP-1R recycling. **a** Dissociation curve indicating FRET between FITC-agonist complexed with surface SNAP-GLP-1R, after inhibition of internalization, using $NaN_3$ and 2-deoxyglucose[9] (Supplementary Fig. 9), 30 min agonist exposure, washout, and exendin(9-39) blockade, $n = 4$. Unmodified (non-FITC) agonist **b** residence time ($1/k_{off}$), **c** association rate constant ($k_{on}$), and **d** affinity, measured by TR-FRET in competition with exendin-4-FITC, with internalization inhibitors as above, and calculated using competitive kinetic method[42], $n = 5$, one-way randomized block ANOVA with Dunnett's test vs. exendin-4. **e** Confocal fluorescence indicating co-localization of exendin-4-FITC or exendin-phe1-FITC with SNAP-GLP-1R (labeled with SNAP-Surface-549) after 60 min agonist exposure in MIN6B1-SNAP-GLP-1R cells, representative images from $n = 2$ experiments; scale bars, 8 μm. Individual red and green channels shown in Supplementary Fig. 12. **f** Schematic illustrating endosomal binding protocol. SA-Tb streptavidin-terbium cryptate. **g** Real-time FRET measurement of FITC-agonist complexed with internalized SNAP-GLP-1R after 30 min agonist exposure, washout, exendin (9-39) blockade, and cleavage of SNAP-biotin from surface SNAP-GLP-1R with MesNa, $n = 5$. Exendin-4 **h** residence time and **i** association rate constant ± 10 μM BETP, measured by TR-FRET in competition with exendin-4-FITC, $n = 4$, paired $t$-test. Exendin-4-induced **j** internalization (30 min), and **k** recycling (60 min) ± 3 μM BETP, $n = 4$, paired $t$-test. **l** Prolonged insulin secretion with exendin-4 ± 3 μM BETP in INS-1 832/3 cells, 16 h, $n = 5$, paired $t$-test for $E_{max}$ assessed by four-parameter fit. Exendin-4 **m** cAMP, and **n** β-arrestin-2 responses, in PathHunter CHO-GLP-1R cells ± 3 μM BETP, $n = 3$. Agonists applied at 100 nM, except where indicated, and performed in CHO-SNAP-GLP-1R cells, except where indicated. *$p < 0.05$, ***$p < 0.01$, by statistical test indicated above. Error bars indicate SEM

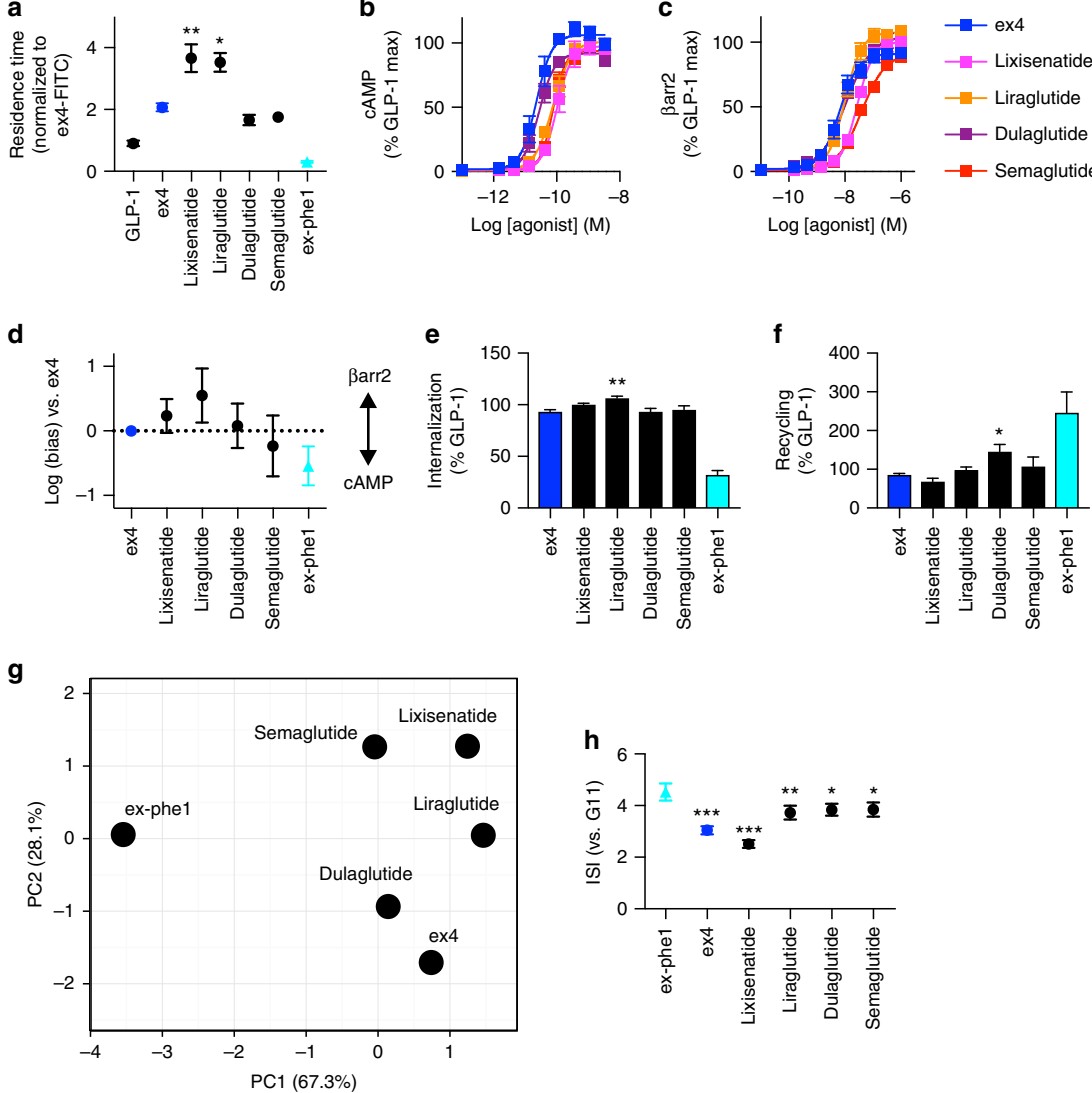

**Fig. 6** Comparison with licensed GLP-1R agonists. **a** Agonist residence time, measured by TR-FRET in competition with exendin-4-FITC in CHO-SNAP-GLP-1R cells, normalized to exendin-4-FITC response per assay, $n = 5$, one-way randomized block ANOVA with Dunnett's test vs. exendin-4 (ex-phe1 not included in statistical analysis). Agonist-induced **b** cAMP, and **c** β-arrestin-2 responses in PathHunter CHO-GLP-1R cells, 90 min, $n = 5$. **d** Bias calculated from data in **b**, **c**, 95% CI shown. Agonist-induced **e** internalization (60 min), and **f** recycling (60 min, measured in presence of 10 μM exendin(9-39) to block rebinding) in CHO-SNAP-GLP-1R cells, normalized to GLP-1 response per assay, $n = 5$, one-way randomized block ANOVA with Dunnett's test vs. exendin-4 (ex-phe1 not included in statistical analysis). Note that exendin-phe1 results, from a different set of experiments, are shown in **a** and **d**–**f**, for purposes of comparison, after normalization to a reference ligand on a per assay basis. **g** Principal component analysis taking into account agonist $k_{off}$, $\Delta \log (\tau / K_A)$ for cAMP and β-arrestin-2, internalization (60 min), and recycling (60 min). **h** Prolonged insulin secretion in INS-1 832/3 cells, 16 h, $n = 6$, one-way randomized block ANOVA with Dunnett's test vs. exendin-phe1. Agonists applied at 100 nM, except where indicated. $*p < 0.05$, $**p < 0.01$, $***p < 0.001$, by statistical test indicated above,. Except where indicated (bias plot), error bars indicate SEM

Overall, these findings show that GLP-1R recycling is influenced by persistence of agonist binding within the endosomal compartment, resulting in enhanced insulin-releasing properties of fast-dissociating agonists which maintain an adequate population of cell surface receptors over prolonged periods.

**Exendin-phe1 outperforms other GLP-1R agonists.** We next compared the pharmacological characteristics and beta cell actions of exendin-phe1 with those of other clinically approved GLP-1R agonists, including Lixisenatide, Liraglutide, Semaglutide, and Dulaglutide. In comparison to the marked differences between N-terminally substituted exendin-4 agonist responses, differential effects on binding kinetics, β-arrestin recruitment, internalization, and recycling were relatively modest within these licensed compounds, albeit statistically significant in some cases (Fig. 6a-f). We

also performed principal component analysis[38] as a further way to compare the overall agonist responses across several readouts. Exendin-phe1 was clearly separated from other GLP-1R agonists (Fig. 6g), as expected from its distinct signaling and trafficking responses. Furthermore, exendin-phe1 was the most efficacious insulin secretagogue when tested in parallel with extended incubations (Fig. 6h). We noted that Semaglutide and Dulaglutide, for which the residence times were shorter than other compounds and also induced modestly increased recycling, were closest to ex-phe1 in this insulin release assay (albeit consistently less efficacious). By contrast, Lixisenatide, which exhibited long residence times and slow recycling, was the least effective.

Therefore, compared to other GLP-1R agonists, exendin-phe1 possesses advantageous pharmacological properties that maximizes insulin secretion.

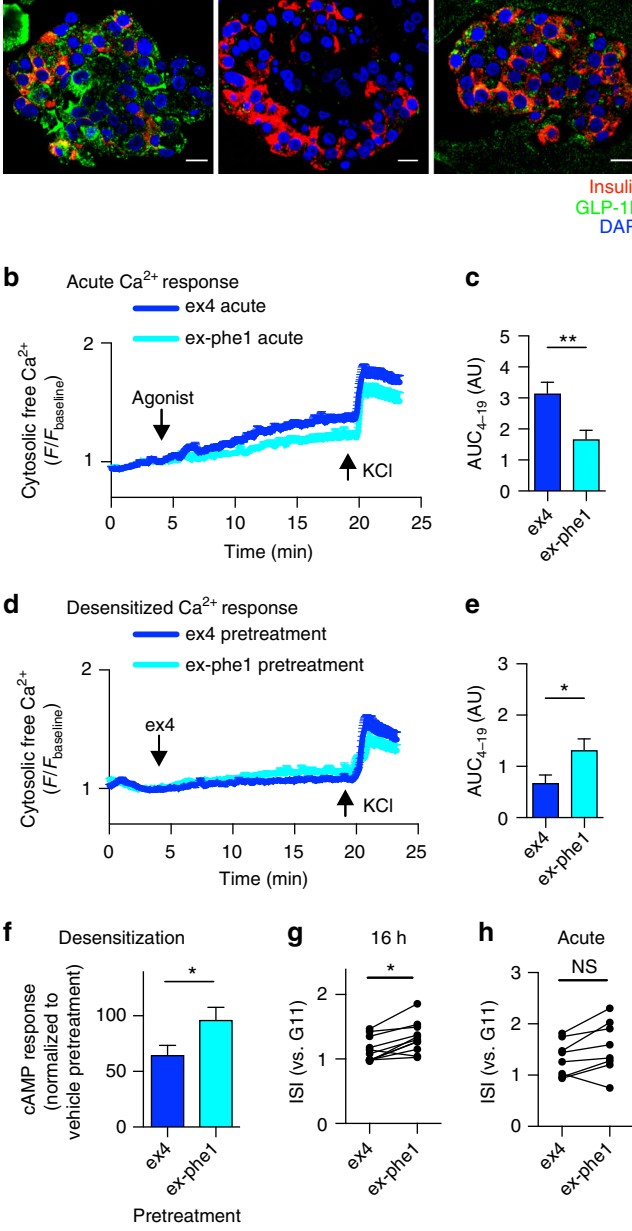

**Fig. 7** Effects of exendin-phe1 in human islets. **a** Residual human islet surface GLP-1Rs labeled with exendin-4-FITC after overnight treatment with G11 ± agonist, representative image from $n = 3$ donors; scale bars, 10 μm. Individual red and green channels shown in Supplementary Fig. 12. $Ca^{2+}$ responses in Fluo-2-loaded human islets to **b** acute stimulation with exendin-4 or exendin-phe1 at G11, or **d** to exendin-4 after overnight pretreatment with exendin-4 or exendin-phe1, $n = 2$ donors, 13–21 islets per condition analyzed. **c, e** Area under the curve (AUC), determined from the point of agonist addition (4 min) to the point of KCl addition (19 min), for traces depicted in **b, d**, respectively, relative to individual islet baselines, unpaired $t$-test. **f** cAMP response to 30 min of 100 nM GLP-1 + 500 μM IBMX in human islets treated overnight ± 100 nM agonist, expressed relative to vehicle pretreatment, $n = 5$ donors, paired $t$-test. **g** Prolonged insulin secretion in human islets, $n = 11$ donors, 16 h, paired $t$-test. **h** As for **g** but for 1 h stimulation, $n = 8$ donors, paired $t$-test. Agonists applied at 100 nM. *$p < 0.05$, **$p < 0.01$, by statistical test indicated above. Error bars indicate SEM

**Exendin-phe1 responses in human islets**. To gain insight into whether the beta cell effects of exendin-phe1 might translate to humans, we compared the responses of exendin-4 and exendin-phe1 in intact human islets. With prolonged agonist treatment, relative preservation of surface GLP-1Rs was detected with exendin-phe1 (Fig. 7a). Accordingly, exendin-phe1 induced less homologous desensitization, as measured by cytosolic $Ca^{2+}$ and cAMP increases (Fig. 7b–f), and was a more powerful secretagogue than exendin-4 during long incubations (Fig. 7g). We note that even during acute incubations, in the majority of matched experiments, exendin-phe1 was more insulinotropic than exendin-4, but this trend did not reach statistical significance (Fig. 7h).

**GLP-1R trafficking is relevant in vivo**. We next investigated the therapeutic potential of exendin-phe1 in mice fed with a high fat and high sucrose (HFHS) diet, an animal model of T2D. We first established that a single injection of exendin-phe1 led to greater lowering of blood glucose over 8 h compared to exendin-4 (Supplementary Fig. 9a, b). Intraperitoneal glucose tolerance tests (IPGTTs) were then performed at 0, 4, and 8 h after 2.4 nmol kg$^{-1}$ agonist administration, revealing a strikingly persistent antihyperglycemic effect of exendin-phe1 vs. exendin-4, associated with greater insulin release (Fig. 8a–c). Importantly, there were no pharmacokinetic differences between agonists that could explain this differential effect (Fig. 8d). As well as stimulating insulin release, GLP-1R agonists promote satiety and reduce food intake, leading to weight loss. Intriguingly, despite clear differences in beta cell effects, appetite suppression was similar for each treatment (Fig. 8e); however, pica (consumption of nonnutritive materials), a rodent correlate of nausea[47], was more frequently observed with exendin-4 administration (Fig. 8f, Supplementary Fig. 9c), as determined by behavioral testing[48]. None of the treatments led to conditioned taste aversion (Supplementary Fig. 9d), as also reported elsewhere in GLP-1R agonists administered peripherally in mice[49]. Key results were repeated at a lower dose of 0.24 nmol kg$^{-1}$, comparable to that of exenatide when used clinically in humans after allometric scaling (Supplementary Fig. 9e-j). By extending the in vivo screening to other panel agonists with IPGTTs performed in lean mice, we unveiled an inverse relationship between glucose lowering and internalized receptor (Supplementary Fig. 9k, Fig. 8g).

Therefore, we found that the enhanced insulinotropic effect of exendin-phe1 in vitro is recapitulated in a mouse model of T2D, suggesting that GLP-1R trafficking plays a role in determining the responses to therapeutic GLP-1R agonists in vivo.

**Chronic administration study**. Finally, to determine whether exendin-phe1 beta cell effects persist with chronic administration, we administered agonists continuously to HFHS-fed mice for 2 weeks via subcutaneous minipumps, using a relatively low dose of agonist (0.24 nmol kg$^{-1}$ day$^{-1}$) to reduce the weight-dependent effects. Fasting glucose reductions were apparent with both treatment groups by the end of the study (Fig. 9a). However, when assessed by IPGTT, chronic exendin-4 treatment at this dose failed to exert significant effects on glucose tolerance, whereas exendin-phe1 remained effective (Fig. 9b, c). Pharmacokinetic differences were again excluded (Fig. 9d). Cumulative food intake and body weight reduction with either of the agonists were no different from vehicle (Fig. 9e, f), consistent with a divergence between beta cell and central effects of exendin-phe1. Note that the minor degree of weight loss in all groups is commonly seen with the implantation of osmotic minipumps due to nonspecific stress[50].

In view of the recent interest in GLP-1R agonism as a nonalcoholic fatty liver disease treatment, we evaluated liver

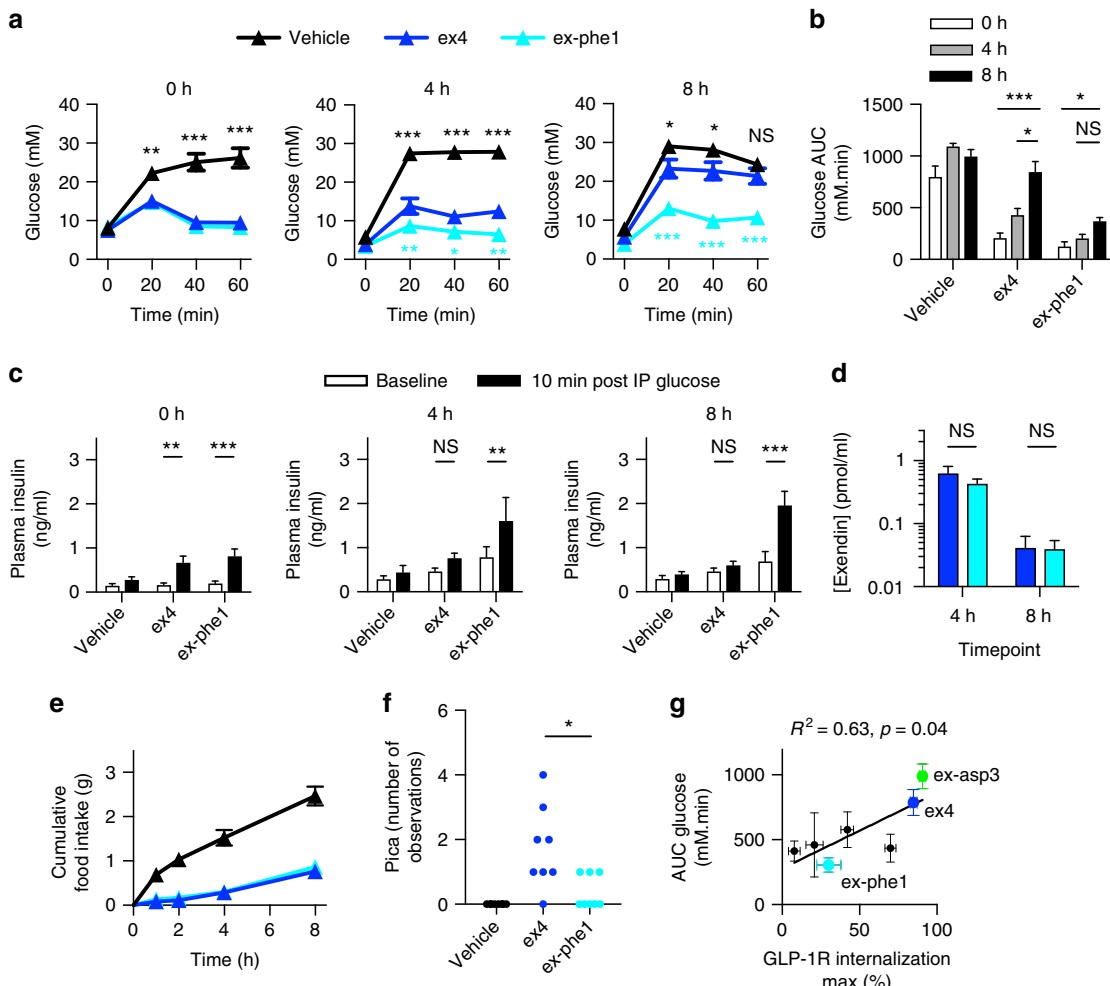

**Fig. 8** Antidiabetic effects of exendin-phe1 in vivo. **a** Blood glucose during IPGTT (2 mg kg$^{-1}$ glucose) performed in HFHS mice at the indicated time-points after intraperitoneal (IP) injection of agonist (2.4 nmol kg$^{-1}$), $n = 10$/group except vehicle group at 4/8 h ($n = 9$), two-way repeat measures ANOVA with Tukey's test, with significance vs. exendin-4 shown. **b** AUC determined from **a**, relative to baseline glucose at $t = 0$, the two-way ANOVA with Dunnett's test vs. 0 h. **c** Plasma insulin before and 10 min after IP administration of glucose (2 g kg$^{-1}$) in HFHS mice, at indicated time-point after IP injection of agonist (2.4 nmol kg$^{-1}$), $n = 10$/group except vehicle 0 h ($n = 8$) and 4 h vehicle/exendin-4 ($n = 9$), two-way repeat measures ANOVA with Sidak's test. **d** Plasma drug level at indicated time-points after IP injection of agonist (24 nmol kg$^{-1}$), $n = 4$ per group, two-way repeat measures ANOVA with Sidak's test. **e** Cumulative food intake in fasted HFHS mice after IP injection of agonist (2.4 nmol kg$^{-1}$), $n = 8$/group. **f** Observed pica behavior in fasted lean mice treated with IP agonist (2.4 nmol kg$^{-1}$), $n = 8$/group, Mann–Whitney test comparing exendin-4 vs. exendin-phe1. **g** Relationship between agonist-induced GLP-1R internalization efficacy (Supplementary Fig. 1) and glucose lowering during IPGTT (Supplementary Fig. 10g), assessed as AUC relative to glucose at $t = 0$, relationship quantified by linear regression. *$p < 0.05$, **$p < 0.01$, ***$p < 0.001$, by statistical test indicated above. Error bars indicate SEM

histology after chronic treatment, showing greater steatosis resolution with exendin-phe1 (Fig. 9g, Supplementary Fig. 9h). This interesting finding, occurring without weight loss, is consistent with preclinical and clinical studies showing weight-independent effects of GLP-1R agonism on liver steatosis[51–53].

Therefore, metabolic improvements from acute and chronic administration of exendin-phe1 exceeded those of exendin-4. The pharmacology of this compound may be an effective approach to improve therapeutic outcomes without loss of tolerability.

## Discussion

In this study, we have identified how single amino acid substitutions to exendin-4 can dramatically enhance insulin secretion via modulation of GLP-1R binding kinetics, biased signaling, and trafficking. Findings from human islets and an in vivo T2D model

suggest that these compounds may have the potential to improve T2D treatment. These effects were achieved under reduced activation of noncanonical signaling pathways including receptor internalization and β-arrestin recruitment, an initially surprising observation as these have previously been linked to increased GLP-1R responses when measured acutely[10,14,36]. However, our study focuses specifically on pharmacological aspects of GLP-1R agonism, with prolonged incubations deliberately chosen to mimic in vivo drug exposure, and knockdown of both β-arrestin isoforms to better approximate the response to exendin-phe1 and related agonists. We propose that the effects of cumulative loss of surface receptor with fast-internalizing, slow-recycling, lysosome-targeting agonists, along with increases in β-arrestin-induced desensitization, become more apparent with time, eventually overriding any positive effects associated with noncanonical signaling. Our findings do not exclude a positive role for endosomal signaling or β-arrestin-1 recruitment in acute responses to

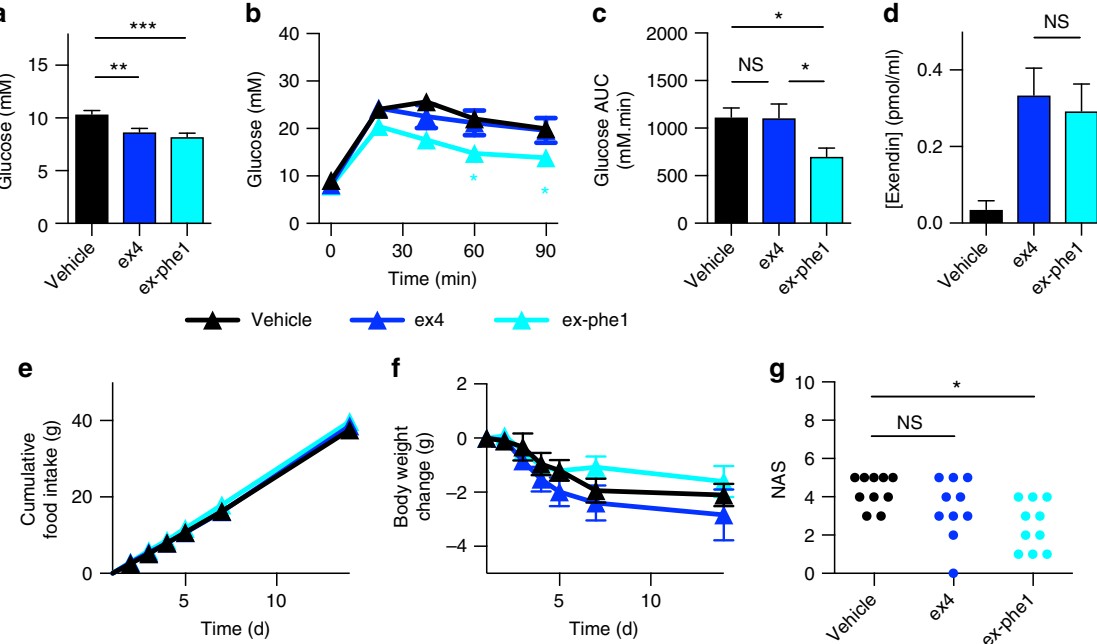

**Fig. 9** Effects of chronic treatment with exendin-phe1. **a** Fasting blood glucose in HFHS mice treated with continuous subcutaneous agonist or vehicle for 16 days, $n = 10$/group, one-way ANOVA with Tukey's test. **b** Blood glucose during IPGTT (2 g kg$^{-1}$) performed after 14 days continuous agonist treatment, $n = 10$/group, two-way repeat measures ANOVA with Tukey's test, with significance for exendin-phe1 vs. exendin-4 indicated. **c** AUC calculated from **b**, relative to baseline glucose at $t = 0$, one-way ANOVA with Tukey's test. **d** Plasma drug level after 16 days treatment with indicated agonist (2.4 nmol kg$^{-1}$ day$^{-1}$) or vehicle in lean mice, $n = 5$/group, unpaired $t$-test. **e** Cumulative food intake, and **f** body weight change with continuous administration of agonist, $n = 10$/group. **g** Histologically determined steatohepatitis, quantified as nonalcoholic activity score (NAS), after 16 days agonist administration, $n = 10$/group, Kruskal–Wallis with Dunn's test. Except where indicated (pharmacokinetic study), agonist administered at 0.24 nmol kg$^{-1}$ day$^{-1}$. *$p < 0.05$, **$p < 0.01$, **$p < 0.001$, by statistical test indicated above. Error bars indicate SEM

physiological GLP-1, but highlight how the kinetic context must be taken into account when exploiting noncanonical pathways during drug development. The relevance of this observation extends to the wider family of GPCRs, with individual effects almost certainly receptor-specific, as both β-arrestin-biased[54] and G protein-biased[55] ligands have proved effective for individual receptors in preclinical studies.

Our study reveals how agonist-related differences in GLP-1R surface loss are predictive of prolonged signaling and secretory responses. It was apparent, however, that agonists diverged both in internalization and recycling capacity, both of which contribute to the net changes to GLP-1R PM residence over time. We were therefore not able to differentiate the relative contribution of each to the overall insulinotropic response. Receptor endocytosis, an intermediate step toward downregulation, is also necessary for resensitization via dephosphorylation and recycling[11], thus, totally inhibiting endocytosis will likely result in reduced signaling as surface receptors become desensitized via β-arrestin recruitment. Indeed, previous reports indicate that GLP-1R signaling is blunted when dynamin-dependent endocytosis is inhibited chemically[10] or genetically[56]. Adapting these studies to investigate how endocytosis inhibition impacts prolonged insulin secretion is challenging due to the nonspecific effects of chemical approaches[57], as well as the role of dynamin in insulin granule exocytosis[58]. Also, rapid desensitization of surface receptors that are prevented from entering the endocytic pathway may be less relevant with agonists such as exendin-phe1, for which β-arrestin recruitment is markedly reduced. We provide evidence here for the importance of recycling via our studies with BETP, which reduced recycling without affecting acute signaling or endocytosis, and accordingly reduced exendin-4-induced insulin secretion. In the future, further insights into this question may be

gained by identification of additional proteins with specific roles in GLP-1R sorting along the endocytic pathway as targets for genetic manipulation. A caveat when interpreting our results is that in many cases we treated cells with relatively high doses of agonist in order to establish differences in maximal response. Nevertheless, in vitro exendin-phe1-induced insulin release exceeded that of exendin-4 in the low nanomolar range (Fig. 1g), suggesting that the mechanisms we unveiled are likely active at doses corresponding to those in vivo[59], and indeed were supported by our own in vivo results. Overall, the net effect of agonist trafficking differences on surface and total receptor downregulation, however achieved, appears highly relevant to patterns of insulin release.

We found here that agonist trafficking properties, as well as insulin secretion, mirrored β-arrestin recruitment propensity, contrasting with some[14,33] (but not all[34]) previous work suggesting that β-arrestins do not impact GLP-1R trafficking[14,34,35], as well as reports of coupling of GLP-1R to insulin secretion via β-arrestin-1[14] (although not β-arrestin-2[60]). The potential for redundancy between β-arrestin isoforms prompted us to investigate the effect of depleting both β-arrestins together. Suggesting that variable β-arrestin recruitment of our agonists may indeed influence their capacity for pharmacological insulin secretion, insulin release from dual β-arrestin knockdown was consistently increased. This was corroborated by a marked effect on cAMP signaling in β-arrestin-less HEK293 cells, which progressively diverged with longer exendin-4 treatment, reminiscent of the effect of exendin-phe1 on beta cell cAMP production. Interestingly however, a more modest effect was seen on GLP-1R endocytosis when β-arrestins were ablated, suggesting that the role of desensitization by β-arrestins supersedes their role in GLP-1R trafficking. This is relevant to our studies with exendin-asp3,

for which differences in loss of surface GLP-1R vs. exendin-4 were small; bias towards β-arrestin-induced desensitization is an additional factor that could explain the reduced insulinotropic efficacy of this compound.

Recently, a biased GLP-1R agonist with reduced β-arrestin-1 recruitment was described as poorly insulinotropic in vivo[61], contrasting with our observations for exendin-phe1 and several other analogs. The reasons for this disparity are not clear, but methodological differences are likely as we specifically sought evidence of beta cell desensitization by performing delayed IPGTTs. Furthermore, while both our study and that of Zhang et al.[58] identified bias between β-arrestin and G protein signaling, there are undoubtedly further intracellular signaling pathways linking GLP-1R to insulin release which may differ but were not measured in either study. Untargeted or semi-targeted approaches, such as phospho-proteomic or kinomic analyses may be required to highlight agonist-related differences in global signaling networks, as for other GPCRs, such as the angiotensin II type 1 receptor[62].

We did not elucidate the specific GLP-1R interactions made by N-terminally modified exendin-4 analogs responsible for their marked differences in signaling and trafficking. A large mutagenesis study identified key residues in the GLP-1R surface region which either interact with the ligand or assist with propagation of occupancy to the cytosol[20]. The end of the ligand N-terminus, most relevant to our biased analogs, was less well defined in the homology model derived from this and other data. Further reciprocal mutagenesis studies will be required to identify key interactions responsible for the contrasting effects of exendin-phe1, exendin-asp3 and other analogs, facilitating future design of agonists with even greater selectivity. Major advances in the understanding of the structure of class B GPCRs have recently been made, including for the GLP-1R[63,64]. It is hoped these may eventually allow identification of receptor conformations associated with selective engagement of particular intracellular effectors.

Despite efficacious beta cell effects of exendin-phe1 in vivo vs. exendin-4, both peptides performed similarly for appetite suppression, indicating that while exendin-phe1 matches the weight-lowering properties of exendin-4, it exhibits additional selective promotion of beta cell-mediated glycemic benefits. The mechanisms responsible for this dichotomy remain to be established. One possibility is that GLP-1R endocytosis is required for agonist uptake by the brain via hypothalamic tanycytes, as described for leptin[65]. As GLP-1R agonists mediate their appetite-reducing effects within the central nervous system, the trafficking phenotype of exendin-phe1 could result in reduced brain penetration, mitigating against presumed advantageous effects on neuronal GLP-1R desensitization. Notably, mice lacking GLP-1R exhibit reduced brain uptake of fluorescently labeled Liraglutide[66].

The potential clinical relevance of this finding relates to the dose-limiting effect of nausea with existing GLP-1R agonists. A total of 30–50% of patients taking this class of drug at clinically licensed doses experience nausea[15]. Careful analyses of trial withdrawal statistics suggest that nausea may in fact be underrecorded, and is strongly associated with treatment discontinuation[67]. Moreover, several clinical studies show that glycemia improves when doses are increased but with unacceptable rates of gastrointestinal adverse events[16–18], suggesting that maximum benefits from GLP-1R agonism are in fact not realized with current treatments. In contrast, the powerful beta cell action of exendin-phe1 provides a potential means to avoid tolerability issues associated with high doses of other agents, and might therefore be a novel therapeutic option for T2D.

## Methods

**Peptides**. Exendin-4, exendin(9–39), and GLP-1 (7–36)NH$_2$ were from Bachem; Liraglutide, Lixisenatide, and Dulaglutide from Imperial College Healthcare NHS Trust pharmacy; and custom peptides from Insight Biotechnology.

**Cell culture and generation of stable cell lines**. PathHunter CHO-GLP-1R β-arrestin-1/-2 reporter cell lines (DiscoverX) were maintained in the manufacturer's Culture Medium. INS-1 832/3 were maintained in RPMI-1640 at 11 mM D-glucose, supplemented with 10% fetal bovine serum (FBS), 10 mM HEPES, 2 mM L-glutamine, 1 mM pyruvate, 50 μM β-mercaptoethanol, and 1% penicillin/streptomycin[21]. MIN6B1 cells (a kind gift from Prof. Philippe Halban, University of Geneva, Switzerland) were maintained in DMEM at 25 mM D-glucose supplemented with 15% FBS, 50 μM β-mercaptoethanol, and 1% penicillin/streptomycin[22]. Stable CHO-SNAP-GLP-1R and MIN6B1-SNAP-GLP-1R cells were generated by transfecting pSNAP-GLP-1R (Cisbio) into wild-type CHO-K1 (ECACC) or MIN6B1, G418 (1 mg ml$^{-1}$) selection, and single-cell sorting by fluorescence-activated cell sorting (FACS) following SNAP-Surface-488 (New England Biolabs) labeling. Wild-type (ECACC) and β-arrestin-less HEK293 cells[41] stably expressing SNAP-GLP-1R were generated by G418 selection and maintained in DMEM at 25 mM D-glucose supplemented with 10% FBS and 1% penicillin/streptomycin. Stable EndoC-βH1 shRNA cell sublines were generated by infection of the parental EndoC-βH1 line[40] with lentiviral particles expressing shRNA duplexes previously cloned into pLKO.3G (Addgene #14748). See Supplementary Table 1 for shRNA target sequences. Lentiviruses were generated by co-transfection of each shRNA construct with envelope plus packaging vectors (pMD2.G, Addgene #12260 and psPAX2, Addgene #12259) and viral supernatants purified by ultra-centrifugation onto 20% sucrose gradients. EndoC-βH1 sublines were maintained in DMEM at 5 mM D-glucose supplemented with 2% BSA, 50 μM β-mercaptoethanol, 10 mM nicotinamide, 5.5 μg ml$^{-1}$ transferrin, 6.7 ng ml$^{-1}$ sodium selenite, and 1% penicillin/streptomycin[40]. Mycoplasma testing was performed yearly.

**Human islet isolation and culture**. Human islet studies were approved by the National Research Ethics Committee London (REC 07/H0711/114). Islets were obtained from normoglycemic donors, according to the local ethics rules (including next-of-kin consent) and isolation techniques, and cultured in RPMI supplemented with 5.5 mM D-glucose, 10% FCS, 100 U penicillin, 100 μg streptomycin, and 0.25 μg μl$^{-1}$ fungizone (37 °C, 5% CO$_2$)[28]. Experiments were performed with randomly allocated, size-matched islets. Donor characteristics and isolation center details are indicated in Supplementary Table 2.

**cAMP assays**. Cyclic AMP accumulation was determined by HTRF (cAMP Dynamic 2, Cisbio). CHO-SNAP-GLP-1R were stimulated at 37 °C in serum-free DMEM. For dose responses, a full GLP-1 curve was included to establish the assay $E_{max}$ with curve fitting to a four-parameter fit. INS-1 832/3 and MIN6B1 cells were stimulated in serum-free media plus 3 mM glucose and IBMX as indicated, with results expressed relative to time-point-specific IBMX-only baseline.

**β-arrestin recruitment assay**. PathHunter CHO-GLP-1R β-arrestin-1 and -2 reporter cell lines were used, following the manufacturer's instructions; in short, cells were stimulated in the manufacturer's stimulation buffer for the indicated time period at 37 °C, before addition of the chemiluminescent substrate and luminescence read after a further 60 min incubation at room temperature. A full GLP-1 dose response was included to establish the assay $E_{max}$.

**Quantitation of bias**. Bias between cAMP, β-arrestin-1, and β-arrestin-2 responses was determined using a modified operational model of agonism. Concentration response data was fitted with described equations[39] to derive transduction ratios ($\tau/K_A$) for each agonist and pathway. Log ($\tau/K_A$) values were normalized by subtracting log ($\tau/K_A$) for exendin-4 in each pathway, giving Δlog ($\tau/K_A$). To determine the bias between the two pathways, Δlog ($\tau/K_A$) values were subtracted, yielding ΔΔlog ($\tau/K_A$). Statistical analysis was performed by determining Δlog ($\tau/K_A$) for each agonist and experiment, and propagating error from average Δlog ($\tau/K_A$) from several experiments to calculate 95% confidence intervals (CI); statistical significance was inferred if the 95% CI of ΔΔlog ($\tau/K_A$) estimate did not cross zero.

**Cell surface labeling plate reader trafficking**. GLP-1R internalization and recycling in CHO-SNAP-GLP-1R cells was measured by surface labeling with anti-GLP-1R antibody or the SNAP-tag probe Lumi4-Tb (Cisbio). For antibody labeling, after treatments at 37 °C, cells were placed on ice to arrest further endocytosis before fixation, and the surface GLP-1R was detected by ELISA with monoclonal anti-human GLP-1R antibody[68] (Mab 3F52, Developmental Studies Hybridoma Bank (DSHB), 1/100) plus horseradish peroxidase (HRP)-conjugated rabbit anti-mouse secondary (ab97046, Abcam, 1/5,000). 3,3′,5,5′- tetramethylbenzidine (TMB) substrate was added and the absorbance was read at 450 nm after 1 M HCl addition. For Lumi4-Tb, cells were labeled at 40 nM at 4 °C, followed by time-resolved (TR) fluorescence measurement in a Molecular Devices i3x (excitation 335 nm, emission 620 nm). Residual surface expression was determined from

peptide- vs. control-treated wells. Recycling was measured by comparing the surface receptor immediately after internalization vs. a further period of recycling with agonist washed off and replaced with 10 µM exendin(9-39); and recycling calculated as percentage recovery of surface GLP-1R in the recycling plate vs. surface receptor loss in the internalization plate.

**DERET assay.** GLP-1R internalization in real-time[26] was measured in CHO-SNAP-GLP-1R cells labeled with Lumi4-Tb and incubated with 24 µM fluorescein in HBSS. TR fluorescence was serially read with a Flexstation 3 (Molecular Devices), with 340 nm excitation, 520 nm (cut-off 495 nm) and 620 nm (cut-off 570 nm) emission, 400 µs delay and 1500 µs integration time. To prevent endocytosis, metabolic inhibitors (10 mM NaN₃, 20 mM 2-deoxyglucose)[9] were added 20 min before labeling completion and maintained throughout the experiment when indicated. Surface receptor loss was calculated from 620 nm over 520 nm signals.

**Surface dissociation kinetics.** Binding of FITC-agonist to SNAP-GLP-1R was measured in real-time by TR-FRET[42]. CHO-SNAP-GLP-1R cells were labeled with Lumi4-Tb plus metabolic inhibitors as above. FITC-agonists were added for the last 30 min. Agonist dissociation was monitored immediately after washing in 10 µM exendin(9-39) by TR-FRET at 37 °C, with 50 µs delay and 300 µs integration time, and binding quantified as 520 nm over 620 nm signal after nonspecific binding subtraction with excess (10 µM) unlabeled agonist. To calculate dissociation kinetics, a one-phase exponential decay function was fitted to specific binding data.

**Competitive association kinetics.** TR-FRET was monitored on simultaneous addition of four or more concentrations of unlabeled agonist plus 10 nM exendin-4-FITC to CHO-SNAP-GLP-1R cells labeled with Lumi4-Tb and pre-incubated with metabolic inhibitors. Association and dissociation rate constants were calculated using the kinetics of competitive binding algorithm in GraphPad Prism[43]. Residence time = $1/k_{off}$; $K_d = k_{off}/k_{on}$.

**Endosomal binding.** CHO-SNAP-GLP-1R cells were labeled with 5 µM BG-SS-biotin, a cleavable SNAP-Surface probe containing a disulfide bond between O⁶-benzylguanine and biotin moieties (a gift from New England Biolabs) plus streptavidin-terbium cryptate (Cisbio, 0.24 µg ml⁻¹ active moiety). FITC-agonist was added for 30 min at 37 °C to induce internalization, then removed and washed with cold HBSS to arrest further endocytosis, and cells treated with 500 mM ice-cold sodium 2-sulfanylethanesulfonate (MesNa), a membrane-impermeable reducing agent, in alkaline TNE buffer (100 mM NaCl, 50 mM Tris-HCl, pH 8.6) to cleave BG-SS-biotin specifically from cell surface receptors. Exendin(9-39) was then added to block further surface receptor binding. FRET between FITC-agonist and terbium-labeled internalized SNAP-GLP-1R was measured as above.

**Endosomal pH investigation.** The effect of pH on FITC-agonist fluorescence was determined as fluorescence intensities [485 nm excitation, 520 nm emission (cut-off 495 nm)] of agonist solutions at specific pHs and data fitted using linear regression. CHO-SNAP-GLP-1R cells were pretreated ± 100 nM bafilomycin to generate neutral endosomal pH[30] and exposed to 100 nM FITC-agonists for 30 min at 37 °C to induce internalization. After washing, internalized agonist fluorescence was measured over 60 min. Average pH of endosomal compartments was estimated using the relative signal change ± bafilomycin (bafilomycin assumed to achieve a pH of 7.4) in conjunction with pH calibration for each FITC-agonist.

**Immunofluorescence and confocal microscopy.** Cells were labeled at 37 °C with 1 µM SNAP-Surface 488 in HEPES-bicarbonate buffer (120 mM NaCl, 4.8 mM KCl, 24 mM NaHCO₃, 0.5 mM Na₂HPO₄, 5 mM HEPES, 2.5 mM CaCl₂, and 1.2 mM MgCl₂, saturated with 95% O₂/5% CO₂, pH 7.4) plus 3 mM glucose and 1% BSA, stimulated with agonists at 11 mM glucose for the indicated times, fixed and processed for immunofluorescence with mouse anti-Rab11 (610657, BD Biosciences, 1/20) or anti-rodent GLP-1R monoclonal antibody (Mab 7F38, DSHB, 2 µg µl⁻¹) plus Alexa Fluor 546 secondary (Life Technologies, 1/200), mounted in Prolong Diamond antifade reagent with 4,6-diamidino-2-phenylindole (Life Technologies) and imaged with a Zeiss LSM-780 inverted confocal laser-scanning microscope in a ×63/1.4 numerical aperture oil-immersion objective and analyzed in Image J. Surface GLP-1R downregulation was measured from images taken with fixed microscope settings throughout the experiment using a ×20 objective. Mean intensities were quantified from several images per treatment following thresholding to cell-occupied areas.

**Human islet histology.** Human islets were incubated overnight in medium plus 11 mM glucose ± 100 nM agonist prior to transfer to 4 °C to arrest endocytosis and labeling with 1 µM exendin-FITC, PFA fixation, 70% ethanol dehydration and suspension in 4% agarose small plugs. Once set, plugs were dehydrated through a serial ethanol gradient and HistoChoice (Sigma) before paraffin embedding with the Histoembedder station (Leica) and cutting to 1 µm sections with a Leica Jung RM2035 microtome.

Sections were dewaxed and stained with a rabbit polyclonal anti-FITC (711900, Thermo Fisher Scientific, 1/100) plus secondary Alexa Fluor 488 antibody (Life Technologies, 1/200), and guinea pig anti-human insulin (A0564, Dako, 1/1000) plus secondary Alexa Fluor 546 antibody (Life Technologies, 1/1000) prior to confocal imaging as above.

**FACS trafficking.** MIN6B1-SNAP-GLP-1R or CHO-SNAP-GLP-1R cells were pre-incubated in HEPES-bicarbonate buffer plus 3 mM glucose and 1% BSA before labeling at 37 °C with 1 µM cleavable BG-SS-488 SNAP-Surface probe (a gift from New England Biolabs) and stimulation with 100 nM agonist in HEPES-bicarbonate buffer plus 11 mM glucose and 1% BSA, and trafficking assays performed as follows[27]. Briefly, for receptor internalization, cells were placed at 4 °C to arrest endocytosis following incubation at 37 °C for the indicated times, or directly put at 4 °C for 0 min time-point, and treated with ice-cold alkaline TNE buffer ± 100 mM MesNa to strip surface-exposed label (plus MesNa) or to measure total labeled receptor (minus MesNa). For recycling, cells were incubated with agonists at 37 °C for 15 min, followed by endocytosis arrest at 4 °C, surface label removal as above, 30 min incubation at 37 °C with 10 µM exendin(9-39) to prevent further internalization, and a second round of surface label removal at 4 °C.

For surface downregulation, labeling was performed with SNAP-Surface-488 in MIN6B1-SNAP-GLP-1R cells before fixation, or with anti-rodent GLP-1R (2 µg µl⁻¹) in fixed MIN6B1 and INS-1 832/3 cells plus FITC-conjugated anti-mouse secondary (F0257, Sigma, 1/200).

Cells were resuspended in ice-cold PBS plus 0.1% BSA and processed using a BD LSR II flow cytometer (10,000 cells/sample) for receptor trafficking, or BD LSRFortessa X-20 for surface downregulation. The data was analyzed with FlowJo: median fluorescence emission at 525 nm from living, single cells was measured. For MIN6B1-SNAP-GLP-1R, highly auto-fluorescent cells were excluded by dual 525 and 585 nm fluorescence measurement.

The percentage of internalized receptor was calculated as follows:
Equation (1)

$$\frac{(F_{+Me}(t_x)/F_{-Me}(t_x)) - (F_{+Me}(t_0)/F_{-Me}(t_0))}{1 - (F_{+Me}(t_0)/F_{-Me}(t_0))} \times 100,$$

where $F_{+Me}(t)$ and $F_{-Me}(t)$ are median fluorescence ± MesNa at time $t_x$ (15, 30, or 60 min) or $t_0$ (0 min). The percentage of recycled receptor was calculated by subtracting residual median fluorescence following the recycling protocol from that measured at $t = 15$ min, and normalizing to the percentage of internalized receptor at the same $t = 15$ min time-point.

**Degradation assay.** Cells were treated in serum-free medium plus 11 mM glucose ± agonist (100 nM) before lysis (20 mM Tris base, 150 mM NaCl, 1 mM EDTA, 1 mM EGTA, 0.5% Triton X-100 plus complete mini EDTA-free inhibitor (Roche)), addition of urea sample buffer (100 mM Tris-HCl pH 6.8, 2.5% SDS, 4 M urea, 100 mM DTT, 0.05% bromophenol blue), 10 min incubation at 37 °C and immunoblotting.

**Electron microscopy.** MIN6B1-SNAP-GLP-1R cells cultured on Thermanox coverslips (Agar Scientific) were labeled with 2 µM cleavable SNAP-Surface biotin probe in HEPES-bicarbonate buffer plus 3 mM glucose and 1% BSA, followed by incubation with 5 µg ml⁻¹ NaN₃-free Alexa Fluor 488 Streptavidin, 10 nm colloidal gold conjugate (Molecular Probes) and stimulation with 100 nM agonist in buffer plus 11 mM glucose. Conventional EM was performed as described[69]. Briefly, cells were fixed, processed, mounted on Epon stubs, polymerized at 60 °C, and 70 nm sections cut en face with a diamond knife (DiATOME) in a Leica Ultracut UCT ultramicrotome before examination on an FEI Tecnai G2-Spirit TEM. Images were acquired in a charge-coupled device camera (Eagle), and gold particles quantified in at least five cells per experiment using Image J.

**Insulin secretion assays.** Cells were pre-incubated overnight in 3 mM glucose medium. For INS-1 832/3, agonists were added in HBSS (acute incubations) or complete medium (prolonged incubations), at 11 mM glucose, at the time of seeding into plates. MIN6B1 were treated as INS-1 832/3 but in Krebs-HEPES-bicarbonate (KHB) buffer (140 mM NaCl, 3.6 mM KCl, 1.5 mM CaCl₂, 0.5 mM MgSO₄, 0.5 mM NaH₂PO₄, 2 mM NaHCO₃, 10 mM HEPES, and 1% BSA; saturated with 95% O₂/5% CO₂; pH 7.4). EndoC-βH1 sublines were treated as INS-1 832/3.

For human islets, acute incubations were performed in KHB buffer at the indicated concentration of glucose ± agonist[28]. Overnight incubations were performed in complete RPMI plus 11 mM glucose ± 100 nM agonist. Samples were obtained for secreted and, where indicated, total insulin, and analyzed by HTRF (Cisbio). Percentage of release was calculated, and agonist-stimulated results expressed relative to 11 mM glucose alone as insulin stimulation index.

**Caspase and TUNEL assays.** INS-1 832/3 cells were exposed overnight to thapsigargin (1 µM) in serum-free RPMI plus 11 mM glucose. Apoptosis was

determined using Caspase Glo 3/7 (Promega). Signal was expressed relative to thapsigargin-only wells.

MIN6B1 cells were incubated overnight with serum-free media plus 25 mM glucose and 0.5 mM palmitate/BSA ± 100 nM agonists, fixed, processed with the In Situ Cell Death Detection Kit, TMR red (Roche), and imaged as above. Five images were taken in random areas and a minimum of 500 cells counted per experiment.

**Calcium assay in CHO-GLP-1R cells.** PathHunter GLP-1R cells were loaded with calcium dye (Calcium 6, Molecular Devices) in HBSS plus 20 mM HEPES and 2.5 mM probenecid ± 1 µM agonist for 90 min at 37 °C. Agonist was washed and fresh dye added for 30 min. Fluorescence before and after robotic addition of GLP-1 was serially read using a Flexstation 3 with 485 nm excitation, 525 nm emission (cut-off 515 nm), and expressed relative to average baseline to establish fold change, with GLP-1 concentration response data analyzed by four-parameter fit.

**Human islet calcium imaging.** $Ca^{2+}$ imaging was performed as described[29]. Briefly, islets were incubated for 1 h with Fluo-2 (10 µM) in HEPES-bicarbonate buffer plus 11 mM glucose. For homologous desensitization, islets were pre-incubated overnight with 100 nM agonist. Fluorescent signals were normalized using $F/F_{baseline}$ where $F$ is fluorescence at a given time-point and $F_{baseline}$ is average fluorescence between 2 and 4 min.

**RNA interference.** siRNA transfections were performed with Lipofectamine 2000 for 72 h. Sequences for MIN6B1 were ON-TARGET plus Non-targeting Control Pool and ON-TARGET plus Mouse Arrb1 and Arrb2 SMARTpools (Dharmacon). For INS-1 832/3, Silencer Select siRNA (Thermo Fisher Scientific) targeting rat Arrb1 (ID: 129662) and Arrb2 (ID: 129665), or negative control siRNA, were used. See Supplementary Table 1 for siRNA target sequences.

**Quantitative PCR.** Knockdown efficiency was determined by qRT-PCR using standard methodologies. For INS-1 832/3, Rn01648673_m1 (Arrb1) and Rn01456874_g1 (Arrb2) Taqman probes were used, with 18S as endogenous control. For MIN6B1, SYBR Green primers were designed using PerlPrimer. See Supplementary Table 3 for qRT-PCR primer sequences.

**SDS–PAGE and western blotting.** Samples were fractioned by SDS–PAGE on 8% gels under reducing conditions, immunoblotted onto nitrocellulose membranes (GE Healthcare) and bands detected by enhanced chemiluminescence (GE Healthcare) onto films developed on a Xograph Compact X5 processor.

Antibodies were primaries rabbit anti-β-arrestin-1/2 (D24H9, Cell Signaling, 1/1000), rabbit anti-SNAP tag (New England Biolabs, 1/500), mouse anti-α-tubulin (T5168, Sigma, 1/1000), rabbit anti-β-actin (4970, Cell Signaling, 1/1000), and mouse anti-GAPDH (6C5, Merck, 1/10,000); and IgG-HRP secondaries (Santa Cruz Biotechnology).

**Animal studies.** All animal procedures were approved by the British Home Office under the UK Animal (Scientific Procedures) Act 1986 (Project Licence 70/7596). Male C57BL/6 J mice (8–10 weeks, Charles River) were maintained at 21–23 °C and light-dark cycles (12:12 h schedule, lights on at 07:00). Ad libitum access to water and normal chow (RM1, Special Diet Services) or a HFHS diabetogenic diet (AIN-76A, TestDiet) was provided unless otherwise stated. HFHS animals were initially group housed (5 per cage) for >4 months before transfer to single cages. Treatments were randomly allocated according to body weight. Group sizes of 8–10 were deemed adequate to detect treatment-related differences as per initial dose-finding glycaemia. During experiments, one researcher was aware of treatment allocation but others were blinded. No animals were excluded from the analysis.

**Dose-finding glycaemia study.** HFHS mice were fasted for 2 h before IP injection of 50 µl agonist or vehicle (0.9% NaCl). Blood samples were obtained at indicated time-points, and glucose measured using a Contour glucose meter (Bayer).

**Intraperitoneal glucose tolerance tests.** Mice were fasted overnight (HFHS mice) or the morning of the procedure (lean mice). Agonist was administered by IP injection, and D-glucose (2 g kg$^{-1}$) injected IP immediately, 4 or 8 h afterwards. Tail vein samples were obtained for immediate glucose measurement as above, or into lithium heparin-coated microvette tubes for plasma insulin measurement using a mouse insulin-specific HTRF (Cisbio).

**Pharmacokinetic study.** Plasma agonist concentration after a single IP injection (24 nmol kg$^{-1}$) was measured with an exendin-4 C-terminus-specific ELISA (Phoenix Pharmaceuticals), a peptide region that does not differ between agonists, hence equally detecting exendin-4 and exendin-phe1 (confirmed by analysis of known concentrations of each agonist).

**Acute food intake study.** Mice were fasted overnight and access to their normal diet was returned after IP agonist injection with weight monitoring of food intake.

**Behavioral satiety study.** Lean mice were fasted overnight before IP agonist injection. Observers were blinded to treatment allocation. A 30 min after agonist injection, diet was returned. Behavior of each mouse was observed for 5 s every 3 min for 60 min. Behaviors were classified as feeding, drinking, pica (consumption of nonnutritive material), activity (locomotor, rearing, grooming), or stationary. Number of observations of each behavior per mouse was recorded.

**Conditioned taste aversion study.** Lean mice were trained to consume their daily water requirements over 1 h, then given access to saccharin-sweetened Kool-Aid, followed immediately by IP injection of a potentially aversive stimulus (vehicle, agonist, or LiCl 0.15 M in a volume equivalent to 2% body weight as positive control). After 24 h recovery, mice were given a free choice of water or Kool-Aid. Taste preference was calculated as Kool-Aid/total fluid consumed.

**Chronic administration study.** Subcutaneous osmotic minipumps (ALZET model 2004, Charles River) filled with agonist or vehicle (0.9% NaCl) to ensure delivery of a weight-adjusted dose of 0.24 nmol kg$^{-1}$ day$^{-1}$ were inserted under gas anesthesia. Mice and diet were weighed 1 day post-surgery and body and food weight measured at indicated intervals. IPGTTs were performed on day 14 and mice sacrificed by decapitation during fasting. As the exendin-4 ELISA above was not sensitive enough to detect circulating drug levels at this dose, a separate cohort of lean animals received 2.4 nmol$^{-1}$ kg$^{-1}$ day$^{-1}$ agonist via osmotic minipump and, with samples taken after 16 days and exendin-4 levels analyzed as above.

**Liver histology.** Liver tissue was PFA-fixed and dehydrated in 70% ethanol. Heaematoxylin- and eosin-stained sections were scored by a histopathologist blinded to treatment allocation using the Nonalcoholic Activity Score[70] with fat scored 0–3, ballooning 0–2, and lobular inflammation 0–2.

**Statistical testing.** GraphPad Prism 6.0 was used for all analyses. Curve fitting and bias calculation were performed as described above in the relevant Methods section. For in vitro experiments, intra-experimental replicate mean was treated as a single replicate. ANOVAs or two-tailed $t$-tests were performed throughout, with data visually confirmed as approximately normally distributed. Dunnett's post-hoc tests were performed for specifically comparing agonist responses with exendin-4, and Tukey's for differences between all groups. Other post-hoc tests are indicated in the figure legends.

**Data availability.** Data supporting the findings of this manuscript are available from the corresponding authors upon reasonable request. Non-normalized data sets from selected results are shown in Supplementary Fig. 10. Full scans of blots presented in cropped form in the manuscript are shown in Supplementary Fig. 11. Individual red and green channels of RBG images from the main figures are shown in Supplementary Fig. 12.

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

## Acknowledgements

The Section of Endocrinology and Investigative Medicine is funded by grants from the MRC, BBSRC, NIHR, an Integrative Mammalian Biology (IMB) Capacity Building Award, an FP7-HEALTH-2009-241592 EuroCHIP grant and is supported by the NIHR Biomedical Research Center Funding Scheme; B.J. was funded by an MRC Clinical Research Training Fellowship; B.M.O. was funded by a Sir Henry Dale Fellowship (105545/Z/14/Z) jointly funded by the Wellcome Trust and the Royal Society; R.S. was supported by a Wellcome Trust Clinical Research Training Fellowship; A.I. was funded by the Japan Science and Technology (JST), PRESTO (JPMJPR1331), the Japan Society for the Promotion of Science (JSPS), KAKENHI (17K08264), and the PRIME from the Japan Agency for Medical Research and Development (AMED); B.J. C. was supported by an International Atherosclerosis Society Fellowship; G.A.R. was supported by a Wellcome Trust Senior Investigator Award (WT098424AIA), MRC Program grants (MR/J0003042/1, MR/L020149/1) and Experimental Challenge Grant (DIVA, MR/L02036X/1), BBSRC (BB/J015873/1), MRC (MR/N00275X/1), Diabetes UK (BDA/11/0004210, BDA/15/0005275, BDA 16/0005485) and Imperial Confidence in Concept (ICiC) grants, and a Royal Society Wolfson Research Merit Award; A.T. was funded by an MRC New Investigator Research Grant (MR/M012646/1) and a Diabetes UK Early Career Small Grant (16/0005441). The views expressed are those of the authors and not necessarily those of the funders, the NHS, the NIHR or the Department of Health. MIN6B1 cells were kindly provided by Prof. Philippe Halban (University of Geneva, Switzerland) with permission from Prof. Jun-ichi Miyazaki (University of Osaka, Japan) who produced the maternal MIN6 cell line. INS-1 832/3 cells were kindly provided by Prof. Christopher Newgard (Duke University, USA). Human islets from Milan and Geneva were provided through the European Consortium for Islet Transplantation (ECIT), sponsored by JDRF (1-RSC-2014-90-I-X and 1-RSC-2014-100-I-X, respectively). Human islets from Edmonton were provided by the Clinical Islet Transplant Program and by the Alberta Diabetes Institute Islet Core at the University of Alberta with the assistance of the Human Organ Procurement and Exchange (HOPE) program, Trillium Gift of Life Network (TGLN) and other Canadian organ procurement organizations. Human islets from Oxford were isolated within the Diabetes Research and Wellness Foundation (DRWF) Human Islet Isolation Facility. This was supported by the National Institute for Health Research (NIHR) Oxford Biomedical Research Center (BRC) and by a grant from the Juvenile Diabetes Research Foundation (JDRF). We thank Gala Farooq for assistance with glucose tolerance tests, Zainab Malik, Laura-Jane Ball, and Natarin Caengprasath for technical assistance, Dr. Emlyn Corrin for assistance with trafficking calculations, Dr. James Minnion for helpful discussions, and Prof. Arthur Christopoulos for advice on biased agonism.

## Author contributions

A.T. and S.R.B. contributed equally to and jointly supervised this work. B.J., T.T., G.A.R., A.T., and S.R.B. conceived the study and designed all experiments. B.M.O. provided expertize on in vivo study design. B.J., T.B., N.K., P.C., B.M.O., R.S., B.J.C. and A.T. performed all experiments and analyzed data. R.G. and N.A. performed and interpreted liver histology. I.R.C., A.C.H., and A.I. contributed novel reagents. D.B., P.R.J., L.P., P.M., and A.M.J.S. supplied human islets. B.J., G.A.R., A.T., and S.R.B. wrote the paper. All authors reviewed and approved the paper.

## Additional information

**Competing interests:** The authors declare no competing interests.

