## [Peer Review File · Nature Communications]

Reviewers' comments:

Reviewer #1 (Remarks to the Author):

This is an interesting and important study.

It identifies minimal modifications within the N-terminus of exendin-4 modified peptides that alter ligand behaviour. Specifically they appear to alter ligand residence times, ligand-dependent receptor trafficking, and signalling (bias). An exemplar peptide with rapid off-rate had longer lasting benefit in glucose handling in animal models of HFHS diet, compared with exendin-4; additional potential benefit for steatosis was also seen. In contrast no differences in body weight or aversion were observed, and pharmacokinetics were not different.

The differences were principally linked to prolonged "exposure" scenarios, and may relate to differential effects on receptor regulation, although this is not directly established.

This Phe1-Ex-4 peptide may have therapeutic advantage over current GLP-1R agonists.

Clinically used GLP-1R peptide mimetics had minor differences in binding kinetics that did not appear to correlate with specifics of receptor trafficking.

Some caveats that should be emphasised.

Most of the data on receptor trafficking are established with high pharmacological concentrations of ligands (far in excess of what would be seen in vivo). As noted in the discussion, recycling versus lysosomal degradation is influenced by ligand concentration. As such the high concentrations used for these studies likely predisposes the system to reduced recycling of control ligands, relative to what is likely to occur physiologically, or even pharmacologically (at in vivo concentrations).

The same is likely to impact on prolonged exposure studies. It would have been interesting to see if in vitro differences were also seen if long-term pre-exposure was manifest at lower concentrations (eg. subnanomolar).

Nonetheless, the studies identify distinctions in ligand behaviour that can be correlated to distinct function in full-concentration response analyses.

I am not sure why the authors find a lack of correlation between cAMP and insulin secretion surprising. For example, cumulative cAMP is virtually identical in low and high glucose in INS-1 832 cells (despite no insulin secretion in the former).

There needs to be greater emphasis on the gaps in the study – particularly with reference to other potential effects of the ligand substitutions on signalling. There are likely many other effects (not measured in the current study) that occur as a consequence of the altered sequence. This does not invalidate the very interesting results but needs to be highlighted.

While the current study provides important new data, how the different signal components interplay for optimal insulin secretion is still not understood.

It is pleasing to see the author's cognisance of the influence of kinetics in bias determination. However, measuring all endpoints at the same time point does not alleviate issues in data interpretation. Indeed, it may equally miss more relevant signalling points. I am not even sure what 90 min of accumulated cAMP is representative of, physiologically. At what time points do peak responses occur? What is the relative difference in responses if measured at peak versus delayed time points?

The chosen Path-hunter cell line operates by enzyme complementation; this is likely to impact on all signalling outcomes.

In conditions of low receptor expression, peptide-agonist driven arrestin recruitment is transient, with a peak in minutes. The long sustained response that is observed in some systems is an artefact. This is also a general problem with interpretation of the DiscoverX path-hunter assay; as a complementation assay it is both poorly reversible and highly amplified.

Also, while not apparent from their website, DiscoverX often substitutes the V2 vasopressin C-tail onto receptors for the Path-hunter assay. Can the authors confirm that this was not done for the system in the current work?

For Figure 7, I am impressed that the authors have used human islets, however, I am not entirely convinced by the presented data. In the iCa^{2+} data, there are variable KCl responses that do not seem to have been taken into account. Significance in the acute versus 16h appears to be influenced by a single point in the acute (where ex-phe1 is lower than ex-4).

Phe1-exendin has an ~ 1 log lower affinity compared to the other peptides (ex-4; D3-Ex-4), and furthermore, FITC-Phe1-Ex-4 is nearly 1 log lower potency than the unlabelled parent (in contrast to the "comparable" claim in the text). This should be explicitly noted.

The authors note that they fit data with a 4-parameter logistic equation. As such they should report the Hill slopes.

Minor:

Abstract – last line, "...greater clinical efficacy.." is not correct, I assume they mean "in vivo".

Reviewer #2 (Remarks to the Author):

General Comments:

This manuscript by Jones et al. deals with a very interesting and timely question. The role of endocytosis in the therapeutic action of GLP1 agonists and whether or not a biased agonist favoring or not endocytosis could have better therapeutic value. The manuscript presents interesting data using a variety of pharmacological, cell biology and in vivo approaches. However, in its present form, it is quite difficult to follow. Many questions are addressed, some of them in a superficial way, resulting in a dilution of the main message. Many of the experiments are only superficially described making them difficult to interpret. Also, in many cases, the presentation of the data is not satisfactory since very little raw data are shown and the type of normalization varies from experiments to experiments sometime being in %, sometime in fold stimulation. Also, some of the data are shown for the 3 peptides whereas other are only shown for 2. The reasons for these differences are not explained and not intuitively evident. At the methodological levels the authors used a number of original approaches that are interesting but they are not always well described and their validation with proper control is not presented. Two examples are the measurement of the pH in different endosomal compartment where reside the receptor following activation by different peptides and the use of BETP to assess the importance of the residency time. Finally, the far-reaching conclusions as presented in the abstract are not fully supported by a careful analysis of the data.

Specific comments:

Introduction: The sentence: 'Sustained signaling by internalized GLP1-Rs was reported to be required for insulin release⁸; however, the gastric emptying effect of pharmacological GLP-1R agonists undergoes rapid tachyphylaxis, consistent with a desensitizing role of internalization' is confusing. Desensitization is classically not the result of internalization but rather a reduced ability of the receptors to productively engage their effectors as a result of phosphorylation and arrestin binding. The internalization is generally seen as either a mechanism of resensitization and recycling, non-canonical signaling or an intermediate toward receptor down-regulation depending of the receptors and the time of stimulation. Also, the authors cite reference 8 to support that sustained GLP1-mediated signaling in the endosomes is required for insulin secretion. In fact in the

paper by Kuna et al cited here, the impact of endocytosis on insulin secretion is tested with the use of dynasore (an inhibitor of endocytosis that can be quite toxic) and the treatment with dynasore did not reduce the GLP1-promoted insulin secretion it only increase the basal insulin secretion making the difference between basal and GP1-stimulated secretion smaller; a result difficult to interpret.

Supplementary Fig 1: Statistical analysis should be performed to assess the difference in internalization cAMP production and Insulin secretion to identify which analogue is significantly different from ex4. Also the Emax of the cAMP production should be presented.

Fig 1 the authors compared the relationships between cAMP production or internalization with insulin secretion. For cAMP they present the potencies whereas for internalization, they present the efficacy. To make pharmacological sense, the same pharmacological parameters (ie: efficacy or potency) should be compared. In fact, both should be looked at and full dose responses for all responses should be shown. This is done partially in Fig 3 but not for internalization and unfortunately the responses are normalized. As it stands the conclusion drawn by the authors is not supported by the data especially that the insulin secretion is measure at a single concentration (presumably a maximal concentration; the concentration of the analogues used for the insulin secretion should be shown) thus making the possibility of a correlation with the potency of cAMP production highly improbable in any case.

Fig 1: The data for Ex4 should be show in addition to those of ex-asp3 and ex-phe1. This is needed to make an appropriate comparison. The difference between ex-phe1 and ex-asp3 shown in panel e is not convincing (albeit apparently statistically significant), the difference is mainly due to a single point in the ex-asp3 condiiton. It is also notable that only 3 data points are shown for ex-asp3 whereas 4 are presented for ex-phe1. It is not clear why the format of the figures presenting the insulin secretion in INS-1 cells is different than for MN6B1 cells. It is also not clear why the results are shown in absolute value of the 16 hour treatment for INS-1 cells but in % of ex4 for all the others. All panels should be in absolute data. I suspect that for the MN6B cells no statistical significance would be detected between ex-4 and ex-phe1.

Supplementary Fig 2: It is not clear why the authors state `...a similar agonist rank order for protection against apoptosis during...' In fact in contrast to what is seen for insulin secretion in supplementary fig 1b and Fig 1d, ex-phe1 is not better than ex4 to promote apoptosis, it is equi-efficacious. The importance of the apoptosis data for the main story is not intuitively evident.

It is not clear why the endocytosis Is presented in absolute value for the CHOK1cells but in % in MIN6B cells. Also in contrast with what was shown in Supplementary Fig1b and Fig 1a, the ex-asp3 is not promoting more internalization than ex4. This lack of internal consistency of the data raises concerns about the robustness of the assays. Another example of internal inconsistency is the time of residence for ex4 reported as about 25 min in Fig 6b whereas it is 40 min in Fig 4b yet the residence time for ex-phe1 or ex-asp3 are not shown in Fig 6b. This would be needed for comparison purposes.

Microscopy images for ex-asp3 should also be shown in Fig 2e. The illustrations shown in Fig 2g are not convincing since different regions of the cells are shown for the two different peptides. To make the interpretation possible, the same regions that would illustrate that a different number of receptors are found in the different structures would be needed.

Fig 2d: How recycling vs internalization rates were estimated needs to be better explained. Steady state loss of cell surface receptor number is usually an equilibrium between endocytosis and recycling. How the authors differentiated between the two is not clear. In particular, how recycling could be deduced from the image in supplementary Fig 3 need to be explained. The data seems to be showing that a lower number of receptor is in the endosomes following treatment with ex-asp3 than with ex-4 which is counter-intuitive given the internalization data shown in Fig1.

The data presented in Fig 2 ce and in Supplementary Fig 3a and b seem to be very similar and reporting the same thing. Why are these data shown twice with differences in the presentation?

Fig 2i j: the effect of ex-phe1 pre-treatment on the calcium response was much more dramatic than the effect on cAMP which appears to be marginal. Because the results are shown as % of pretreatment for one assay and fold response in the other, it is difficult to conclude on the meaning of these differences. Showing the raw data would help better understanding the meaning of these results.

Supplementary Fig 6: The authors interpret the results to mean that ex-phe1 stimulate cAMP production for longer period of time. Because the data are shown as fold of forskolin stimulation a value of 1 or 0.5 still means that there is stimulation thus all compounds could maintain stimulation for 24 hours. The real difference is that for some reason at 6 and 24 hours the cAMP stimulation is now getting higher for ex-phe1 whereas this was not the case at 3 hours. This may indicate a different balance between production and degradation of cAMP but it is not clear why it would appear only a 6 hours and not 3 hours since the internalization occurs in the minute time scale and the difference between the compounds is more obvious at earlier time points (ex 20 min) than at later ones.

Fig 3 g,g,h: Although they reach statistical significance the effect of the siRNA on the endocytosis and on the insulin secretion are marginal. Given the variance in the data in Fig 3h, it is even surprising that it a statistical difference could be found. These results should be confirmed using another approach to knock down arrestins; for example, using dominant-negative mutants or arrestin KO cells. Also confirming the role of endocytosis in the lack of sustained insulin secretion should be tested using other endocytosis inhibitors.

The observation that persistent cAMP production following endocytosis is observed with ex-4 and ex-asp3 but not ex-phe1 is in contradiction with the main hypothesis. This needs further experiments to rationalize this difference. The explanation of a possible higher number of receptor in CHO-K1 is insufficient.

The experiment with BETP needs to be better explained. It is not evident for this reviewer how to interpret these data.

Fig 6: The authors state that differential effects of the clinically available compounds are modest. This is certainly not the case for the binding kinetics. Where two of the compounds show significant difference. Some of them also show difference in internalization and recycling values reported.

Although Ex-phe1 is indeed the most efficacious (not potent as written by the authors; potency refers to EC50 not maximal response), the difference with 3 of the drugs is not that marked. This need to be discussed in relation with their internalization and binding properties.

Fig. 7g and h: The authors conclude that there is a difference between ex4 and ex-phe1 for the chronic treatment (16h) and not for the acute treatment. However, this is only due to a difference in the variability for the acute response and most likely only one experiment explains why the difference between ex4 and ex-phe1 does not reach statistical difference in the acute case.

Fig 8: It is surprising that both ex4 and ex-phe1 promote a very dramatic reduction in cumulative food intake over a 6h time period in Fig 8 but that no such decrease is observed in chronic treatment for many days in Fig 9. Yet there is a reduction in body weight in these chronic treatments even for the vehicle. These inconsistencies need to be explained and controlled for.

The authors used a conditional taste aversion test as a surrogate for nausea. They conclude that

ex-phe 1 did not show evidence of nausea. Yet no difference was found with ex-4, How can this be interpreted in relation to the different properties of ex-4 and ex-phe1 regarding internalization? the other clinically used ligands should have been used for comparative purposes in this assay.

Reviewer #3 (Remarks to the Author):

These elegant and extremely thorough studies by Jones and colleagues potentially establish a role for GPCR-agonist interactions that specifically modulate receptor kinetics (internalization, recycling and trafficking) with phenotypic outcomes. Specifically, the authors have identified mutants that modify the dynamics of the GLP-1 receptor (GLP-1R) and suggest that GLP-1R agonists that increase the availability of GLP-1R at the cell surface have improved insulinotropic and glucose-lowering properties. This adds to the growing interest of the phenotypic effects of "biased signaling" via GPCRs but adds a novel aspect of assessing the effects not on signaling events but on receptor dynamics. The studies utilize a variety of novel and state-of-the art molecular techniques to thoroughly assess the effect of mutants of the GLP-1 R agonist exendin-4 (ex4) on various aspects of GLP-1R dynamics (internalization, recycling, residence time of the agonists on the receptor both extra- and intracellularly, etc.) and test the efficacy of these mutants on in vivo glucose control and both cell-based and in vivo insulin secretion. The overarching conclusion is that agonists that promote extended cell surface expression of the GLP-1R may provide a novel therapeutic strategy for improved GLP-1 based drugs. This is a significant finding that merits further exploration. Having said this, there are several issues that need to be addressed. In general, the two main issues revolve around: 1) the feasibility of correlating molecular observations made in a CHO line over-expressing the GLP-1R with a phenotypic effect in beta cell lines expressing endogenous GLP-1R levels (this is partially addressed in some, but not all, experiments), and 2) correlating effects of acute ex4/modified ex4 peptide exposure on GLP-1R internalization/trafficking on differences chronic insulin secretion, especially since acute insulin secretion appears to be unaffected by the various ex4 peptides.

Major points:

1. There is a significant issue with regards to the kinetics of the various measurements. The studies begin with a correlation between acute measurements (GLP-1R internalization, cAMP production) and a "chronic" insulin secretion index. The internalization and cAMP experiments were conducted following a 90 min exposure to the various GLP-1R agonists, yet the insulin secretion index that showed a different effect of the various agonists was determined following a 16h exposure of the agonists. On the surface, this would not seem like an issue, except for the fact that the authors found that acute exposure of the agonists (60 min – similar to the time frame of the internalization experiments) led to no difference in the insulin secretion index induced by ex-phe1 (low GLP-1R internalizer) and ex-asp3 (high GLP-1R internalizer). Therefore, if the authors had performed a correlation analysis of the acute GLP-1R internalization capacity vs. the acute insulin secretion index of the various ex4 molecules, I presume that they would have found either no correlation or a positive correlation. One experiment that could address this issue is to measure chronic GLP-1R internalization and correlate that to chronic insulin secretion, and preferably in the same cell line (see points #1 and #2 below).

2. It appears that some of the observations are cell type-dependent. It is understood that there will be differences between CHO cells over-expressing the GLP-1R and beta cell lines (INS1 and MIN6 cells) that express endogenous levels of the GLP-1R, and to the authors' credit, this is briefly discussed. However, there were also differences between INS1 and MIN 6 cells. For example, when looking at the effects of ex-phe1 vs. ex-asp3 on acute insulin secretion, there were no differences between the peptides in INS1 cells but there were in MIN6 cells (Figs. 1f and 1g). This issue is brought up since different assays were sometimes performed in some cell lines but not all. For example, as indexes of desensitization, Ca⁺ signaling was done in one cell type and cAMP

generation was done in another ((Figs. 21 and 2j). It raises the possibility that perhaps some of the positive observations were not observed in all cell lines. It would have been more appropriate for consistency to either perform all assays in all three cell lines or just to choose one cell line and run all assays on this one cell line. Again, to the authors' credit, they did generate beta cell lines with reagents that allowed them to look at certain events (e.g. MIN6B1-SNAP-GLP1R cells) in a "native context" and did compare INS1 and MIN6 cells with almost every experiment.

3. Following from points #1 and #2 above, Fig. 1e shows a difference in the 16h insulin secretion capacity between ex-phe1 and ex-asp3 in MIN6 cells. However, what is not shown is whether there is a significant difference in the insulin secretion capacity between native ex4 and ex4-phe1 in MIN6 cells. This is an important point since the authors later show a difference in GLP-1R internalization between ex4 and ex4-phe1 in MIN6 cells. Therefore, to strengthen the argument that internalization capacity is correlated with insulin secretion capacity, the authors should show differences between ex4 and ex4-phe1 in both internalization and insulin secretion in the same cell line (whether it is MIN6 or INS1).

4. The design of the "acute" and "chronic" experiments measuring insulin secretion does not allow for a truly fair comparison on potential effects of acute vs. chronic peptide exposure on insulin secretion. As described, the acute experiments began with beta cell lines seeded for 24h in low glucose (3mM) and then treated with high glucose media (11mM) with/without agonists for 1h. For the chronic experiments, the cells were seeded in high glucose media and were exposed to agonists overnight (16h). Therefore, the cells used for the chronic exposure were not exposed to the same low glucose stress as the cells in the acute experiments. To make a true comparison of both conditions, the chronic experiments should be conducted in cells exposed to low glucose for 24h and then switched to a high glucose +/- peptides overnight. Alternatively, the cells in the acute experiments should be exposed directly to high glucose media.

5. With regards to the chronic insulin secretion experiments, it is not clear whether the elevated insulin levels were due to a persistently higher secretion of insulin or whether most of the insulin secretion occurred early during the peptide exposure period and the insulin lingered (not quite sure what the half-life is of insulin in cell culture conditions). Perhaps a better approach would be to look at a time course of insulin levels throughout the chronic exposure and not just at the 16h time point.

6. Many of the experiments were conducted with very small n's (2-3) and it is difficult to see how statistical significance was achieved with such low numbers of replicates.

Minor points:

1. The authors state in Supplementary Figs. 2b,c that there is a similar rank order of ex4 peptides with regards to receptor internalization and protection from ER stress or glucolipotoxicity. This does not appear to be the case since there appears to be a difference between ex4 and ex4-phe1 with regards to receptor internalization (in CHO and MIN6 cells) but not with regards to ER stress in INS1 cells or glucolipotoxicity in MIN6 cells. Again, highlighting points made above, why were different cell lines used for ER stress vs. glucolipotoxicity experiments? Is it that the effects of the various peptides were cell type-dependent? – e.g. was ex4-phe1 detrimental towards the response to ER stress in MIN6 cells?

2. The next to last paragraph on Page 8 ("GPCR recycling restores...") described data focusing on receptor responsiveness and desensitization, yet the last line in the paragraph details data (Supplementary Fig. 6) that do not truly support any issues of receptor desensitization and even highlight many of the major issues discussed above. Sustained cAMP generation in the presence of ex4-phe1 vs. ex4 is not necessarily indicative of any issues of receptor sensitization. Furthermore, if ex4-phe1 promotes more sustained cAMP generation compared to ex4 in INS1 cells (Supplementary Fig. 6a), then could this not be a factor in the greater chronic insulin secretion in

INS1 cells (Fig. 1d and Supplementary Fig. 1b)? This would contradict the authors' initial assertion that "Surprisingly, cAMP response was poorly predictive of prolonged insulin release" (Page 6). This again highlights the importance of correlating acute molecular events with acute phenotypes and chronic molecular events with chronic phenotypes (Major point #1).

Response to Reviewers' Comments:

Reviewer #1 (Remarks to the Author):

This is an interesting and important study. It identifies minimal modifications within the N-terminus of exendin-4 modified peptides that alter ligand behavior. Specifically they appear to alter ligand residence times, ligand-dependent receptor trafficking, and signaling (bias). An exemplar peptide with rapid off-rate had longer lasting benefit in glucose handling in animal models of HFHS diet, compared with exendin-4; additional potential benefit for steatosis was also seen. In contrast no differences in body weight or aversion were observed, and pharmacokinetics were not different. The differences were principally linked to prolonged "exposure" scenarios, and may relate to differential effects on receptor regulation, although this is not directly established. This Phe1-Ex-4 peptide may have therapeutic advantage over current GLP-1R agonists. Clinically used GLP-1R peptide mimetics had minor differences in binding kinetics that did not appear to correlate with specifics of receptor trafficking.

Some caveats that should be emphasized: Most of the data on receptor trafficking are established with high pharmacological concentrations of ligands (far in excess of what would be seen in vivo). As noted in the discussion, recycling versus lysosomal degradation is influenced by ligand concentration. As such the high concentrations used for these studies likely predisposes the system to reduced recycling of control ligands, relative to what is likely to occur physiologically, or even pharmacologically (at in vivo concentrations). The same is likely to impact on prolonged exposure studies. It would have been interesting to see if in vitro differences were also seen if long-term pre-exposure was manifest at lower concentrations (e.g. subnanomolar).

We thank the referee for his/her careful and detailed review of our manuscript. We deal with the points raised here below:

In line with the above and other reviewers' comments, we repeated several experiments over wider dose ranges to better understand the behavior of the compounds measured in this study. In particular, we now provide dose-response data for internalization and insulin secretion for all ligands (Supplementary Fig. 1, with data for exendin-4, exendin-phe1 and exendin-asp3 shown in Fig. 1f,g), as well as insulin secretion after β -arrestin depletion (Fig. 4g). These internalization measurements have identified clear differences in maximal responses (with prolonged incubations); we also detected no agonist-induced internalization below 1 nM in this assay. Consequently, we did not examine the details of trafficking effects at sub-nanomolar concentrations. We note also that an early study of GLP-1R internalization found no evidence of surface receptor loss when GLP-1 was applied in the high picomolar range¹. However, we also recognize that this system cannot exclude small numbers of GLP-1Rs (relative to the total number expressed) undergoing agonist-induced endocytosis when treated at lower doses.

From the insulin secretion data, it is apparent that increased agonist efficacy during prolonged incubations is generally accompanied by reduced potency, meaning that, at low concentrations, agonists with increased maximal effect are less insulintropic than exendin-4. Exendin-phe1 becomes more insulintropic than exendin-4 at concentrations above ~1 nM with the longer 16 h incubation period. Whilst extrapolating *in vitro* concentrations to the *in vivo* situation is always problematic, we note peak exendin-4 concentrations at or above 10 nM have been reported from the Amylin development program^{2,3}. The rapid internalization, and slow recycling properties of the GLP-1R exposed to high affinity ligands is relevant here, as even if periods of high circulating agonist concentration are short, they can exert a longer lasting effect via the mechanisms discussed in our article. Overall, while the cellular effects favoring agonists such as exendin-phe1 tend to be most apparent at higher doses, the fact that enhanced insulintropism was also clearly seen *in vivo* at

standard doses supports, in our view, the validity of the conditions used to establish these findings. Nevertheless, we recognize the importance of the issue raised by the reviewer and have added this as a discussion point (page 18, paragraph 1).

Nonetheless, the study identifies distinctions in ligand behavior that can be correlated to distinct function in full-concentration response analyses.

I am not sure why the authors find a lack of correlation between cAMP and insulin secretion surprising. For example, cumulative cAMP is virtually identical in low and high glucose in INS-1 832/3 cells (despite no insulin secretion in the former).

Our rationale for comparing cAMP potency with insulin secretion was that several studies⁴⁻⁸ implicate cAMP as the primary signaling intermediate linking GLP-1R activation to insulin release, and, consequently, GLP-1R agonist cAMP responses are often measured as an important part of candidate selection during drug development. We wished to make the point that this strategy may have missed the potentially advantageous enhanced maximal insulin release seen with exendin-phe1 and similar compounds. However, we agree that formally correlating cAMP potency with single dose insulin secretion is not the best way to show this - this comparison has been removed and the text has been rewritten to better explain our reasoning (page 7, paragraph 1).

There needs to be greater emphasis on the gaps in the study – particularly with reference to other potential effects of the ligand substitutions on signaling. There are likely many other effects (not measured in the current study) that occur as a consequence of the altered sequence. This does not invalidate the very interesting results but needs to be highlighted. While the current study provides important new data, how the different signal components interplay for optimal insulin secretion is still not understood.

We agree with this point made by the reviewer and have added this as a discussion point, mentioning in particular how untargeted –omic-type approaches may be needed to identify additional pathways involved (page 19, paragraph 2).

It is pleasing to see the author’s cognizance of the influence of kinetics in bias determination. However, measuring all endpoints at the same time point does not alleviate issues in data interpretation. Indeed, it may equally miss more relevant signaling points. I am not even sure what 90 min of accumulated cAMP is representative of, physiologically. At what time points do peak responses occur? What is the relative difference in responses if measured at peak versus delayed time points?

We thank the reviewer for making this important point. Recruitment of G proteins and arrestins occurs rapidly (within seconds)^{9,10}, and peak cAMP response is typically seen at ~10 min with endogenous receptor expression levels¹¹, as we found with INS-1 832/3 cells in our manuscript (Fig. 3a). Therefore, we repeated measurements of cAMP and β -arrestin signaling responses at 10 min for exendin-4, exendin-phe1 and exendin-asp3. These experiments (Fig. 4a) recapitulated the bias profile seen at 90 min, with the caveat that β -arrestin-1 responses to exendin-phe1 were essentially absent, preventing bias quantification. We also examined β -arrestin recruitment by confocal microscopy at an early time-point (see point below, images are shown in Fig. 4e).

The chosen Path-hunter cell line operates by enzyme complementation; this is likely to impact on all signaling outcomes. In conditions of low receptor expression, peptide-agonist driven arrestin recruitment is transient, with a peak in minutes. The long sustained response that is observed in some systems is an artifact. This is also a general problem with interpretation of the DiscoverX path-hunter assay; as a complementation assay it is both poorly reversible and highly amplified. Also, while not apparent from their website, DiscoverX often substitutes the

V2 vasopressin C-tail onto receptors for the PathHunter assay. Can the authors confirm that this was not done for the system in the current work?

We specifically confirmed with DiscoverX that the GLP-1R PathHunter systems used in our manuscript do not include a V2R tail (correspondence available on request). We certainly agree with the reviewer that the irreversible β -arrestin recruitment with the PathHunter system differs from the physiological situation. The high signals obtained are indicative of propensity for β -arrestin recruitment but disregard any differences in dissociation rates. It is worth noting that the GLP-1R- β -arrestin-2 interaction, as measured by FRET microscopy, was found to be long-lived (>30 min), albeit in an overexpressing system¹². Nevertheless, we have added a comment in the manuscript to emphasize this aspect of the PathHunter assay (page 11, paragraph 1). We also corroborated our findings by observing β -arrestin recruitment by microscopy, which was readily apparent with exendin-4 and exendin-asp3 but not exendin-phe1 (Fig. 4e).

For Figure 7, I am impressed that the authors have used human islets, however, I am not entirely convinced by the presented data. In the iCa^{2+} data, there are variable KCl responses that do not seem to have been taken into account. Significance in the acute *versus* 16 h appears to be influenced by a single point in the acute (where ex-phe1 is lower than ex-4).

We presume the query about the KCl responses refers to AUC calculations. If so, we agree that our description of what the AUC refers to missed the important explanatory point that this was calculated for the period between addition of agonist and addition of KCl, i.e., KCl responses were not included. We have consequently adjusted the figure legend to highlight this.

We also note that it could be argued that the KCl responses are not identical between conditions and perhaps this could indicate underlying differences (by chance) in islets used for particular experiments, i.e. this could provide a convenient way of correcting for the inevitable differences between the responsiveness of different human islet preparations, reflecting a range of donor characteristics (age, sex, BMI, etc.). However, further analysis of these differences will not, in our view, significantly alter the interpretation of the data for agonist responses. Specifically, for acute responses (Fig. 7b,c), before the point of KCl addition, agonist-related differences have already raised the baseline from which the KCl effect begins (greater response to exendin-4). Calculating AUC following KCl addition relative to this new baseline shows both sets of islets responded essentially identically to this positive control.

In the desensitization experiment (Fig. 7d,e), we acknowledge that the responses to KCl in exendin-phe1-pretreated islets are less marked than in those pretreated with exendin-4. However, the pretreatment-related differential responses to re-challenge with acute exendin-4 (homologous desensitization) cannot be explained by differences in overall islet responsiveness, as this difference in KCl response suggests, if anything, that the measured response to exendin-4 after exendin-phe1 pretreatment is an underestimate. These AUCs are presented below (Fig. R1) for the reviewer's scrutiny, though in our view it is not essential to include them in the manuscript.

Fig. R1. Responses to KCl in human islets during GLP-1R agonist treatment. (a) AUC for KCl response after acute treatment with exendin-4 (ex4) or exendin-phe1 (ex-phe1), corresponding to Fig. 7b in main manuscript. (b) AUC for KCl response after overnight pretreatment with exendin-4 or exendin-phe1 (as indicated) and re-challenge with exendin-4, corresponding to Fig. 7d in main manuscript. In each case, AUC calculated relative to baseline established from average fluorescence 1 min prior to KCl addition. * $p < 0.05$ by unpaired t-test. Error bars indicate SEM.

Regarding the insulin responses, the reviewer is absolutely correct that the statistical significance of exendin-4 *versus* exendin-phe1 acute stimulation is influenced by one particular islet batch. If this “outlier” is removed, a significant difference emerges between the two treatments (favoring exendin-phe1). In the absence of a clear reason to exclude this particular experiment, we leave it in the manuscript and have added a comment in the text highlighting the apparent trend (page 15, paragraph 1). As cell type-related differences in the kinetics of differential insulin secretion are expected (for example, Fig. 1h,i), we suggest that this observation does not impact overall conclusions, and if anything, increases the potential clinical utility of exendin-phe1.

Phe1-exendin has an ~1 log lower affinity compared to the other peptides (ex-4; D3-Ex-4), and furthermore, FITC-Phe1-Ex-4 is nearly 1 log lower potency than the unlabeled parent (in contrast to the “comparable” claim in the text). This should be explicitly noted.

We thank the reviewer for this suggestion and have now indicated that the potency of exendin-phe1-FITC was up to 1 log unit lower than the non-FITC equivalent (page 8, paragraph 2).

The authors note that they fit data with a 4-parameter logistic equation. As such they should report the Hill slopes.

We agree, and Hill slopes for all agonists have now been added (Supplementary Fig. 1b). For the insulin secretion experiments, we found that globally constraining the Hill slope in each experiment provided more reliable curve fitting. Note that for insulin secretion, the Hill slope was significantly less than one, and significantly greater than one for some of the pharmacological responses (particularly cAMP). Therefore, we consider 4-parameter rather than 3-parameter a better fitting strategy.

Minor: Abstract – last line, “...greater clinical efficacy...” is not correct, I assume they mean “*in vivo*”.

We have modified the abstract as suggested.

Reviewer #2 (Remarks to the Author):

General Comments:

This manuscript by Jones et al. deals with a very interesting and timely question. The role of endocytosis in the therapeutic action of GLP1 agonists and whether or not a biased agonist favoring or not endocytosis could have better therapeutic value. The manuscript presents interesting data using a variety of pharmacological, cell biology and in vivo approaches. However, in its present form, it is quite difficult to follow. Many questions are addressed, some of them in a superficial way, resulting in a dilution of the main message. Many of the experiments are only superficially described making them difficult to interpret. Also, in many cases, the presentation of the data is not satisfactory since very little raw data are shown and the type of normalization varies from experiments to experiments sometime being in %, sometime in fold stimulation. Also, some of the data are shown for the 3 peptides whereas others are only shown for 2. The reasons for these differences are not explained and not intuitively evident. At the methodological levels the authors used a number of original approaches that are interesting but they are not always well described and their validation with proper control is not presented. Two examples are the measurement of the pH in different endosomal compartment where reside the receptor following activation by different peptides and the use of BETP to assess the importance of the residency time. Finally, the far-reaching conclusions as presented in the abstract are not fully supported by a careful analysis of the data.

We thank the referee for this summary, which we take on board. Specific points are addressed below, but in general:

1) We have aimed in the revised text to improve the explanation for experimental approaches, particularly for the two examples raised, which we agree were described a little too briefly. Further details are of course available in the Methods section but we hope that the rationale and principles for the assays are now apparent from the main text. We also aimed to improve the graphical depictions of some SNAP-tag-based assays.

2) Regarding normalization, we recognize there are different opinions on this matter. Although normalization has been performed in different ways in our manuscript, we have aimed for a consistent approach in line with recent recommendations¹³. In general, we prefer to express responses relative to either a basal or maximal response depending on the magnitude of the change. We find that, in beta cells with endogenous expression levels of GLP-1R, fold change from “basal” tend to be modest, making this an appropriate metric to quantify responses; for insulin release, when quantified relative to the glucose-alone response, this has the additional advantage of indicating the “incretin effect” (i.e. additional insulin release compared to glucose-stimulation alone). In contrast, responses obtained using cell lines overexpressing GLP-1R to high levels tend to involve very large fold changes (see graphs below), and, in many cases, establishing the basal response with a high degree of accuracy is difficult in a high-throughput setting due to limitations in assay dynamic ranges. In these cases, we prefer to express responses relative to the maximal response obtained with a control ligand or other stimulating compound, depending on the question asked by the assay. We note that expressing responses relative to a defined “max” is common in pharmacology research¹⁴⁻¹⁷. We have adapted our normalization strategy in selected cases; examples include the experiments with BETP, which, as an “ago-PAM”, inherently increases “basal” insulin secretion (in the absence of exendin-4), meaning that we consider displaying the data as “% release” more informative so the relative effect of BETP on exendin-4 response can be easily seen (Fig. 5I); and cAMP responses in stable SNAP-GLP-1R-expressing wild-type vs. “ β -arrestin-less” HEK293 CRISPR cells, where responses are adjusted to take account of the different levels of receptor expression (analogous to the approach taken by Wootten *et al.* when assessing signaling responses to GLP-1R mutants¹⁷).

3) We have made adjustments to the text in line with where the reviewer has highlighted possible differences in the conclusions that can be drawn from the data.

Specific comments:

Introduction: The sentence: ‘Sustained signaling by internalized GLP1-Rs was reported to be required for insulin release⁸; however, the gastric emptying effect of pharmacological GLP-1R agonists undergoes rapid tachyphylaxis, consistent with a desensitizing role of internalization’ is confusing. Desensitization is classically not the result of internalization but rather a reduced ability of the receptors to productively engage their effectors as a result of phosphorylation and arrestin binding. The internalization is generally seen as either a mechanism of resensitization and recycling, non-canonical signaling or an intermediate toward receptor down-regulation depending of the receptors and the time of stimulation. Also, the authors cite reference 8 to support that sustained GLP1-mediated signaling in the endosomes is required for insulin secretion. In fact in the paper by Kuna et al cited here, the impact of endocytosis on insulin secretion is tested with the use of dynasore (an inhibitor of endocytosis that can be quite toxic) and the treatment with dynasore did not reduce the GLP1-promoted insulin secretion it only increase the basal insulin secretion making the difference between basal and GP1-stimulated secretion smaller; a result difficult to interpret.

We recognize the points made by the reviewer and have amended this paragraph accordingly. Thus, in page 5, paragraph 2, we now state: “Sustained signaling by internalized GLP-1Rs has been reported, but without increasing insulin release^{18,19}”. The latter study also identified lysosomes as a major post-endocytic GLP-1R destination, raising the possibility that prolonged agonist exposure might result in GLP-1R degradation and down-regulation. In contrast, a proportion of GLP-1Rs is recycled back to the plasma membrane^{1,20}, an important resensitization mechanism^{21,22}.”

Supplementary Fig 1: Statistical analysis should be performed to assess the difference in internalization, cAMP production and insulin secretion to identify which analogue is significantly different from ex4. Also the E_{max} of the cAMP production should be presented.

We now present data in Supplementary Fig. 1 indicating full dose responses for all agonists for internalization, cAMP, β -arrestin-2 recruitment, and insulin secretion (16 h and 1 h). This includes logEC₅₀ values, E_{max}, and Hill slopes. Statistical analysis has been performed and is indicated in the tables. We found several agonists with statistically significant differences with regards to efficacy for internalization and prolonged insulin release. Results for exendin-4 *versus* exendin-phe1 and exendin-asp3 are given graphically at various points in the manuscript, but we did not do this for all agonists for the sake of brevity. We hope that providing these results numerically will be useful to readers who are interested to see these data.

Fig 1: The authors compared the relationships between cAMP production or internalization with insulin secretion. For cAMP they present the potencies whereas for internalization, they present the efficacy. To make pharmacological sense, the same pharmacological parameters (ie: efficacy or potency) should be compared. In fact, both should be looked at and full dose responses for all responses should be shown. This is done partially in Fig 3 but not for internalization and unfortunately the responses are normalized. As it stands the conclusion drawn by the authors is not supported by the data especially that the insulin secretion is measure at a single concentration (presumably a maximal concentration; the concentration of the analogues used for the insulin secretion should be shown) thus making the possibility of a correlation with the potency of cAMP production highly improbable in any case.

In line with the point made by the reviewer, we performed further experiments to obtain full dose responses for internalization (cell surface receptor loss) as well as insulin secretion for all agonists, presented in Supplementary Fig. 1. As the major finding of our manuscript relates to greater maximal insulin release, we have replotted the correlation analysis to focus on efficacies for insulinotropism *versus* internalization (in Fig. 1). By this analysis, it is still clear that agonists with reduced

internalization efficacy have increased maximal insulin responses. In the text (page 7, paragraph 1), we continue to make the point that cAMP EC₅₀ is a major metric used to evaluate GLP-1R agonists for potential clinical use, with high potency agonists usually preferred. It is relevant that the high efficacy (for insulin secretion) agonists described in this manuscript would be unlikely to be selected for further study on the basis of their reduced potencies (for cAMP) compared to exendin-4. Nevertheless, we accept the reviewer's point that a direct comparison of efficacy and potency measures is of limited use and have removed this plot.

Fig 1: The data for Ex4 should be shown in addition to those of ex-asp3 and ex-phe1. This is needed to make an appropriate comparison. The difference between ex-phe1 and ex-asp3 shown in panel e is not convincing (albeit apparently statistically significant), the difference is mainly due to a single point in the ex-asp3 condition. It is also notable that only 3 data points are shown for ex-asp3 whereas 4 are presented for ex-phe1. It is not clear why the format of the figures presenting the insulin secretion in INS-1 cells is different than for MN6B1 cells. It is also not clear why the results are shown in absolute value of the 16-hour treatment for INS-1 cells but in % of ex4 for all the others. All panels should be in absolute data. I suspect that for the MN6B cells no statistical significance would be detected between ex-4 and ex-phe1.

To address this issue, we have performed additional sets of acute and prolonged insulin secretion experiments in both cell types used (INS-1 832/3 and MIN6B1). Furthermore, the acute insulin results have been expanded to include all exendin-4-derived agonists described herein. These new data now replace the original dataset presented in the manuscript, but do not change the major conclusions. These results are provided in Supplementary Fig. 1, and shown graphically in Fig. 1a-d. All insulin secretion results are now expressed in the format used for the 16 h INS-1 832/3 experiments as suggested by the reviewer, which we agree is preferable.

Supplementary Fig 2: It is not clear why the authors state ‘...a similar agonist rank order for protection against apoptosis during...’ In fact in contrast to what is seen for insulin secretion in supplementary fig 1b and Fig 1d, ex-phe1 is not better than ex4 to promote apoptosis, it is equi-efficacious. The importance of the apoptosis data for the main story is not intuitively evident.

We have reworded the manuscript to emphasize that there was no additional anti-apoptotic effect of exendin-phe1 compared to exendin-4. Whilst our studies focus on regulated insulin release, we performed these experiments as the anti-apoptotic effect of GLP-1R agonism was reported as a major downstream consequence of β -arrestin recruitment²³; as such, we wished to highlight the somewhat surprising finding that an agonist with increased β -arrestin recruitment was paradoxically less able to prevent beta cell apoptosis.

It is not clear why the endocytosis is presented in absolute value for the CHO-K1 cells but in % in MIN6B1 cells. Also in contrast with what was shown in Supplementary Fig1b and Fig 1a, the ex-asp3 is not promoting more internalization than ex4. This lack of internal consistency of the data raises concerns about the robustness of the assays. Another example of internal inconsistency is the time of residence for ex4, reported as about 25 min in Fig 6b whereas it is 40 min in Fig 4b yet the residence time for ex-phe1 or ex-asp3 are not shown in Fig 6b. This would be needed for comparison purposes.

We again thank the referee for these insightful observations. We have assessed GLP-1R endocytosis using several methodologies, and, consequently, quantified these in different ways as appropriate to the assay. The terminology we used therefore reflects what is actually measured in the assay.

In Fig. 1 and Supplementary Fig. 1, agonist-induced loss of cell surface receptors was determined by post-stimulation labeling of residual receptors, with signal expressed relative to vehicle-stimulated

cells. We recognize that this measure includes contributions from endocytosis and recycling and is thus a “net” measure, as in fact are all internalization measurements in our manuscript.

In the DERET assay (now in Supplementary Fig. 3 along with a graphical explanation of the assay principle), units are arbitrary as they reflect a ratiometric FRET signal resulting from surface receptors (labeled prior to agonist addition) in close proximity to fluorescein-containing extracellular buffer, with changes in signal indicative of movement of receptor away from the cell surface; in this case, it is the change in FRET signal relative to individual well baseline which is presented to minimize contribution of well-to-well differences in labeling efficiency. The ratio was determined as 620 nm / 520 nm signal prior to baseline correction, so that internalization resulted in a positive inflection on the graph.

In contrast, for the FACS measurements presented in what is now Fig. 2b and c, internalization was measured by labeling receptor prior to agonist addition, followed by cleavage of surface receptor, with measurement of internalized fluorescence by FACS (i.e. internalized receptor was directly detected rather than inferred from loss of surface receptor); in this case, parallel measurement of labeled cells which were vehicle-treated and did not undergo surface receptor cleavage was used to establish “total receptor” (100%), against which agonist effects are shown. This assay also has a new graphical scheme to make the principle clearer (Fig. 2a).

The reviewer rightly points out that the increased internalization with exendin-asp3 compared to exendin-4, identified in our original data from Fig. 1 (now replaced with full dose response analysis), was not consistently seen when trafficking effects were evaluated using other assays in Fig. 2 and Supplementary Fig. 3. There are several possible explanations for this observation: 1) A notable property of exendin-asp3 compared to exendin-4 is a reduced rate of receptor recycling (as indicated in Fig. 2c); in Fig. 1, net internalization was measured at 90 min, but only up to 60 min in Fig. 2, and hence differences in internalization may have become apparent with longer incubations due to reduced recycling with exendin-asp3; 2) In Fig. 1 (and Supplementary Fig. 1), CHO-SNAP-GLP-1R cells were used, but Fig. 2c shows results from MIN6B1-SNAP-GLP-1R beta cells, in which trafficking effects might be subtly different; and 3) The DERET assay now in Supplementary Fig. 3 provides useful kinetic information, but its dynamic range (maximum signal change from baseline ~3x) makes discriminating small differences between maximum agonist internalization difficult. We should highlight that the response differences between exendin-4 and exendin-asp3 in Fig. 1 are quite modest (although reproducible) in comparison to exendin-4 vs. exendin-phe1. To investigate some of these possibilities, we performed further experiments with longer incubations (16 h) to determine if differences in net surface loss became wider (Fig. 2i-k, Supplementary Fig. 5b and 5d-h, described in page 9, paragraph 2). Interestingly, the difference between exendin-4 and exendin-asp3 was maintained, at least in some assays, but did not widen. It may be that a maximal response is already attained at intermediate time-points, at least at the dose tested; we note that the potency for cell surface receptor loss under these longer incubations (Supplementary Fig. 5b) indicates differences might be more apparent at lower doses.

The reviewer also points out the discrepancy for residence time measurements for exendin-4 in different series of experiments. As with the majority of our experiments, treatments displayed on the same panel have been compared in parallel, which we submit allows valid conclusions to be drawn by comparison with the control ligand (usually exendin-4), even when the control ligand does not perform identically in independent sets of experiments. Various factors may influence the response to control ligands, such as cell passage number with associated phenotypic drift (we have noted for example significant effects on binding kinetics resulting from other membrane components), as well as cumulative effects of minor variability in aliquoting, buffers, etc. Furthermore, as residence time is determined as the reciprocal of the agonist dissociation rate constant (k_{off}), the latter being the parameter determined by curve fitting, numerically small differences in k_{off} can translate to apparently larger differences in residence time. Nevertheless, as we determined the kinetic binding parameters for exendin-4-FITC in each assay, we now additionally present the data from the same experiments

for exendin-phe1 expressed relative to the results for exendin-4-FITC in that experiment (now in Supplementary Fig. 9). We did the same for internalization and recycling measurements performed separately but using the same protocol as for the “clinical” GLP-1R agonists. In our view, this provides a convenient strategy to compare the results from comparator compounds with those of exendin-phe1.

Microscopy images for ex-asp3 should also be shown in Fig 2e. The illustrations shown in Fig 2g are not convincing since different regions of the cells are shown for the two different peptides. To make the interpretation possible, the same regions that would illustrate that a different number of receptors are found in the different structures would be needed.

We have now performed additional experiments that demonstrate a lack of co-localization of exendin-asp3 with Rab11 (Fig. 2e). Regarding the electron microscopy images, images shown in the main figure (Fig. 2f) are of course representative, and have been quantified in Fig. 2g, but we now also provide images covering a wider area (see Supplementary Fig. 5).

Fig 2d: How recycling vs. internalization rates were estimated needs to be better explained. Steady state loss of cell surface receptor number is usually an equilibrium between endocytosis and recycling. How the authors differentiated between the two is not clear. In particular, how recycling could be deduced from the image in supplementary Fig 3 need to be explained. The data seems to be showing that a lower number of receptors is in the endosomes following treatment with ex-asp3 than with ex-4 which is counter-intuitive given the internalization data shown in Fig1.

We agree this an important point. We have aimed to improve the description of the methods used to describe trafficking. Full details are provided in the Methods section. In particular, we present an improved schematic for flow cytometry measurements of recycling in Fig. 2a. For the confocal microscopy images in Supplementary Fig. 3 – these are intended to demonstrate net internalization of SNAP-GLP-1R, labeled in advance, after exposure to different agonists or vehicle, and do not show recycling. The images with exendin-asp3 were obtained on a different occasion to the other agonists, which most likely explains the findings in the imaging noted by the reviewer. Therefore, we have subsequently repeated these experiments, with exendin-4, exendin-phe1 and exendin-asp3 treatments performed in parallel; we now include these images instead (Fig. 2d, Supplementary Fig. 3e). Nevertheless, the small increase in internalization seen with exendin-asp3, as indicated in Fig. 1, is difficult to demonstrate visually.

The data presented in Fig 2c and in Supplementary Fig 3a and b seem to be very similar and reporting the same thing. Why are these data shown twice with differences in the presentation?

In Fig. 2c (now 2b), FACS internalization results are shown in MIN6B1-SNAP-GLP-1R cells, whereas in Supplementary Fig. 3a (now 3c), results for CHO-SNAP-GLP-1R cells are shown. The figure legend titles now clearly indicate that all data in Fig. 2 is from MIN6-SNAP-GLP-1R cells and from CHO-SNAP-GLP-1R cells in Supplementary Fig. 3.

Fig 2i,j: the effect of ex-phe1 pre-treatment on the calcium response was much more dramatic than the effect on cAMP which appears to be marginal. Because the results are shown as % of pretreatment for one assay and fold response in the other, it is difficult to conclude on the meaning of these differences. Showing the raw data would help better understanding the meaning of these results.

These data are now presented in Fig. 3. As several cell types are used in this figure, each panel is now clearly labeled. In this case, the Ca^{2+} measurements (Fig. 3h) were performed in CHO-GLP-1R cells, whereas the cAMP results referred to are from INS-1 832/3 beta cells. We now present the

cAMP desensitization data relative to “basal” response (as we have consistently done for insulin secretion throughout), which we agree is more informative (Fig. 3g). The magnitude of the difference between agonists remains modest in this assay but is highly reproducible. Coupling to signaling pathways in CHO-GLP-1R cells is considerably greater than in beta cells (see point below) and we suspect this underlies the greater differences seen in this assay. Note that we have also performed further experiments (shown in Fig. 3c-f, described on page 10, paragraph 1) to investigate prolonged signaling in beta cells in the context of continuous agonist exposure with IBMX added only at the very end of the assay, to cause cAMP accumulation at a rate indicative of the incident signaling at that particular time point.

Supplementary Fig 6: The authors interpret the results to mean that ex-phe1 stimulate cAMP production for longer period of time. Because the data are shown as fold of forskolin stimulation a value of 1 or 0.5 still means that there is stimulation thus all compounds could maintain stimulation for 24 hours. The real difference is that for some reason at 6 and 24 hours the cAMP stimulation is now getting higher for ex-phe1 whereas this was not the case at 3 hours. This may indicate a different balance between production and degradation of cAMP but it is not clear why it would appear only a 6 hours and not 3 hours since the internalization occurs in the minute time scale and the difference between the compounds is more obvious at earlier time points (ex 20 min) than at later ones.

These data are now displayed in Fig. 3a,b. However, we suggest that our description was a little misleading, as, in the graphs referred to, all results are expressed relative to IBMX alone, and not to forskolin as suggested by the reviewer. Therefore, in these experiments that indicate accumulation of cAMP in the continuous presence of IBMX ± agonist, at each time-point, cells stimulated with IBMX alone are assigned a value of 1, so any value above 1 in the presence of agonist indicates net stimulation of cAMP production.

We agree with the reviewer that it is interesting that, at least for MIN6B1 cells, cAMP accumulation with exendin-phe1 becomes more apparent at 6 h. We submit that this reflects not just internalization differences, but also the propensity of exendin-phe1 to trigger faster receptor recycling, resulting in a larger number of re-stimulation events in the continual presence of agonist. Here, when cAMP degradation is relatively reduced by IBMX, net accumulation results over time. Of note, this appears to be cell type specific, as, for INS-1 832/3 cells, the peak cAMP response was observed at 10 min. For the latter cell type however, the reduced accumulation over time with all agonists was nevertheless measurably less with exendin-phe1 compared to exendin-4. Note that we have also performed further experiments (shown in Fig. 3c-f) to investigate prolonged signaling in beta cells in the context of continuous agonist exposure but with IBMX added only at the very end of the assay to cause cAMP accumulation at a rate indicative of the incident signaling at that particular time-point. The “rank order” of agonists is again the same, but we note that in MIN6B1 cells, cAMP synthesis rate by this point appears reduced below the “basal” level. This might reflect loss of constitutive GLP-1R activity via degradation (see additional experiments in Fig. 2h and Supplementary Fig. 5c) or loss of ability to respond to locally produced GLP-1; we have mentioned this in the manuscript (page 10, paragraph 1). Clearly, there are complex and cell-specific differences in the relative balance of cAMP synthesis and degradation, with and without IBMX, and with acute and prolonged stimulation, which are beyond the remit of this article to investigate in depth.

Fig 3g,h: Although they reach statistical significance the effect of the siRNA on the endocytosis and on the insulin secretion are marginal. Given the variance in the data in Fig 3h, it is even surprising that it a statistical difference could be found. These results should be confirmed using another approach to knock down arrestins; for example, using dominant-negative mutants or arrestin KO cells. Also confirming the roles of endocytosis in the lack of sustained insulin secretion should be tested using other endocytosis inhibitors.

We fully agree with the reviewer that the effects observed in our β -arrestin knockdown experiments were smaller than the agonist-mediated differences. To address this issue, we have performed a number of further experiments to better understand the possible effects of β -arrestin ablation on GLP-1R function: 1) We established dose responses for sustained insulin release in INS-1 832/3 cells (Fig. 4g), which again demonstrated that, after dual β -arrestin knockdown, efficacy for insulin secretion was increased; 2) We increased n for the analogous experiment in MIN6B1 cells (added to Fig. 4h); 3) We established a subline of the human beta cell line EndoC- β H1 with both arrestins depleted by lentiviral-encoded shRNA. These cells displayed increased exendin-4-induced sustained insulin release compared to controls (Fig. 4i); 4) We obtained HEK293 cells with both β -arrestins deleted by CRISPR-Cas9 from our collaborators Dr Hanyaloglu and Prof Inoue and stably expressed SNAP-GLP-1R in these; compared to wild-type cells, we found that potency for cAMP production was increased, along with a progressive increase in efficacy with time (Supplementary Fig. 7i-k). We also found a delay in internalization in keeping with our earlier results, but, interestingly, this effect on GLP-1R endocytosis remained only partial despite total β -arrestin deletion, with complete internalization seen after incubation with agonist for 60 min. Thus, these results appear to indicate a role, but not an absolute requirement, for β -arrestins in mediating GLP-1R endocytosis. A further possibility is that alternative endocytosis pathways are up-regulated. We address these further experiments and issues in the text (page 12, paragraph 1; and in the discussion – page 18, paragraph 2).

Given the signaling responses seen in β -arrestin-less *versus* wild type HEK293 cells here, we suspect that the actions of β -arrestins on GLP-1R function are not limited to trafficking, with a likely contribution from desensitization as well. This might partly explain why exendin-asp3-induced insulin release is significantly blunted despite quite small differences in net cell surface GLP-1R loss, as the β -arrestin-bias of this peptide may result in greater desensitization irrespective of trafficking differences. We hope it is clear that whilst we found a strong correlation between agonist β -arrestin recruitment and internalization propensity, we consider it likely that the overall agonist trafficking and insulinotropic phenotypes are multifactorial.

Regarding the use of endocytosis inhibitors – we are inclined to agree with the reviewer that endocytosis in itself plays an important role in initiating the recycling and resensitization process, and, without this mechanism, receptors trapped in the plasma membrane will become desensitized by β -arrestin recruitment. We now emphasize this in the Introduction (page 5, paragraph 2), and Discussion (page 17, paragraph 2). Whether this applies to the same extent with agonists such as exendin-phe1, which recruits little β -arrestin, is not clear. We consider the use of chemical inhibitors of endocytosis to assess the effect on insulin secretion a non-ideal approach as these molecules are either toxic or non-specific over longer periods²⁴, and furthermore may interfere with insulin granule exocytosis²⁵. We have, however, performed experiments to measure acute signaling responses both in the presence of “metabolic inhibitors”¹, used to restrain the receptor at the plasma membrane for our surface binding assays, and with the dynamin inhibitor Dyngo4a²⁶ (Fig. R2). A significant issue with these approaches, which is frequently ignored in the literature, is that these treatments exert significant effects on cAMP synthesis / degradation processes, as assessed by responses to forskolin. When expressed relative to the forskolin response, we found that exendin-4-induced cAMP response was apparently greater in MIN6B1 and CHO-GLP-1R cells (the latter not for Dyngo4a treatment). Nevertheless, in our view these assays are of limited utility in explaining the enhanced insulinotropic efficacy of GLP-1R agonists identified in our manuscript, and we have therefore not included these figures.

Fig R2. Exendin-4-induced cAMP responses with endocytosis inhibitors. (a) Response to 10 min exendin-4 stimulation in CHO-SNAP-GLP-1R cells with 20 min pretreatment with or without metabolic inhibitors (2-deoxyglucose + sodium azide), $n=4$. (b) As for (a) but with the dynamin inhibitor Dyngo4a, $n=4$. (c) As for (a) but with MIN6B1 cells, in presence of 500 mM IBMX, $n=4$. Results are expressed in all cases relative to response obtained with forskolin (FSK, 10 μ M) with corresponding pretreatments. Error bars indicate SEM.

The observation that persistent cAMP production following endocytosis is observed with ex-4 and ex-asp3 but not ex-phe1 is in contradiction with the main hypothesis. This needs further experiments to rationalize this difference. The explanation of a possible higher number of receptor in CHO-K1 is insufficient.

We concur with the reviewer on this point. Of course, our manuscript pertains mainly to GLP-1R actions in beta cells; we use overexpression systems where the high signals obtained are useful to demonstrate agonist-related differences, especially across dose ranges, and aim to validate findings in beta cells (and *in vivo*) wherever possible. The finding of increased sustained signaling with exendin-4 / exendin-asp3 in CHO-K1 cells is clearly at odds with the situation demonstrated in beta cells.

We have therefore performed further experiments to understand how cell-specific factors might contribute to this discrepancy. In addition to the anticipated differences in GLP-1R expression levels when expressed endogenously *versus* via plasmid vectors designed for high expression levels, we present data below demonstrating how coupling of GLP-1R activation to cAMP accumulation is far greater in CHO-K1 *versus* INS-1 832/3 and MIN6B1 cells. Specifically, in the absence of the phosphodiesterase inhibitor IBMX, exendin-4 induces marked cAMP accumulation in CHO-SNAP-GLP-1R cells, but no increase is apparent in either of the beta cell lines, in which IBMX at moderate-high concentrations is required (Fig. R3a-c). Relative fold changes are shown in Fig. R3d and emphasize the magnitude of the difference. These data indicate that, in CHO-K1 cells, the balance of cAMP synthesis *versus* degradation is heavily in favor of the former, in comparison to the situation in beta cells. This cell-specific factor, in our view, clearly indicates that prolonged cAMP generation in the CHO-K1 cell line is of limited utility for aiding the understanding of signaling responses in beta cells exposed to agonists for long periods. This of course does not invalidate use of CHO-K1 cells to measure agonist-related differences in cAMP potency over acute stimulations, e.g. for bias calculations. As a result of these new data, we have elected to remove the “washout” experiment graph for sake of clarity of message.

Fig R3. Cyclic AMP responses in CHO-K1 versus beta cells. CHO-SNAP-GLP-1R (a), INS-1 832/3 (b), or MIN6B1 (c) cells stimulated with exendin-4 (ex4) in absence or presence of indicated concentration of IBMX for 10 min. (d) E_{max} values from (a)-(c) expressed as relative change from baseline for each IBMX concentration. Error bars indicate SEM.

The experiment with BETP needs to be better explained. It is not evident for this reviewer how to interpret these data.

We note the comment of the reviewer. These data are now shown in Fig. 5h-n. We have aimed to improve the description of the rationale for this particular study (page 13, paragraph 2) to highlight how we wished specifically to use BETP as an independent means to modulate agonist binding kinetics to test our hypothesis that the latter was an important factor influencing recycling rate. Indeed, we found that BETP slowed down the dissociation of exendin-4 from the GLP-1R, and in keeping with our hypothesis, reduced the rate of recycling. We present the cAMP and β -arrestin data to make the point that these did not differ in the presence of BETP; thus, changes specifically to these signaling pathways are unlikely to be responsible for the effect of BETP on insulin release.

Fig 6: The authors state that differential effects of the clinically available compounds are modest. This is certainly not the case for the binding kinetics where two of the compounds show significant difference. Some of them also show difference in internalization and recycling values reported.

Significant differences are indeed found, for example the residence times of Lixisenatide and Liraglutide are longer than that of exendin-4. In reply to the point made by the reviewer about comparing exendin-phe1 with these other compounds, we have shown this comparison now in Supplementary Fig. 9. From these, it is apparent that, for example, the difference in residence time between exendin-4 and ex-phe1 is ~10 fold compared to up to ~2 fold for the other GLP-1R agonists. We have adjusted the text (page 14, paragraph 2) to make this point.

Although Ex-phe1 is indeed the most efficacious (not potent as written by the authors; potency refers to EC50 not maximal response), the difference with 3 of the drugs is not that marked. This needs to be discussed in relation with their internalization and binding properties.

We agree. This has now been highlighted (page 14, paragraph 2) and we have replaced potent with efficacious.

Fig. 7g and h: The authors conclude that there is a difference between ex4 and ex-phe1 for the chronic treatment (16 h) and not for the acute treatment. However, this is only due to a difference in the variability for the acute response and most likely only one experiment explains why the difference between ex4 and ex-phe1 does not reach statistical difference in the acute case.

The same point was made by reviewer 1, and we copy our response here: Regarding the insulin responses, the reviewer is absolutely correct that the statistical significance of exendin-4 vs. exendin-phe1 acute stimulation is influenced by one particular islet batch. If this “outlier” is removed, a significant difference emerges between the two treatments (favoring exendin-phe1). In the absence of clear reason to exclude this particular experiment, we prefer to leave it in and have added a comment in the text highlighting the apparent trend (page 15, paragraph 1). As cell-type related differences in kinetics of differential insulin secretion are expected (for example, Fig. 1h,i), we suggest this observation does not invalidate our overall conclusions, and if anything increases the potential clinical utility of exendin-phe1.

Fig 8: It is surprising that both ex4 and ex-phe1 promote a very dramatic reduction in cumulative food intake over a 6 h time period in Fig 8 but that no such decrease is observed in chronic treatment for many days in Fig 9. Yet there is a reduction in body weight in these chronic treatments even for the vehicle. These inconsistencies need to be explained and controlled for.

The reviewer again makes an important point. These observations are related to dose and mode of administration. In Fig. 8, agonists were administered by intraperitoneal injection, which results in a rapid peak drug level, quickly resulting in appetite inhibition. The dose administered in Fig. 8e was 2.4 nmol/kg. At a lower dose of 0.24 nmol/kg (Supplementary Fig. 10e), food intake was still reduced but to a lesser extent. During the chronic treatment study presented in Fig. 10, 0.24 nmol/kg of each agonist were administered, but, rather than by single injections, this was achieved over a 24 h period via continuous infusion. In comparison to the 8 h experiment in Fig. 8e and Supplementary Fig. 10e, the dose (per unit time) is effectively lower, and the steady pharmacokinetic profile obtained by this mode of administration is very different from the wide fluctuations seen with single subcutaneous injections. The lack of food intake reduction in Fig. 10f is likely due to these factors. The reviewer points out the body weight loss is seen not only in agonist-treated but also control group in Fig. 10f. This is a common finding when osmotic minipumps are surgically implanted and reflects non-specific stress²⁷; we have added a comment in the text (page 16, paragraph 2).

The authors used a conditional taste aversion test as a surrogate for nausea. They conclude that ex-phe 1 did not show evidence of nausea. Yet no difference was found with ex-4. How can this be interpreted in relation to the different properties of ex-4 and ex-phe1 regarding internalization? The other clinically used ligands should have been used for comparative purposes in this assay.

We concur with the point made by the reviewer. In the conditioned taste aversion experiment, the positive control (LiCl) showed clear evidence of aversive behavior (reduced preference for grape Kool-Aid). However, there was no apparent effect from exendin-4 itself. Whilst we were careful to state that

the experiment found no evidence of increased nausea with exendin-phe1, it does not by itself exclude this possibility. Unfortunately, measuring nausea in rodents with GLP-1R agonists is problematic. Whilst we are aware of one study in which conditioned taste aversion was seen in mice treated with Liraglutide²⁸, others have found lack of an aversive response to peripheral exendin-4 in mice²⁹. Note that several studies using CTA to assess GLP-1R-induced nausea in mice use central administration of agonist directly into the brain. On balance, having performed the experiment, we decided to include the result in the manuscript, as it provides limited evidence against an increased nauseating effect of exendin-phe1, particularly in conjunction with the lack of difference between the two peptides for acute reductions in food intake. We have however adjusted the text to emphasize the caveats with this study (page 16, paragraph 2). Regarding repeating the experiment with the other GLP-1R agonists, our animal facility indicated that, given the lack of a clear effect with exendin-4, it would not be ethical to perform further experiments using this methodology with little chance of a positive result.

Reviewer #3 (Remarks to the Author):

These elegant and extremely thorough studies by Jones and colleagues potentially establish a role for GPCR-agonist interactions that specifically modulate receptor kinetics (internalization, recycling and trafficking) with phenotypic outcomes. Specifically, the authors have identified mutants that modify the dynamics of the GLP-1 receptor (GLP-1R) and suggest that GLP-1R agonists that increase the availability of GLP-1R at the cell surface have improved insulinotropic and glucose-lowering properties. This adds to the growing interest of the phenotypic effects of “biased signaling” via GPCRs but adds a novel aspect of assessing the effects not on signaling events but on receptor dynamics. The studies utilize a variety of novel and state-of-the art molecular techniques to thoroughly assess the effect of mutants of the GLP-1 R agonist exendin-4 (ex4) on various aspects of GLP-1R dynamics (internalization, recycling, residence time of the agonists on the receptor both extra- and intracellularly, etc.), and test the efficacy of these mutants on in vivo glucose control and both cell-based and in vivo insulin secretion. The overarching conclusion is that agonists that promote extended cell surface expression of the GLP-1R may provide a novel therapeutic strategy for improved GLP-1 based drugs. This is a significant finding that merits further exploration. Having said this, there are several issues that need to be addressed. In general, the two main issues revolve around: 1) the feasibility of correlating molecular observations made in a CHO line over-expressing the GLP-1R with a phenotypic effect in beta cell lines expressing endogenous GLP-1R levels (this is partially addressed in some, but not all, experiments), and 2) correlating effects of acute ex4/modified ex4 peptide exposure on GLP-1R internalization/trafficking on differences chronic insulin secretion, especially since acute insulin secretion appears to be unaffected by the various ex4 peptides.

We thank the reviewer for these very kind and insightful comments. We deal with the specific points raised below.

Major points:

1. There is a significant issue with regards to the kinetics of the various measurements. The studies begin with a correlation between acute measurements (GLP-1R internalization, cAMP production) and a “chronic” insulin secretion index. The internalization and cAMP experiments were conducted following a 90 min exposure to the various GLP-1R agonists, yet the insulin secretion index that showed a different effect of the various agonists was determined following a 16h exposure of the agonists. On the surface, this would not seem like an issue, except for the fact that the authors found that acute exposure of the agonists (60 min – similar to the time frame of the internalization experiments) led to no difference in the insulin secretion index induced by ex-phe1 (low GLP-1R internalizer) and ex-asp3 (high GLP-1R internalizer). Therefore, if the authors had performed a correlation analysis of the acute GLP-1R

internalization capacity vs. the acute insulin secretion index of the various ex4 molecules, I presume that they would have found either no correlation or a positive correlation. One experiment that could address this issue is to measure chronic GLP-1R internalization and correlate that to chronic insulin secretion, and preferably in the same cell line (see points #1 and #2 below).

The referee raises a very valid point. We have performed further experiments to address this issue. Firstly however, we should emphasize that differences in kinetics between signaling outputs (e.g. cAMP) and accumulation of insulin in supernatant during static secretion experiments are to be expected. This of course partly reflects the fact the cAMP production is a proximal event in the signaling chain whereas insulin secretion occurs downstream, meaning that there is a lag between the two events. Further, as detecting changes in insulin secretion in this assay relies on significant accumulation of released insulin, a longer incubation period is needed. In this context, 60 min is considered an acute incubation. We now present insulin secretion data for all exendin-4-derivatives over a shorter 60 min incubation period (Fig. 1 and Supplementary Fig. 1), that, as expected, demonstrated no clear difference in maximal insulin secretion, which appears only at later time-points (see also time-course experiments in Fig. 1h,l, as suggested below by the reviewer). We acknowledge that the small fold changes in insulin secretion seen with acute incubations somewhat preclude accurate identification of relative potency differences, but do in our view establish that efficacy for insulin secretion at this time point is similar for all compounds.

The trafficking differences (internalization and recycling) we measured in our studies were, as stated by the reviewer, performed with relatively short incubation times (15-90 min depending on the assay). However, we believe that these findings are likely to be valid over longer time periods, during which GLP-1Rs may cycle back to the membrane and undergo several rounds of re-stimulation and re-internalization. To support the insulin secretory effects we observed after 16 h, we performed further experiments to measure residual surface GLP-1R (reflecting the net effect of internalization, recycling and degradation) after this prolonged period of time, in INS-1 832/3 and MIN6B1 (for endogenous GLP-1R expression) by immunofluorescence and flow cytometry, as well as in MIN6B1-SNAP-GLP-1R cells. These data are presented in Fig. 2i-k and Supplementary Fig. 5d-h. Overall, these studies revealed greater net preservation of surface receptor with exendin-phe1 compared with exendin-4, and reduced preservation, with respect to the parent compound, with exendin-asp3. We observed that differences with exendin-asp3, as with acute incubations, remained small and indeed were not seen in every assay. We presume that the clearly blunted insulin secretion profile of this agonist depends not just on its trafficking profile but also on other effects – for example β -arrestin-mediated desensitization. This has now been discussed in the text (page 19, paragraph 1). Furthermore, in line with our earlier electron microscopy work showing reduced lysosomal targeting of SNAP-GLP-1R with exendin-phe1 treatment compared to exendin-4, we have now measured SNAP-GLP-1R degradation in MIN6B1-SNAP-GLP-1R and CHO-SNAP-GLP-1R cells, and found that less receptor is degraded with exendin-phe1 than with exendin-4 (Fig. 2h and Supplementary Fig. 5c). Overall, we feel that these new experiments add significance, and support mechanistically our observations of increased insulin secretion over time with exendin-phe1.

2. It appears that some of the observations are cell type-dependent. It is understood that there will be differences between CHO cells over-expressing the GLP-1R and beta cell lines (INS1 and MIN6 cells) that express endogenous levels of the GLP-1R, and to the authors' credit, this is briefly discussed. However, there were also differences between INS1 and MIN 6 cells. For example, when looking at the effects of ex-phe1 vs. ex-asp3 on acute insulin secretion, there were no differences between the peptides in INS1 cells but there were in MIN6 cells (Figs. 1f and 1g). This issue is brought up since different assays were sometimes performed in some cell lines but not all. For example, as indexes of desensitization, Ca^{2+} signaling was done in one cell type and cAMP generation was done in another ((Figs. 21 and 2j). It raises the possibility that perhaps some of the positive observations were not observed in all cell lines. It

would have been more appropriate for consistency to either perform all assays in all three cell lines or just to choose one cell line and run all assays on this one cell line.

We take on board the reviewer's point. Acute (and prolonged) insulin secretion experiments have been repeated for both cell types to cover the full panel of exendin-4 derivatives (see Supplementary Fig. 1). In these independent experiments, exendin-asp3 was no longer more insulinotropic than exendin-4 in MIN6B1 cells at the acute time-point (60 min). This might reflect the fact that some of the original experiments referred to with MIN6B1 cells were performed separately for exendin-phe1 *versus* exendin-asp3, and thus susceptible to passage-related differences in response (in our experience, phenotypic drift with beta cell lines can be a significant issue). With the more recent experiments, all treatments were done in parallel, minimizing potential contributions from differences in cell responsiveness, etc. Nevertheless, we agree that some differences are still apparent in the different beta cell types used in our studies; for example, the improvement in insulin secretion with exendin-phe1 in INS-1 832/3 cells exceeds the effect (whilst still present) in MIN6B1. In a limited number of cases, a difference *versus* exendin-4 was seen with one peptide in one cell line but not the other (for example, net residual surface GLP-1R expression after 16 h was reduced with exendin-asp3 in MIN6B1 but not INS-1 832/3 cells, by confocal microscopy – Supplementary Fig. 5f-h), but in no case did we find examples where the opposite conclusion could be drawn depending on the cell line.

Overall, we believe that presenting data from cell types from two different species (and, indeed, human islet data, as well as newly performed experiments using the human beta cell line EndoC-βH1) reduces the likelihood of identifying a species-specific phenomenon, thereby increasing the potential for future clinical utility, and adds substantially to the robustness of our results. We would emphasize that our most important finding - improved insulin release with exendin-phe1 under conditions of prolonged exposure - was consistently seen across all *in vitro* systems, and also *in vivo*.

Regarding the specific examples raised by the reviewer from Fig. 2i,j (now in Fig. 3): ascertaining the degree of desensitization in CHO-GLP-1R cells by measurement of cAMP was not possible as in this cell type, GLP-1R activation is extremely highly coupled to generation of cAMP (indeed, ongoing cAMP generation, without addition of phosphodiesterase inhibitors, was detectable 24 h after agonist washout; see also Fig. R3 in reply to a query from Reviewer 2). Hence, we deliberately used a less well-coupled pathway (Ca²⁺ release) for this cell type. However, in beta cells endogenously expressing the GLP-1R, cAMP production is less well coupled, and only detectable biochemically in the presence of IBMX, making it a suitable system to measure desensitization - which has the advantage of being the primary signaling intermediate we were interested in for this study. The data from INS-1 cells have been re-plotted relative to the “basal” response to address a query from reviewer 2. We additionally performed further experiments in both INS-1 832/3 and MIN6B1 cells to assess incident cAMP production at the end of a 16 h exposure period by adding IBMX at this point (Fig. 3c,d). This again showed the expected pattern of signaling responses. For a variety of reasons (weaker responses, cells easily washed from the plate), high-throughput Ca²⁺ measurements with beta cells to measure desensitization were not performed. Of course, data could be obtained by live cell imaging, but in our opinion the additional information gained would not considerably add to the findings of the paper, and is somewhat superseded by the measurements taken in human islets, which are clearly of greater relevance.

Again, to the authors' credit, they did generate beta cell lines with reagents that allowed them to look at certain events (e.g. MIN6B1-SNAP-GLP1R cells) in a “native context” and did compare INS1 and MIN6 cells with almost every experiment.

3. Following from points #1 and #2 above, Fig. 1e shows a difference in the 16h insulin secretion capacity between ex-phe1 and ex-asp3 in MIN6 cells. However, what is not shown is whether there is a significant difference in the insulin secretion capacity between native ex4 and ex4-phe1 in MIN6 cells. This is an important point since the authors later show a difference

in GLP-1R internalization between ex4 and ex4-phe1 in MIN6 cells. Therefore, to strengthen the argument that internalization capacity is correlated with insulin secretion capacity, the authors should show differences between ex4 and ex4-phe1 in both internalization and insulin secretion in the same cell line (whether it is MIN6 or INS1).

We agree. Further experiments performed with all exendin-4 derivatives in MIN6B1 (and INS-1 832/3) cells now confirm that exendin-phe1, and similar compounds, achieve greater insulin release than exendin-4 over prolonged periods in MIN6B1 cells (see Supplementary Fig. 1c), in line with the main hypothesis of our paper. This difference was highly reproducible and statistically significant, albeit smaller than the effect in INS-1 832/3 cells. Corroborating the short term internalization (and recycling) effects observed in MIN6B1 cells expressing the SNAP-GLP-1R, we performed further experiments at the 16 h time-point to measure residual surface expression of SNAP-GLP-1R in MIN6B1 cells after exposure to exendin-4, exendin-phe1 and exendin-asp3 (Fig. 2i-k). We also performed similar experiments in wild-type INS-1 832/3 and MIN6B1 cells with endogenous levels of expression using immunofluorescence detection of surface antibody by confocal microscopy and flow cytometry (Supplementary Fig. 5d-h). Note that, rather than subjective selection of apparent plasma membrane regions, we quantified mean fluorescence of entire cell areas after background subtraction; this method inevitably includes contributions from cytoplasmic fluorescence, which means that absolute levels of receptor cannot be inferred (indeed, numerically the results would suggest high levels of residual surface receptor expression, out of keeping with our main hypothesis), but different treatments can still be compared. We found here that the trafficking differences of exendin-phe1 *versus* exendin-4, when allowed to continue for longer time periods, do indeed result in relative preservation of cell surface receptor.

4. The design of the “acute” and “chronic” experiments measuring insulin secretion does not allow for a truly fair comparison on potential effects of acute vs. chronic peptide exposure on insulin secretion. As described, the acute experiments began with beta cell lines seeded for 24 h in low glucose (3 mM) and then treated with high glucose media (11 mM) with/without agonists for 1h. For the chronic experiments, the cells were seeded in high glucose media and were exposed to agonists overnight (16 h). Therefore, the cells used for the chronic exposure were not exposed to the same low glucose stress as the cells in the acute experiments. To make a true comparison of both conditions, the chronic experiments should be conducted in cells exposed to low glucose for 24 h and then switched to high glucose +/- peptides overnight. Alternatively, the cells in the acute experiments should be exposed directly to high glucose media.

All insulin secretion experiments were performed with an overnight incubation in low glucose medium prior to agonist stimulation. We apologize that this point was omitted from the Methods section due to the word limit, but it is explicitly stated now (page 27).

5. With regards to the chronic insulin secretion experiments, it is not clear whether the elevated insulin levels were due to a persistently higher secretion of insulin or whether most of the insulin secretion occurred early during the peptide exposure period and the insulin lingered (not quite sure what the half-life is of insulin in cell culture conditions). Perhaps a better approach would be to look at a time course of insulin levels throughout the chronic exposure and not just at the 16 h time point.

We performed additional experiments to address this question (Fig. 1h,i). The time course indicates that in both cell types, extra accumulation of insulin with exendin-phe1 is detectable at around 6 h, but is more marked after 16 h. Interestingly, this approximates the time course for improved glucose tolerance seen *in vivo*, with a detectable difference when IPGTT was performed at 4 h with exendin-phe1 *versus* exendin-4, but a greater effect at 8 h (Fig. 8a). For exendin-asp3, the difference is apparent earlier in INS-1 832/3 cells, but not in MIN6B1.

6. Many of the experiments were conducted with very small n's (2-3) and it is difficult to see how statistical significance was achieved with such low numbers of replicates.

Additional replicates have been performed to ensure statistical comparisons have only been made with at least three biological replicates, with the vast majority including four or more.

Minor points:

1. The authors state in Supplementary Fig. 2b,c that there is a similar rank order of ex4 peptides with regards to receptor internalization and protection from ER stress or glucolipototoxicity. This does not appear to be the case since there appears to be a difference between ex4 and ex4-phe1 with regards to receptor internalization (in CHO and MIN6 cells) but not with regards to ER stress in INS1 cells or glucolipototoxicity in MIN6 cells. Again, highlighting points made above, why were different cell lines used for ER stress vs. glucolipototoxicity experiments? Is it that the effects of the various peptides were cell type-dependent? – e.g. was ex4-phe1 detrimental towards the response to ER stress in MIN6 cells?

We have reworded the manuscript to highlight how exendin-phe1 failed to reduce apoptosis compared to exendin-4 in either cell line (page 7, paragraph 2; this point was also raised by reviewer 2). Of course, the loss of anti-apoptotic effect of exendin-asp3 is congruent with this compound's reduced insulinotropic effect, which remains interesting as apoptosis reduction in beta cells was previously identified as a major downstream consequence of beta cell β -arrestin signaling²³. Yet, in our study, a peptide biased towards β -arrestin recruitment (exendin-asp3) was ineffective for this readout. Regarding the use of different assays for different cell types: from previous work, in our hands, GLP-1R agonists fail to substantially protect INS-1 832/3 against glucolipototoxicity, limiting the utility of this assay to identify differences between agonists; hence, we routinely used ER stress to induce apoptosis in this cell type as previously reported with the parental INS-1E line³⁰. However, glucolipototoxicity responds better to GLP-1R agonist treatments in MIN6 cells, and we prefer to use this modality where possible as it better reflects the diabetic milieu. As the effect on apoptosis is not the main focus of our investigation, to ensure timely completion of revisions, we did not perform repeat assays using different pro-apoptotic stimuli in each cell type.

2. The next to last paragraph on Page 8 (“GPCR recycling restores...”) described data focusing on receptor responsiveness and desensitization, yet the last line in the paragraph details data (Supplementary Fig. 6) that do not truly support any issues of receptor desensitization and even highlight many of the major issues discussed above. Sustained cAMP generation in the presence of ex4-phe1 vs. ex4 is not necessarily indicative of any issues of receptor sensitization. Furthermore, if ex4-phe1 promotes more sustained cAMP generation compared to ex4 in INS1 cells (Supplementary Fig. 6a), then could this not be a factor in the greater chronic insulin secretion in INS1 cells (Fig. 1d and Supplementary Fig. 1b)? This would contradict the authors' initial assertion that “Surprisingly, cAMP response was poorly predictive of prolonged insulin release” (Page 6). This again highlights the importance of correlating acute molecular events with acute phenotypes and chronic molecular events with chronic phenotypes (Major point #1).

We fully agree with the reviewer that sustained cAMP generation with exendin-phe1 compared to exendin-4 is likely to be a significant contributor to greater cumulative insulin secretion over time with the former peptide. We also concede the reviewer's point about our original comment about cAMP response being poorly predictive of prolonged insulin release. We meant to imply here that the acute cAMP experiments in CHO-GLP-1R cells provided potency ratios which would seemingly not register exendin-phe1 and related compounds as promising insulin secretagogues, and have changed the manuscript accordingly to clarify this (page 7, paragraph 1). However, we maintain that receptor

resensitization, via faster recycling with exendin-phe1, is a likely contributor to sustained insulin secretion over time (although not conclusively demonstrated). We submit that accumulation of cAMP with prolonged exposure reflects a balance between rate of production and rate of degradation. In this assay, IBMX was used to inhibit cAMP degradation, although we cannot exclude agonist-related differences in degradation of this signaling molecule (one intriguing possibility could be differential recruitment of phosphodiesterases to the adenylate cyclase-PKA-AKAP complexes dependent on differential β -arrestin recruitment; it is out-with this study to investigate this possibility in detail). The rate of cAMP synthesis likely depends on the inherent ability of the agonist to induce G protein activation once bound, and the number of receptors bound by agonist. Similar efficacy (E_{max}) for cAMP production after 10 min agonist exposure in “naïve” INS-1 832/3 cells (now in Fig. 3a) suggests each compound is a full agonist for the pathway when amount of receptor is not limiting. It seems a reasonable possibility then that agonist-variable depletion of receptor number with longer incubations is at least partly responsible for differential cAMP accumulation.

To further understand this, we performed experiments in which cells were stimulated for prolonged periods with agonist but no IBMX (no detectable cAMP accumulation), but then IBMX was added for a final 10 min to induce rapid cAMP accumulation at a rate which we suggest is predominantly indicative of rate of cAMP synthesis (Fig. 3c,d). Here, E_{max} for exendin-phe1 was again greater than for exendin-4; we suggest that this finding supports our suggestion that exendin-phe1 retains the ability to continually re-stimulate the beta cell over long periods. We also investigated whether agonist-related differences might exist in down-regulation of post-receptor signaling; we consider this possibility unlikely as incremental cAMP responses to forskolin in cells treated with different agonists for prolonged periods were similar (Fig. 3e,f).

We would like to thank again each of the referees and the Editorial team for their helpful comments, which we believe have substantially improved our manuscript.

References

1. Widmann, C., Dolci, W. & Thorens, B. Agonist-induced internalization and recycling of the glucagon-like peptide-1 receptor in transfected fibroblasts and in insulinomas. *Biochem J* **310** (Pt 1), 203–214 (1995).
2. Gao, W. & Jusko, W. J. Pharmacokinetic and pharmacodynamic modeling of exendin-4 in type 2 diabetic Goto-Kakizaki rats. *J Pharmacol Exp Ther* **336**, 881–890 (2011).
3. Chen, T., Mager, D. E. & Kagan, L. Interspecies modeling and prediction of human exenatide pharmacokinetics. *Pharm. Res.* **30**, 751–760 (2013).
4. Drucker, D. J., Philippe, J., Mojsov, S., Chick, W. L. & Habener, J. F. Glucagon-like peptide I stimulates insulin gene expression and increases cyclic AMP levels in a rat islet cell line. *Proc Natl Acad Sci USA* **84**, 3434–3438 (1987).
5. Lester, L. B., Langeberg, L. K. & Scott, J. D. Anchoring of protein kinase A facilitates hormone-mediated insulin secretion. *Proc Natl Acad Sci USA* **94**, 14942–14947 (1997).
6. Kang, G., Chepurny, O. G. & Holz, G. G. cAMP-regulated guanine nucleotide exchange factor II (Epac2) mediates Ca²⁺-induced Ca²⁺ release in INS-1 pancreatic beta-cells. *J Physiol (Lond)* **536**, 375–385 (2001).
7. Kang, G. *et al.* Epac-selective cAMP analog 8-pCPT-2'-O-Me-cAMP as a stimulus for Ca²⁺-induced Ca²⁺ release and exocytosis in pancreatic beta-cells. *Journal of Biological Chemistry* **278**, 8279–8285 (2003).
8. Roger, B. *et al.* Adenylyl cyclase 8 is central to glucagon-like peptide 1 signalling and effects of chronically elevated glucose in rat and human pancreatic beta cells. *Diabetologia* **54**, 390–402 (2011).
9. Nuber, S. *et al.* β -Arrestin biosensors reveal a rapid, receptor-dependent activation/deactivation cycle. *Nature* **531**, 661–664 (2016).
10. Sungkaworn, T. *et al.* Single-molecule imaging reveals receptor-G protein interactions at cell

- surface hot spots. *Nature* **550**, 543–547 (2017).
11. Goulding, J., May, L. T. & Hill, S. J. Characterisation of endogenous A2A and A2B receptor-mediated cyclic AMP responses in HEK 293 cells using the GloSensor™ biosensor: Evidence for an allosteric mechanism of action for the A2B-selective antagonist PSB 603. *Biochem. Pharmacol.* (2017). doi:10.1016/j.bcp.2017.10.013
 12. Al-Sabah, S. *et al.* The GIP receptor displays higher basal activity than the GLP-1 receptor but does not recruit GRK2 or arrestin3 effectively. *PLoS ONE* **9**, e106890 (2014).
 13. Curtis, M. J. *et al.* Experimental design and analysis and their reporting: new guidance for publication in BJP. *Br J Pharmacol* **172**, 3461–3471 (2015).
 14. Jorgensen, R., Martini, L., Schwartz, T. W. & Elling, C. E. Characterization of glucagon-like peptide-1 receptor beta-arrestin 2 interaction: a high-affinity receptor phenotype. *Mol. Endocrinol.* **19**, 812–823 (2005).
 15. Jorgensen, R., Kubale, V., Vrecl, M., Schwartz, T. W. & Elling, C. E. Oxyntomodulin differentially affects glucagon-like peptide-1 receptor beta-arrestin recruitment and signaling through Galpha(s). *J Pharmacol Exp Ther* **322**, 148–154 (2007).
 16. Wootten, D. *et al.* Allosteric modulation of endogenous metabolites as an avenue for drug discovery. *Mol. Pharmacol.* **82**, 281–290 (2012).
 17. Wootten, D. *et al.* The Extracellular Surface of the GLP-1 Receptor Is a Molecular Trigger for Biased Agonism. *Cell* **165**, 1632–1643 (2016).
 18. Jones, B. J. *et al.* Potent Prearranged Positive Allosteric Modulators of the Glucagon-like Peptide-1 Receptor. *ChemistryOpen* doi:10.1002/open.201700062
 19. Kuna, R. S. *et al.* Glucagon-like peptide-1 receptor-mediated endosomal cAMP generation promotes glucose-stimulated insulin secretion in pancreatic β -cells. *Am J Physiol Endocrinol Metab* **305**, E161–70 (2013).
 20. Bueno, A. B. *et al.* Positive Allosteric Modulation of the Glucagon-like Peptide-1 Receptor by Diverse Electrophiles. *J Biol Chem* (2016). doi:10.1074/jbc.M115.696039
 21. Klein Herenbrink, C. *et al.* The role of kinetic context in apparent biased agonism at GPCRs. *Nat Commun* **7**, 10842 (2016).
 22. Yu, S. S., Lefkowitz, R. J. & Hausdorff, W. P. Beta-adrenergic receptor sequestration. A potential mechanism of receptor resensitization. *Journal of Biological Chemistry* **268**, 337–341 (1993).
 23. Quoyer, J. *et al.* GLP-1 mediates antiapoptotic effect by phosphorylating Bad through a beta-arrestin 1-mediated ERK1/2 activation in pancreatic beta-cells. *J Biol Chem* **285**, 1989–2002 (2010).
 24. Park, R. J. *et al.* Dynamin triple knockout cells reveal off target effects of commonly used dynamin inhibitors. *J. Cell. Sci.* **126**, 5305–5312 (2013).
 25. Min, L. *et al.* Dynamin is functionally coupled to insulin granule exocytosis. *Journal of Biological Chemistry* **282**, 33530–33536 (2007).
 26. McCluskey, A. *et al.* Building a better dynasore: the dyngo compounds potently inhibit dynamin and endocytosis. *Traffic* **14**, 1272–1289 (2013).
 27. Freeman, J. N., do Carmo, J. M., Adi, A. H. & da Silva, A. A. Chronic central ghrelin infusion reduces blood pressure and heart rate despite increasing appetite and promoting weight gain in normotensive and hypertensive rats. *Peptides* **42**, 35–42 (2013).
 28. Sisley, S. *et al.* Neuronal GLP1R mediates liraglutide's anorectic but not glucose-lowering effect. *J Clin Invest* **124**, 2456–2463 (2014).
 29. Talsania, T., Anini, Y., Siu, S., Drucker, D. J. & Brubaker, P. L. Peripheral exendin-4 and peptide YY(3-36) synergistically reduce food intake through different mechanisms in mice. *Endocrinology* **146**, 3748–3756 (2005).
 30. Yusta, B. *et al.* GLP-1 receptor activation improves beta cell function and survival following induction of endoplasmic reticulum stress. *Cell Metab* **4**, 391–406 (2006).

Reviewers' comments:

Reviewer #1 (Remarks to the Author):

The manuscript has been improved and I am satisfied that my major concerns have been adequately addressed.

Reviewer #2 (Remarks to the Author):

The authors should be commended for adding a significant amount of additional data and clarifications in the text to address the issues raised by the reviewers. However, there remains a number of issues that makes it difficult to accept the conclusions at face value. In particular, although it is true that cell-type specific differences are to be expected, these differences nevertheless question some of the conclusions reached. Also, the authors acknowledge some of the discrepancies between data-sets and assay types but argue that these differences can be explained and do not change the conclusions. Although, it is true that different assays have different caveats and may lead to different results, it is not always clear for this reviewer why one set of results that is consistent with the hypothesis would be superior to the others. As one example of selective conclusions, the authors argue on the one hand that the aversion test is not a very good assay and cannot distinguish between GLP1 agonists but yet still show this data in the manuscript and conclude against an increased nauseating effect of exendin-phe1. Yet there was no effect either for exendin-4, thus preventing any comparison between the two compounds. Nevertheless, in the abstract the authors conclude that compounds with specific GLP-1R trafficking profiles have greater tolerability. This conclusion is not supported by the data. The authors rightfully indicate that the regulation of insulin secretion is multifactorial, and overall the rebuttal clearly support this, yet in the manuscript, the authors are trying to make the point that the lack of endocytosis and increased recycling resulting in longer cell surface residency time for the receptor is determinant for the insulinotropic action of the analogues. Although many of the assays support this, the complications of some differences obtained in the different cell types (ex the marginal difference in insulin secretion observed between the compounds in MIN6B1 cells) as well as the difficult comparison between acute and chronic effects still challenges the conclusion. Also, the link between cell surface residency and insulin secretion is challenged but the fact compounds leading to internalization varying between a few percent to 75% of maximum yielded identical insulin secretion (Fig 1). Finally, in many instances providing the raw data rather than or in addition to normalized data would allow a better estimate of the robustness of the assays and reproducibility of the data.

Minor:

In sup fig 1, the LogEC50 should be (M) and not (nM)

In Fig 1 it is not clear why statistical differences are indicated in panel h and i and not f and g.

Reviewer #3 (Remarks to the Author):

The revised manuscript has adequately addressed the points raised in the original review. There are minor, primarily aesthetic points that need to be addressed:

1. Figure 3b and 3d. It is not clear why these data in MIN6B1 cells are not presented as CRCs as they are for INS1 cells in Figs. 3a and 3c. The figure legend states that the data in (b) and (d) are presented as in (a) and (c), respectively, but this is not the case. It is not clear what the concentration of agonists were used in Figs. 3b and 3d.

2. The data in Supplementary Figure 9 that include how ex-phe1 compare to the rest of the agonists should really be shown in the main text.

Reviewers' comments:

Reviewer #1 (Remarks to the Author):

The manuscript has been improved and I am satisfied that my major concerns have been adequately addressed.

We thank the reviewer for their positive comment, and were delighted to hear that (s)he believed that our revised manuscript was suitable for publication.

Reviewer #2 (Remarks to the Author):

The authors should be commended for adding a significant amount of additional data and clarifications in the text to address the issues raised by the reviewers. However, there remain a number of issues that makes it difficult to accept the conclusions at face value. In particular, although it is true that cell-type specific differences are to be expected, these differences nevertheless question some of the conclusions reached. Also, the authors acknowledge some of the discrepancies between data sets and assay types but argue that these differences can be explained and do not change the conclusions. Although, it is true that different assays have different caveats and may lead to different results, it is not always clear for this reviewer why one set of results that is consistent with the hypothesis would be superior to the others. As one example of selective conclusions, the authors argue on the one hand that the aversion test is not a very good assay and cannot distinguish between GLP1 agonists but yet still show this data in the manuscript and conclude against an increased nauseating effect of exendin-phe1. Yet there was no effect either for exendin-4, thus preventing any comparison between the two compounds. Nevertheless, in the abstract the authors conclude that compounds with specific GLP-1R trafficking profiles have greater tolerability. This conclusion is not supported by the data.

We recognise the reviewer's concerns regarding the taste aversion test. To address this, we have now, additionally, performed behavioural satiety studies¹⁻³ in mice to identify behaviours typically linked to nausea. These new results support and reinforce our previous findings from the aversion tests.

Thus, we performed separate studies at the two agonist doses used elsewhere in our manuscript (0.24 nmol/kg and 2.4 nmol/kg), with the observer blinded to treatment allocation. Results are now presented in Fig. 8f and Supplementary Fig. 10c and 10h,i, and reveal differences between exendin-4 and exendin-phe1. In particular, at the higher dose, increased pica (i.e. consumption of non-nutritive material such as bedding, indicative of nausea⁴⁻⁶) was observed with exendin-4, compared to exendin-phe1 at the same dose. Loss of locomotor activity, as well as early loss of feeding behaviour, was also observed with exendin-4. At the lower dose, no difference in pica behaviour between the two treatments was apparent. However, the anti-hyperglycaemic effects of exendin-phe1 at 0.24 nmol/kg (Supplementary Fig. 10e,f) exceed those of exendin-4 at 2.4 nmol/kg (Figure 8a,b). Therefore, in our view, these results are indicative of an improved tolerability profile of exendin-phe1, on the basis that greater glucose lowering can be achieved for the same degree of nausea, or alternatively, a lower dose could be used to achieve similar anti-hyperglycaemic efficacy with reduced nausea.

The authors rightfully indicate that the regulation of insulin secretion is multifactorial, and overall the rebuttal clearly support this, yet in the manuscript, the authors are trying to make the point that the lack of endocytosis and increased recycling resulting in longer cell surface residency time for the receptor is determinant for the insulinotropic action of the analogues. Although many of the assays support this, the complications of some differences obtained in the different cell types (ex the marginal difference in insulin secretion observed between

the compounds in MIN6B1 cells) as well as the difficult comparison between acute and chronic effects still challenges the conclusion.

The reviewer is presumably making the specific point that, in MIN6B1 cells, the agonist-related differences in prolonged insulin secretion are of smaller magnitude to those in INS-1 832/3 cells (Figure 1a,b). This is of course correct, but we would like to emphasise the following: Firstly, the differences in the response to different agonists, although smaller in the MIN6B1 cell line, are still highly reproducible and statistically significant (Supplementary Figure 1c). We also note that MIN6B1 cells are less responsive to GLP-1R agonist stimulation in this assay than INS-1 832/3 cells (e.g. insulin stimulation index [ISI] for exendin-4 = 1.8 in MIN6B1 at 100 nM, versus $E_{max} = 2.8$ in INS-1 832/3, data from Supplementary Figure 1c; also see Figure R3 from our previous rebuttal indicating acute exendin-4-induced cAMP responses in both cell types). Therefore, graphically, a greater proportion of the figure is accounted for by “basal” insulin release (response to glucose alone; ISI 0 – 1), contributing to the appearance of smaller agonist-related differences. Subtracting 1 from the ISI to illustrate the insulin secretion *directly attributable to the agonist* (“incretin effect”), INS-1 832/3 remains the more responsive cell type, but the ~2.5-fold differences in insulin secretion between agonists with MIN6B1 cells become more apparent. We indicate this in Figure Rb1 below, showing also that, as expected, insulin secretion differences are highly correlated between cell lines. By convention, we retain the “stimulation index”, i.e. fold change in insulin secretion compared to glucose alone, in the main manuscript.

Figure Rb1. Overnight (16 h) insulin secretion measured in MIN6B1 and INS-1 832/3 cells. Data, derived from Figures 1a,b, are replotted as the incretin effect, i.e. ISI minus 1, thereby indicating the agonist-specific increase in insulin secretion. Relationship quantified by linear regression.

Overall, we consider the broadly similar results obtained with cell lines from two separate species a strength of our manuscript. We also wish to re-emphasise the marked *in vivo* differences obtained during glucose tolerance testing, which support our *in vitro* findings.

Also, the link between cell surface residency and insulin secretion is challenged by the fact compounds leading to internalization varying between a few percent to 75% of maximum yielded identical insulin secretion (Fig 1).

This is indeed a pertinent point. Compounds with the greatest degree of cell surface receptor loss (such as exendin-asp3) are the least insulinotropic, with an approximately linear (negative) relationship between internalisation and insulin release up to some point between 50 and 75% cell surface loss, below which the effect flattens off, and further reductions in cell surface GLP-1R loss do not translate to increases in maximal insulin secretion (Figure 1a). Of course, in our view the exciting discovery here is that, contrary to the standard practice in drug discovery of selecting high affinity / high potency agonists for further development, greater insulinotropic efficacy can be achieved by opting for *lower* potency compounds, an effect we believe relates substantially to their trafficking properties once bound to the GLP-1 receptor. Nevertheless, the reviewer understandably queries why even larger effects on insulin secretion do not result when cell surface loss is reduced

even further (for example, exendin-dTyr1 E_{\max} for internalisation was measured at 7%, but is not more insulinotropic than exendin-phe1 (27%) (data indicated in Supplementary Figure 1)).

This observation can be at least partly explained by the fact that the investigational agonists here do not differ only in their trafficking properties but also in other signaling effects. In particular, it should be noted that compounds that lead to the greatest preservation of GLP-1R surface residence (via reduced internalisation, increased recycling, or both) are also less potent agonists (Supplementary Figure 1b). Whilst all are full agonists for cAMP in over-expressing CHO-K1 cells, it is well established in the pharmacological literature⁷⁻⁹ that, when receptor density is limited, compounds which were thought to be weak full agonists in fact lose efficacy and exhibit partial agonist behaviour.

We consider this relevant to our studies with biased agonists in beta cells with endogenous levels of GLP-1R expression, where the overall signaling response at any point during a prolonged incubation with a “weak” agonist will be determined by, in simplified terms, 1) the number of available receptors, and 2) the inherent ability of the agonist to bind and activate these receptors. Thus, when considering a very weak agonist, such as exendin-dTyr1, versus a moderately weak agonist, such as exendin-phe1: the former might preserve the greatest number of surface receptors but be less able to generate a full response from them, while the latter might preserve a more moderate number of receptors but possesses a greater ability to activate them. Accordingly, it can be envisaged how a balance between these two competing factors could result in both compounds generating the same overall response.

To address this directly, we have now performed additional experiments. Firstly, we measured cAMP responses in INS-1 832/3 cells exposed to each compound for 16 h, with the phosphodiesterase inhibitor IBMX added only right at the end of the incubation, to provide a point estimate of cAMP synthesis (as already performed for exendin-4, exendin-phe1 and exendin-asp3 in our previous revision – Fig. 3c – and now extended to all compounds; full dose response data presented below in Figure Rb2a). A similar pattern to that observed with insulin release was observed; namely, compounds with greatest cell surface receptor loss exhibited the lowest E_{\max} for cAMP, but a flattening off of the relationship below ~50% internalisation was again observed (Figure Rb2b). Thus, the insulin secretion results queried by the reviewer appear to be directly linked to cAMP generation.

Figure Rb2. Cyclic AMP production with prolonged agonist exposure in INS-1 832/3 cells. (a) Dose response for cAMP production with 500 μ M IBMX added for final 10 minutes of 16-hour incubation with indicated agonist, expressed as fold change relative to IBMX alone, $n=5$. (b) Data from (a) plotted against agonist internalization E_{\max} (Supplementary Fig. 1b). Error bars indicate SEM.

Next, we stimulated INS-1 832/3 cells with maximal concentrations (1 μ M) of each agonist for 16 h, before adding IBMX with or without a high concentration (1 μ M) of exendin-4. Here, we tested the maximal GLP-1R activation response after pre-incubation with different compounds, as exendin-4

should maximally activate any remaining receptors. In contrast to the biased agonist-only response (no additional exendin-4), we observed that additional response induced by exendin-4 was related to internalisation rate of the pre-incubated compound throughout the entire range, with the greatest response seen after pre-incubation with exendin-dTyr1 (see Fig. 3e in the updated manuscript). This experiment suggests that agonist internalisation propensity does indeed predict on-going beta cell responsiveness, albeit with a threshold beyond which weakly-internalising agonists are no longer able to fully exploit the greater number of residual surface receptors.

We hope the reviewer agrees that the above experiments provide further mechanistic detail regarding the chronic insulin secretory effects of each agonist, which appear to depend jointly on their ability to retain GLP-1R at the plasma membrane, but also on their ability to activate the pool of available receptors. Thus, there is a threshold beyond which further increases in receptor availability at the plasma membrane are counteracted by reductions in agonist behaviour, i.e. the ability to activate a given receptor molecule once bound. From a translational point of view, an agonist such as exendin-phe1 appears to possess the optimal combination between preservation of surface GLP-1Rs and adequate ability to maximally activate them, with only modest reductions in pharmacological potency, rendering it most effective at concentrations achievable *in vivo*. Note also that in our manuscript we utilised additional techniques to modulate receptor trafficking, including β -arrestin knockdown / knockout (which reduced endocytosis and increased insulin secretion) and allosteric modulation with BETP (which reduced both recycling and insulin secretion).

Overall, whilst we accept the fact that all techniques employed (biased agonists, genetic and pharmacological knockdown, etc.) exert actions beyond their effects on trafficking, the overall body of evidence we present is consistent with a major role for GLP-1R trafficking in determining prolonged agonist responses.

Finally, in many instances, providing the raw data rather than or in addition to normalized data would allow a better estimate of the robustness of the assays and reproducibility of the data.

We now provide non-normalised data in Supplementary Fig. 10. Rather than duplicate all graphs from the manuscript, which is likely to negatively impact readability, we have selected several of the key experiments from the main figures and present the non-normalised data here. Note that these figures represent experimental techniques used elsewhere in the manuscript, and, as such, the non-normalised data provide an estimate of assay robustness, as suggested by the reviewer. The table below indicates which figures have been re-plotted using non-normalised data. We suggest that, whilst some modest increases in variability are apparent, this does not change the main conclusions from the data presented.

Normalised data figure	Non-normalised data figure
Fig. 1a	Supplementary Fig. 10a
Fig. 1b	Supplementary Fig. 10b
Fig. 1f	Supplementary Fig. 10c
Fig. 1g	Supplementary Fig. 10d
Fig. 2b	Supplementary Fig. 10e
Fig. 3e	Supplementary Fig. 10f
Fig. 4c	Supplementary Fig. 10g
Fig. 6a	Supplementary Fig. 10h
Fig. 6e	Supplementary Fig. 10i
Fig. 6f	Supplementary Fig. 10j

Minor: In Sup Fig 1, the LogEC50 should be (M) and not (nM). In Fig 1 it is not clear why statistical differences are indicated in panel h an l and not f and g.

These have now been addressed. Note that statistical significance for Fig. 1f was already provided in Supplementary Fig. 1b, but has now been added to the main figure for clarity.

Reviewer #3 (Remarks to the Author):

The revised manuscript has adequately addressed the points raised in the original review. There are minor, primarily aesthetic points that need to be addressed:

1. Figure 3b and 3d. It is not clear why these data in MIN6B1 cells are not presented as CRCs as they are for INS1 cells in Figs. 3a and 3c. The figure legend states that the data in (b) and (d) are presented as in (a) and (c), respectively, but this is not the case. It is not clear what the concentration of agonists were used in Figs. 3b and 3d.

We have generally performed concentration-response experiments in INS-1 832/3 cells but, having established differences in pharmacological efficacy as a characteristic feature of our biased agonists, we aimed to corroborate findings with a single maximal 100 nM dose in MIN6B1 cells. The legend has now been corrected.

2. The data in Supplementary Figure 9 that include how ex-phe1 compare to the rest of the agonists should really be shown in the main text.

We have moved this figure into Fig. 6, as suggested. Non-normalised data are now found in Supplementary Fig. 10.

References

1. Halford, J. C., Wanninayake, S. C. & Blundell, J. E. Behavioral satiety sequence (BSS) for the diagnosis of drug action on food intake. *Pharmacol. Biochem. Behav.* **61**, 159–168 (1998).
2. Cooke, J. H. *et al.* Peripheral and central administration of xenin and neurotensin suppress food intake in rodents. *Obesity (Silver Spring)* **17**, 1135–1143 (2009).
3. Wright, F. L. & Rodgers, R. J. Behavioural profile of exendin-4/naltrexone dose combinations in male rats during tests of palatable food consumption. *Psychopharmacology (Berl.)* **231**, 3729–3744 (2014).
4. Mack, C. M. *et al.* Antiobesity action of peripheral exenatide (exendin-4) in rodents: effects on food intake, body weight, metabolic status and side-effect measures. *Int J Obes (Lond)* **30**, 1332–1340 (2006).
5. Kanoski, S. E., Rupperecht, L. E., Fortin, S. M., De Jonghe, B. C. & Hayes, M. R. The role of nausea in food intake and body weight suppression by peripheral GLP-1 receptor agonists, exendin-4 and liraglutide. *Neuropharmacology* **62**, 1916–1927 (2012).
6. Lachey, J. L. *et al.* The role of central glucagon-like peptide-1 in mediating the effects of visceral illness: differential effects in rats and mice. *Endocrinology* **146**, 458–462 (2005).
7. Gazi, L. *et al.* Receptor density as a factor governing the efficacy of the dopamine D4 receptor ligands, L-745,870 and U-101958 at human recombinant D4.4 receptors expressed in CHO cells. *Br J Pharmacol* **128**, 613–620 (1999).
8. Hermans, E., Challiss, R. A. & Nahorski, S. R. Effects of varying the expression level of recombinant human mGlu1alpha receptors on the pharmacological properties of agonists and antagonists. *Br J Pharmacol* **126**, 873–882 (1999).
9. Knudsen, L. B., Hastrup, S., Underwood, C. R., Wulff, B. S. & Fleckner, J. Functional importance of GLP-1 receptor species and expression levels in cell lines. *Regul Pept* **175**, 21–29 (2012).

Reviewers' Comments:

Reviewer #2:

None